# Modular Learning of Deep Causal Generative Models for High-dimensional Causal Inference

## Abstract

Pearl's causal hierarchy establishes a clear separation between observational, interventional, and counterfactual questions. Researchers proposed sound and complete algorithms to compute identifiable causal queries at a given level of the hierarchy using the causal structure and data from the lower levels of the hierarchy. However, most of these algorithms assume that we can accurately estimate the probability distribution of the data, which is an impractical assumption for high-dimensional variables such as images. On the other hand, modern generative deep learning architectures can be trained to learn how to accurately sample from such high-dimensional distributions. Especially with the recent rise of foundation models for images, it is desirable to leverage pre-trained models to answer causal queries with such high-dimensional data. To address this, we propose a sequential training algorithm that, given the causal structure and a pre-trained conditional generative model, can train a deep causal generative model, which utilizes the pre-trained model and can provably sample from identifiable interventional and counterfactual distributions. Our algorithm, called WhatIfGAN, uses adversarial training to learn the network weights, and to the best of our knowledge, is the first algorithm that can make use of pre-trained models and provably sample from any identifiable causal query in the presence of latent confounders with high-dimensional data. We demonstrate the utility of our algorithm using semi-synthetic and real-world datasets containing images as variables in the causal structure.

## 1 Introduction

Evaluating the causal effect of an intervention on a system of interest, or understanding what would have happened to a sub-population had a different intervention been taken are fundamental questions that arise across disciplines. Pearl's structural causal models (SCMs) provide a principled approach to answering such queries from data. Using SCMs, today we have a clear understanding of which causal queries can be answered from data on a fundamental level, and which cannot without further assumptions Pearl (1995); Shpitser & Pearl (2008); Huang & Valtorta (2012); Bareinboim & Pearl (2012b). The associated *identification algorithms* find a closed-form expression for an interventional distribution using only observational data, by making use of the causal structure via do-calculus rules Pearl (1995). With sufficient data, these functionals can be evaluated by first estimating the observational joint distribution, then evaluating these expressions, which are also called the *estimands*.

Today, most our datasets contain high-dimensional variables, such as images. With the advancement of modern deep learning architectures, many machine learning problems that require handling high dimensional data, such as prediction, or training generative models, are effectively solved in practice. However, the existing causal inference algorithms are unable to handle high-dimensional variables, i.e., there is no general causal inference algorithm for arbitrary causal structures that can use high-dimensional data. The aforementioned identification algorithms are not applicable since it is practically impossible to obtain a closed-form expression of the joint distribution with high-dimensional image datasets. Other causal inference methods either assume no unobserved confounders Kocaoglu et al. (2018), or solve the problem only for very specific structures Louizos et al. (2017) or causal queries Nemirovsky et al. (2020).

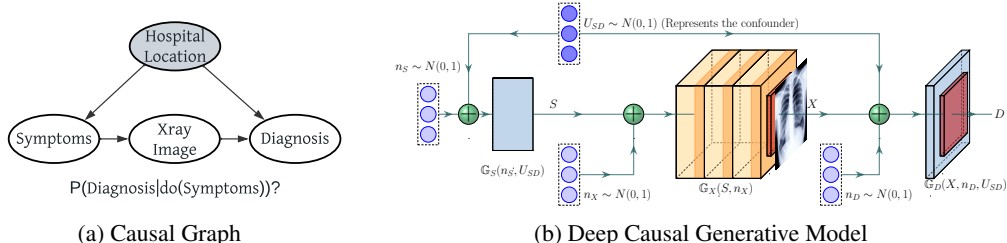

(a) Causal Graph  (b) Deep Causal Generative Model

Figure 1: A causal graph for XrayImage example (left) and its deep generative model representation (right). For each variable, a NN (ex: $\mathbb{G}_S, \mathbb{G}_X, \mathbb{G}_D$) is trained on datasets to mimic the true mechanism.

As an example, consider a healthcare dataset where we observe $\mathrm{Symptoms, Diagnosis}$ and $\mathrm{XrayImage}$ of a collection of patients. Suppose our dataset contains a mixture data from two hospitals in different cities, which we do not observe to ensure patient privacy. One of the cities is more wealthy, where patients are less likely to fall sick on average, and have access to better-trained doctors who are less likely to misdiagnose them. Then, hospital location acts as a latent confounder that affect both the $\mathrm{Symptoms}$ and $\mathrm{Diagnosis}$. Suppose the medical X-ray device is standardized by the World Health Organization and is not affected by the wealth of the cities. In this realistic scenario, the data-generating process can be summarized by the causal graph in Figure 1. We would like to understand how likely an average person across the two cities is to be diagnosed if they get the symptom. This is the well-known *front-door* graph Pearl (2009) where the causal effect of $\mathrm{Symptom}$ on $\mathrm{Diagnosis}$ can be computed from the observational distribution over $\mathrm{Symptom, Diagnosis}$ and $\mathrm{XrayImage}$ as follows: $\sum_{x,s'} p(x|s)p(d|x,s')p(s')$. However, it is not possible to reliably estimate $p(x|s)$, i.e., the probability of observing *a specific X-ray image*, given $\mathrm{Symptom} = s$. To the best of our knowledge, no existing causal inference algorithm can address this causal inference problem.

In this paper, we propose a sampling-based solution to address this problem. Our solution is general and not limited to the front-door graph. For causal queries with scalar, or low-dimensional targets, we can collect samples from the correct interventional distribution and use these samples to estimate the interventional distribution. For high-dimensional targets, our method can be used to obtain samples from the interventional distribution, which is implicitly modeled. Our solution uses deep learning architectures that mimic the causal structure of the system, see Figure 1b. Although the use of such structured deep generative models have been explored recently, the existing solutions either assume no unobserved confounders Kocaoglu et al. (2018), or assume all variables are discrete Xia et al. (2021; 2023). Furthermore, they cannot use pretrained image generators. This is useful since state-of-the-art deep image geneators can only be successfully trained by a few industrial research labs with access to the required resources DALL-E (2022); ChatGPT (2022); Bard (2023).

Our key contribution is a modular training algorithm that can identify which parts of the deep causal generative models, such as the one in Figure 1b, can be trained separately, and which parts should be trained together. We show that after this modularization, there is a *correct* training order for each sub-network, which our algorithm identifies and follows, freezing the weights of networks already trained in the previous steps of the algorithm. The following are our main contributions:

- We propose an adversarial learning algorithm for training deep causal generative models with latent confounders for high-dimensional variables. We show that, after convergence, our model can produce high-dimensional samples according to interventional or counterfactual queries that are identifiable from the data distributions.

- We propose an algorithm to train deep causal generative models in a modular manner using both observational and interventional datasets while preserving their theoretical guarantees after convergence. To the best of our knowledge, this is the first algorithm that can modularize the training process in the presence of latent confounders, thereby enabling the use of large pre-trained models for causal effect estimation.

- We demonstrate the utility of WhatIfGAN with experiments on high-dimensional semi-synthetic datasets Colored-MNIST and real-world COVIDx CXR-3 dataset. Modular training converges better and can correctly sample from interventional and counterfactual distributions compared to two of our closest benchmarks.

## 2 RELATED WORKS

There are many principled algorithms for estimating interventional or counterfactual distributions in different settings. Shpitser & Pearl (2008) estimate causal effects from observational data using their sound and complete identification algorithms. Bareinboim & Pearl (2012b) proposes zID that utilizes observational and all combinations of experimental datasets to identify causal effects that are not possible to estimate from only observvations. Lee et al. (2020) covers both of these problems and suggests the gID algorithm that can make use of arbitrary sets of observational and interventional datasets to estimate a causal query. Similarly, Correa et al. (2021) employs arbitrary combinations of datasets for complete counterfactual identification.

Louizos et al. (2017); Nemirovsky et al. (2020) offer to solve the causal inference problem using deep generative models. Yet, they do not offer theoretical guarantees of causal estimation in general, but for some special cases. Researchers have recently focused on imposing causal structures within neural network structures. Particularly, Kocaoglu et al. (2018) introduces a deep causal model that produces interventional image samples after training on observational data. Pawlowski et al. (2020) applies normalizing flows and variational inference to predict exogenous noise for counterfactual inference. However, a major limitation is the causally sufficiency assumption, i.e., each observed variable is caused by independent unobserved variables. In contrast, the semi-Markovian model having shared unobserved confounders between variables, is quite common in the real world.

For semi-Markovian models, Xia et al. (2021); Balazadeh Meresht et al. (2022) follow a similar approach as Kocaoglu et al. (2018) to arrange neural models as a causal graph. They propose a minimization-maximization method to identify and estimate causal effects. However, these methods use only observational data and do not consider interventional data and hence cannot estimate most counterfactuals. Xia et al. (2023) extends these to identify and estimate counterfactual queries.

Most of the existing methods described above can only handle low-dimensional discrete variables except Bica et al. (2020) and it is not clear how to extend their results to continuous high-dimensional image data. Moreover, if these methods are given a pre-trained neural network model, they do not have the ability to incorporate them in their training. Our approach, to the best of our knowledge, is the first and only solution to address this problem in the presence of unobserved confounders, unlocking the potential of large pre-trained models for causal inference.

## 3 BACKGROUND

**Definition 3.1** (Structural causal model (SCM) (Pearl, 2009)). An SCM $\mathcal{M}$ is a 5-tuple $\mathcal{M} = (\mathcal{V}, \mathcal{N}, \mathcal{U}, \mathcal{F}, P(.))$, where each observed variable $V_i \in \mathcal{V}$ is realized as an evaluation of the function $f_i \in \mathcal{F}$ which looks at a subset of the remaining observed variables $Pa_i \subset \mathcal{V}$, an unobserved exogenous noise variable $E_i \in \mathcal{N}$, and an unobserved confounding variable $U_i \in \mathcal{U}$. $P(.)$ is a product joint distribution over all unobserved variables $\mathcal{N} \cup \mathcal{U}$.

Each SCM induces a directed graph called the *causal graph*, or acyclic directed mixed graph (ADMG) with $\mathcal{V}$ as the vertex set. The directed edges are determined by which variables directly affect which other variable by appearing explicitly in that variable's function. Thus the causal graph is $G = (V, E)$ where $V_i \rightarrow V_j$ iff $V_i \in Pa_j$. The set $Pa_j$ is called the parent set of $V_j$. We assume this directed graph is acyclic (DAG). Under the semi-Markovian assumption, each unobserved confounder can appear in the equation of exactly two observed variables. We represent the existence of an unobserved confounder between $X, Y$ in the SCM by adding a bidirected edge $X \leftrightarrow Y$ to the causal graph. These graphs are no longer DAGs although still acyclic.

$V_i$ is called an ancestor for $V_j$ if there is a directed path from $V_i$ to $V_j$. Then $V_j$ is said to be a descendant of $V_i$. The set of ancestors of $V_i$ in graph $G$ is shown by $An_G(V_i)$. A do-intervention $do(v_i)$ replaces the functional equation of $V_i$ with $V_i = v_i$ without affecting other equations. The distribution induced on the observed variables after such an intervention is called an interventional distribution, shown by $P_{v_i}(\mathcal{V})$. $P_\emptyset(\mathcal{V}) = P(\mathcal{V})$ is called the observational distribution. In this paper, we use $\mathcal{L}_1, \mathcal{L}_2$, and $\mathcal{L}_3$ distributions as notation for observational, interventional, and counterfactual distributions respectively. We follow the same for datasets and queries.

**Definition 3.2** (c-components). A subset of nodes is called a c-component if it is a maximal set of nodes in $G$ that are connected by bi-directed paths.

# 4 DEEP CAUSAL GENERATIVE MODELS WITH UNOBSERVED CONFOUNDERS

The basic idea behind Kocaoglu et al. (2018) that also motivates our work is the following: Suppose the ground truth data-generating SCM is made up of functions $X_i = f_i(Pa_i, E_i)$. If we have these equations, we can simulate an intervention on, say $X_5 = 1$, by evaluating the remaining equations. However, we can never hope to learn the functions and unobserved noise terms from data. The fundamental observation of Pearl is that even then there are some causal queries that can be uniquely identified as some *deterministic function* of the causal graph and the joint distribution between observed variables, e.g., $p(d|do(s)) = \xi(G, p(s, d, x))$ in Figure 1a for some deterministic $\xi$. This means that, if we can, somehow, train a causal model made up of neural networks that fits to the data, and has the same causal graph, then it has to induce the same interventional distribution $p(d|do(s))$ as the ground truth SCM, *irrespective of what functions the neural network uses*. This is a very strong idea that allows mimicking the causal structure, and opens up the possibility of using deep learning algorithms for performing causal inference through sampling even with high-dimensional variables.

Xia et al. (2021) took this idea one step further to *check identifiability of a causal query* by maximizing and minimizing the causal effect, which is now a differentiable function since neural nets are used. However, training such networks with high-dimensional data is challenging, and their work is limited to low-dimensional variables. GANs have been successfully applied to approximately fit high-dimensional image data, which is our training algorithm choice in this paper to handle high-dimensional data. We first define a *deep causal generative model* and show identifiability results, formalizing the simple observation above that allows us to use deep learning to answer causal queries.

**Definition 4.1.** A neural net architecture $\mathbb{G}$ is called a deep causal generative model (DCM) for an ADMG $G = (\mathcal{V}, \mathcal{E})$ if it is composed of a collection of neural nets, one $\mathbb{G}_i$ for each $V_i \in \mathcal{V}$ such that

1. *each $\mathbb{G}_i$ accepts a sufficiently high-dimensional noise vector $N_i$,*

2. *the output of $\mathbb{G}_j$ is input to $\mathbb{G}_i$ iff $V_j \in Pa_G(V_i)$,*

3. $N_i = N_j$ iff $V_i \leftrightarrow V_j$.

Noise vectors $N_i$ replace both the exogenous noise terms in the true SCM and the unobserved confounders. They need to have sufficiently high dimension to be able to induce the observed distribution. We say a DCM is *representative enough for an SCM* if the neural networks have sufficiently many parameters to induce the observed distribution induced by the true SCM. We assume this in this work. With this definition, we have the following, similar to Xia et al. (2021):

**Theorem 4.2.** *Consider any SCM $\mathcal{M} = (G, \mathcal{N}, \mathcal{U}, \mathcal{F}, P(.))$. A DCM $\mathbb{G}$ for $G$ entails the same identifiable interventional distributions as the SCM $\mathcal{M}$ if it entails the same observational distribution.*

Although the theorem focuses on interventional distributions, it can trivially be generalized to any counterfactual distribution by expanding the notion of identifiability to use a given collection of interventional distributions, and requiring the DGM to entail the same interventional distributions for the said collection. Please see Theorem C.3 in Appendix C.3 for the general statement and proof.

Thus, even with high-dimensional variables in the true SCM, given a causal graph, any identifiable interventional query can, in principle, be sampled from by training a DCM that fits to the observational distribuiton. However, trianing both low and high dimensional variables is challenging, and any modularization not only is expected to help train more easily, but also allows the possibility of using pre-trained image generative models. We focus on uncovering how to achieve such modularization.

## 4.1 BASICS OF MODULAR TRAINING WITH UNOBSERVED CONFOUNDERS

Consider the graph $G$ in Figure 2. Suppose, we have an observational dataset $D \sim P(\mathcal{V})$. Our goal is to sample from different $\mathcal{L}_2$ distributions such as $P(X_2|do(X_1))$ and $P(Z_2|do(X_1))$ by training a DCM $\mathbb{G}$ for $G$ that fits the observational data $P(\mathcal{V})$, in accordance with Theorem 4.2.The DCM will contain one feed-forward neural net per observed variable, i.e., $\mathbb{G} = \{\mathbb{G}_{Z_1}, \mathbb{G}_{Z_2}, \mathbb{G}_{Z_3}, \mathbb{G}_{X_1}, \mathbb{G}_{X_2}\}$. The question we are interested in is, which neural nets can be trained separately, and which need to be trained together *to be able to fit the joint distribution*. Let $Q$ be the distribution induced by the DCM.

Suppose we first train the causal generative model $\mathbb{G}_{Z_1}$, i.e., learn a mapping that can sample from $P(Z_1|Z_3, X_1)$. Even if we provide the unobserved confounder $N_1$ (see Definition 4.1), which also

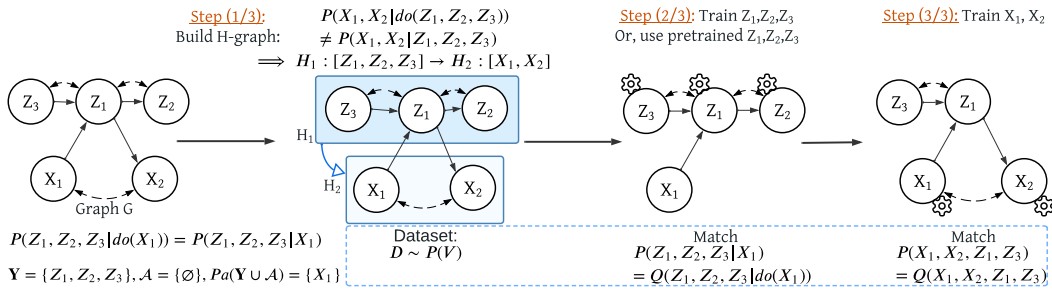

Figure 2: Modular training on H-graph: $H_1 : [Z_1, Z_2, Z_3] \to H_2 : [X_1, X_2]$ with dataset $D \sim P(V)$.

affects $Z_2$ and $Z_3$, the neural network can learn a mapping that later make it impossible to induce the correct dependence between $Z_1, Z_2$, or $Z_1, Z_3$ no matter how $\mathbb{G}_{Z_2}$ or $\mathbb{G}_{Z_3}$ are trained later. This is because fitting the conditional $P(Z_1|Z_3, X_1)$ does not provide any incentive for the model to induce the correct dependency through latent variables. If model ignores the dependence due to the latent confounders, it cannot induce dependence between $Z_3$ and $Z_2$ conditioned on $Z_1$. This observation suggests that the causal mechanisms of variables that are in the same c-component should be trained together. Therefore, we have to train $[\mathbb{G}_{Z_1}, \mathbb{G}_{Z_2}, \mathbb{G}_{Z_3}]$ together; similarly $[\mathbb{G}_{X_1}, \mathbb{G}_{X_2}]$ together.

Another issue is that it is very difficult to *condition* in feedforward models during training, which is the case in a DCM. To sample from $Q(z_1|z_3, x_1)$ it is not sufficient to feed $z_3, x_1$ to the network $\mathbb{G}_{Z_1}$. In fact, observe that this is exactly the intervention operation and we would give us a sample from $Q(Z_1|do(Z_3, X_1))$. Thus, while training DCMs, it is trivial to intervene on the inputs to a neural network, but highly non-trivial to condition since feedforward models cannot easily be used to correctly update the posterior via backdoor paths. This, together with the previous observation motivates us to fitting interventional distributions a central part of our algorithm, as we explain below:

To match the joint $P(V)$ for semi-Markovian models while preserving the integrity of c-components, we propose using Tian's factorization (Tian & Pearl, 2002). It factorizes $P(V)$ into c-factors: the joint distributions of each c-component $C_j$ intervened on their parents, i.e., $P_{pa(C_j)}(C_j)$.

$$P(v) = P(x_1, x_2|\text{do}(z_1))P(z_1, z_2, z_3|\text{do}(x_1)) \tag{1}$$

Due to this factorization, matching $P(V)$ is equivalent to matching each of the c-factors. If we had access to the $\mathcal{L}_2$ distributions from $\text{do}(z_1)$ and $\text{do}(x_1), \forall z_1, x_1$, we could intervene on $\mathbb{G}_{Z_1}$ and $\mathbb{G}_{X_1}$ in the DCM to obtain $\text{do}(z_1)$ and $\text{do}(x_1)$ samples and train the generative models to match these $\mathcal{L}_2$ distributions. However, we only have access to the $P(V)$ dataset. Our key idea is to **leverage the do-calculus rule-**2 (Pearl, 1995) to use observational samples and pretend that they are from these $\mathcal{L}_2$ distributions. This gives us a handle on how to modularize the training process of c-components.

For example, in Figure 2, $P(z_1, z_2, z_3|\text{do}(x_1)) = P(z_1, z_2, z_3|x_1)$ since do-calculus rule-2 applies, i.e., intervening on $X_1$ is equivalent to conditioning on $X_1$. We can then use the conditional distribution as a proxy/alternative to the c-factor to learn $Q(z_1, z_2, z_3|\text{do}(x_1))$ with the DCM. However, $P(x_1, x_2|\text{do}(z_1)) \neq P(x_1, x_2|z_1)$. To overcome this issue, we seek to fit a joint distribution that implies this c-factor, i.e., we find a superset of $X_1, X_2$ on which rule-2 applies. We can include $Z_1$ into the joint distribution that needs to be matched together with $X_1, X_2$ and check if the parent set of $\{X_1, X_2, Z_1\}$ satisfy rule-2. We continue until we reach the joint $P(x_1, x_2, z_1, z_3)$ to be the alternative distribution for $\{X_1, X_2\}$'s c-factor.

After identifying which sub-networks of the DCM can be trained separately, we need to decide *a valid order* in which they should be trained. For the same example, we can first train $[\mathbb{G}_{Z_1}, \mathbb{G}_{Z_2}, \mathbb{G}_{Z_3}]$ together to induce $Q(z_1, z_2, z_3|do(x_1)) = P(z_1, z_2, z_3|x_1) = P(z_1, z_2, z_3|do(x_1))$. This is shown in step (2/3) in Figure 2: We can produce samples from the mechanisms of $Z_1, Z_2, Z_3$ by intervening on their parent $X_1$ with real observations from dataset $D$. Thus, we do not need $\mathbb{G}_{X_1}$ to be pre-trained.

Now, we train mechanisms of the next c-component $[\mathbb{G}_{X_1}, \mathbb{G}_{X_2}]$ in our training order (step 3/3). As discussed, we need to ensure $Q(X_1, X_2, Z_1, Z_3|do(\emptyset)) = P(X_1, X_2, Z_1, Z_3|do(\emptyset)) = P(X_1, X_2, Z_1, Z_3|\emptyset)$. Since mechanisms of $Z_1, Z_3$ were trained in the previous step, we can freeze the neural network weights of $[\mathbb{G}_{Z_1}, \mathbb{G}_{Z_3}]$. These are used to correctly sample from $Z_1$ given $X_1$, and feed this correctly sampled value into the network of $X_2$. In Appendix D.1, we show that the c-factors

in Equation 1 will correctly match the true c-factors after fitting these two conditional probabilities in this order. Therefore, DCM matches the joint distribution $P(\mathcal{V})$ as well. On the other hand, if we first trained the networks $[\mathbb{G}_{X_1}, \mathbb{G}_{X_2}]$, it would not be possible to match the joint distribution $P(x_1, x_2, z_1, z_3)$ as the mechanisms of $Z_1, Z_3$ are not yet trained. Thus, this order would not work.

In the following, we generalize these ideas into an algorithm that can modularize the training process of different c-components, while identifying a valid training order to match the joint distribution.

## 4.2 Modular Training Algorithm for DCMs

To match the joint distribution $P(\mathcal{V})$ by training the DCMs in a modular way, we have to: 1) arrange the c-components in a valid training order and 2) train (sets of) c-components to match their c-factors.

***Arranging the c-components***: Consider a c-component $C_t$. When should a c-component $C_s$ be trained before $C_t$? Since we need rule 2 of do-calculus to hold on the parents of $C_t$ for training, if $C_s$ contains some parents of $C_t$ that are located on the backdoor paths between any two variables in $C_t$, then $C_s$ must be pre-trained before $C_t$. Conditioning and intervening on those parents of $C_t$ in $C_s$ is not the same, i.e., $P(C_t|do(pa(C_t \cap C_s))) \neq P(C_t|pa(C_t \cap C_s))$. Thus we include them in the joint distribution that we want to match for $C_t$, which requires those parents in $C_s$ to be pre-trained.

To obtain a partial order among all c-components, we construct a directed graph structure called $\mathcal{H}$-graph that contains c-components as nodes. While adding edges, if any cycle is formed, we merge c-components on that cycle into a single h-node indicating that they will need to be trained jointly. Thus some h-nodes may contain more than one c-component. The final structure is a DAG and gives us a valid partial order $\mathcal{T}$ for modular training (Proposition D.14). Formally an $\mathcal{H}$-graph is defined as:

**Definition 4.3** ($\mathcal{H}$-graph). Given a causal graph $G$ with c-components $\mathcal{C} = \{C_1, \dots C_n\}$, let $\{H_k\}_k$ be some partition of $\mathcal{C}$. The directed graph $(V_{\mathcal{H}}, E_{\mathcal{H}})$ where $V_{\mathcal{H}} = \{H_k\}_k$ and $H_s \rightarrow H_t \in E_{\mathcal{H}}$ iff $P(H_t|do(pa_G(H_t) \cap H_s)) \neq P(H_t|pa_G(H_t) \cap H_s)$, is called an $\mathcal{H}$-graph for $G$ if it is acyclic.

We run the subroutine $\text{Contruct\_Hgraph}()$ in Algorithm 1 to build an $\mathcal{H}$-graph. We check the edge condition in line 5 and merge cycles in line 7 if any. In Figure 2 step (1/3), we build the $\mathcal{H}$-graph $H_1 : [Z_1, Z_2, Z_3] \rightarrow H_2 : [X_1, X_2]$ for $G$. For a larger example, please see Appendix E.2. Note that, we only use the $\mathcal{H}$-graph to obtain a partial training order via h-nodes. $An(H_k)$ and $Pa(H_k)$ below refer to ancestors and parents in the causal graph $G$, not in the $\mathcal{H}$-graph, for any h-node $H_k$.

***Training c-components***: We follow $\mathcal{H}$-graph's topological order and train the c-components in an h-node $H_k$ to match their corresponding c-factors $P_{pa(C_j)}(C_j), \forall C_j \in H_k$ and matching $P_{pa(H_k)}(H_k)$ ensures this. Although we can generate fake interventional samples from $Q_{pa(H_k)}(H_k)$ induced by the DCM, they can not be used to train $\mathbb{G}_{H_k}$ as we do not have access to real data samples from the interventional distribution $P_{pa(H_k)}(H_k)$. Thus, we train $\mathbb{G}_{H_k}$ to learn a larger joint distribution that can be obtained from the observational dataset as an alternative to its c-factors. We search for a set $\mathcal{A}_k$ that can be added to the joint with $H_k$ such that $P_{Pa(H_k, \mathcal{A}_k)}(H_k, \mathcal{A}_k) = P(H_k, \mathcal{A}_k|Pa(H_k, \mathcal{A}_k))$, i.e., true interventional and conditional distribution are the same. This enables us to take conditional samples from the input dataset and use them as interventional data samples and match their distribution with the DCM-generated fake interventional samples from $Q_{Pa(H_k, \mathcal{A}_k)}(H_k, \mathcal{A}_k)$ to train $\mathbb{G}_{H_k}$. The above condition is generalized into a **modularity condition** below:

**Definition 4.4.** Let $H_k$ be an h-node in the $\mathcal{H}$-graph. A set $\mathcal{A}_k \subseteq An_G(H_k) \backslash H_k$ satisfies the modularity condition if it is the smallest set with $P(H_k, \mathcal{A}_k|do(Pa(H_k, \mathcal{A}_k))) = P(H_k, \mathcal{A}_k|Pa(H_k, \mathcal{A}_k))$.

As mentioned, such $\mathcal{L}_1$ and $\mathcal{L}_2$ distributional equivalence holds when the do-calculus rule-2 applies.

$$P_{Pa(H_k, \mathcal{A}_k)}(H_k, \mathcal{A}_k) = P(H_k, \mathcal{A}_k|Pa(H_k, \mathcal{A}_k)), \quad \text{if } (H_k, \mathcal{A}_k \perp\!\!\!\perp Pa(H_k, \mathcal{A}_k))_{G_{\underline{Pa(H_k, \mathcal{A}_k)}}} \quad (2)$$

This suggests a graphical criterion to find such a set $\mathcal{A}_k$. Intuitively, if the outgoing edges of $Pa(H_k, \mathcal{A}_k)$ are deleted ($G_{\underline{Pa(H_k, \mathcal{A}_k)}}$), they become d-separated from $\{H_k, \mathcal{A}_k\}$, i.e., there exists no backdoor path from $Pa(H_k, \mathcal{A}_k)$ to $\{H_k, \mathcal{A}_k\}$. Therefore, for a specific $H_k$, we start with $\mathcal{A}_k = \emptyset$ and check if $Pa(H_k, \mathcal{A}_k)$ satisfies the conditions of rule-2 for $\{H_k, \mathcal{A}_k\}$. If not, we add parents of $\{H_k, \mathcal{A}_k\}$ in $\mathcal{A}_k$. We include the ancestors since only they can affect $H_k$'s mechanisms from outside of the c-component. We continue the process until $Pa(H_k, \mathcal{A}_k)$ satisfies the condition of rule-2.

After constructing the $\mathcal{H}$-graph at Algorithm 2 in line 3, we train each h-node $H_k$ according to the partial order to match with the c-factors' alternative distribution for the c-components in $H_k$.

---

**Algorithm 1** Construct_Hgraph($G$)

1: **Input:** Causal Graph $G$
2: $\mathcal{C} \leftarrow$ get_ccomponents($G$)
3: Create nodes $H_k = C_k$ in $\mathcal{H}, \forall C_k \in \mathcal{C}$
4: **for each** $H_s, H_t \in \mathcal{H}$ such that $s \neq t$ **do**
5:    **if** $P(H_t | \text{do}(pa(H_t) \cap H_s))$
      $\neq P(H_t | pa(H_t) \cap H_s)$ **then**
6:       $\mathcal{H}.add(H_s \rightarrow H_t)$
7: $\mathcal{H} \leftarrow$ Merge($\mathcal{H}, cyc$), $\forall cyc \in Cycles(\mathcal{H})$
8: **Return:** $\mathcal{H}$

**Algorithm 2** Modular Training($G, \mathbf{D}$)

1: **Input:** Causal Graph $G$, Dataset $\mathbf{D}$.
2: Initialize DCM $\mathbb{G}$
3: $\mathcal{H} \leftarrow$ Construct_Hgraph($G$)
4: **for each** $H_k \in \mathcal{H}$ in partial order   **do**
5:    Initialize $\mathcal{A}_k \leftarrow \emptyset$
6:    **while** IsRule2($H_k, \mathcal{A}_k$) = 0 **do**
7:       $\mathcal{A}_k \leftarrow Pa_G(H_k, \mathcal{A}_k)$
8:       $\mathbb{G}_{H_k} \leftarrow$ TrainModule($\mathbb{G}_{H_k}, G, H_k, \mathcal{A}_k, \mathbf{D}$)
9: **Return:** $\mathbb{G}$

---

(lines 4-8). We initialize a set $\mathcal{A}_k = \emptyset$ to keep track of the joint distribution we need to match to train each h-node $H_k$. We search for the set of ancestors $\mathcal{A}_k$ of the current h-node $H_k$ such that $\mathcal{A}_k$ satisfies the modularity condition for $H_k$ (line 6), which is checked by Algorithm 5: IsRule2(.). Finding a set $\mathcal{A}_k$ satisfying the modularity condition implies that we can train $\mathbb{G}_{H_k}$ by matching:

$$Q_{pa(H_k, \mathcal{A}_k)}(H_k, \mathcal{A}_k) = P(H_k, \mathcal{A}_k | Pa(H_k, \mathcal{A}_k)); \text{Now training: } H_k, \text{Pre-trained: } \mathcal{A}_k \quad (3)$$

We utilize adversarial training to train the generators in $\mathbb{G}_{H_k}$ on observational dataset $\mathbf{D}$ to match the above. This is done by Algorithm 4: TrainModule() called in line 8. More precisely, this sub-routine uses all mechanisms in $\{H_k, \mathcal{A}_k\}$ to produce samples but only updates the mechanisms in $\mathbb{G}_{H_k}$ corresponding to the current h-node and returns those models after convergence. Even though we will train only $\mathbb{G}_{H_k}$ i.e., $\mathbb{G}_V, \forall V \in H_k$, $\mathcal{A}_k$ appears together with $H_k$ in the joint distribution that we need to match. Thus, we use pre-trained causal mechanisms of $\mathcal{A}_k$, i.e., $\mathbb{G}_V, \forall V \in \mathcal{A}_k$ here. Following the partial order of $\mathcal{H}$-graph ensures that we have already trained $\mathcal{A}_k$ before $H_k$.

Training $\mathbb{G}_{H_k}$ to match the conditional distribution of data with the interventional distribution of the DCM, as in Equation 3, is sufficient to learn the c-factors $P_{Pa(C_j)}(C_j), \forall C_j \in H_k$. After training each $\mathbb{G}_{H_k}$ according to the partial order of $\mathcal{H}$-graph, WhatIfGAN will learn a DCM that induces $Q(\mathcal{V}) = P(\mathcal{V})$ from data. For example, in Figure 2 we match $P(z_1, z_2, z_3 | x_1) = Q(z_1, z_2, z_3 | \text{do}(x_1))$ at step (2/3) and $P(x_1, x_2, z_1, z_3) = Q(x_1, x_2, z_1, z_3)$ at step (3/3). Thus, $P(\mathcal{V}) = Q(\mathcal{V})$ in this graph. Finally, the trained DCM can sample from $\mathcal{L}_2$ and $\mathcal{L}_3$ distributions identifiable from $P(\mathcal{V})$ such as $P(X_2 | \text{do}(X_1))$ or $P(Z_2 | \text{do}(X_1))$. These are formalized in Theorem 4.5. Proof is in Appendix D.6.

***Sampling with WhatIfGAN***: For $\mathcal{L}_2$ sampling, we set the intervened variables to fixed values instead of using their neural networks and push forward those values to generate the rest of the variables.

We have the following assumptions, mainly that the causal graph is given[1] and that GAN training can correctly learn the desired conditional data distribution for each h-node.

**Assumption 1:** The true causal graph with the location of latent confounders is known. **Assumption 2:** The causal graph $G$ is semi-Markovian Tian et al. (2006). **Assumption 3:** For any h-node $H_k$ in the $\mathcal{H}$-graph, GAN training of DCM on $\mathbf{D}$ converges to sample from the conditional distribution $P(H_k, \mathcal{A}_k | pa(H_k, \mathcal{A}_k))$ for all $H_k$, where $\mathcal{A}_k$ is from Algorithm 2.

**Theorem 4.5.** *Consider any SCM $\mathcal{M} = (G, \mathcal{N}, \mathcal{U}, \mathcal{F}, P(.))$. Suppose Assumptions 1-3 hold. Algorithm 2 on $(G, \mathbf{D})$ returns a DCM $\mathbb{G}$ that entails $i$) the same observational distribution, and $ii$) the same identifiable interventional distributions as the SCM $\mathcal{M}$.*

## 5 EXPERIMENTAL EVALUATION

We present WhatIfGAN performance on semi-synthetic high-dimensional Colored-MNIST dataset (LeCun et al., 1998) and real-world COVIDx CXR-3 dataset Wang et al. (2020). In Appendix F, we show performance on Sachs and Colored-MNIST-2 (for counterfactual sampling).

### 5.1 PERFORMANCE ON IMAGE MEDIATOR EXPERIMENT AND BENCHMARK COMPARISON

We show WhatIfGAN performance in an experiment on the front-door graph in Figure 3a that involves both low and high-dimensional variables. We constructed a synthetic SCM where a hidden variable

---

[1]Causal discovery is typically an orthogonal problem to inference, and is beyond our scope in this work.

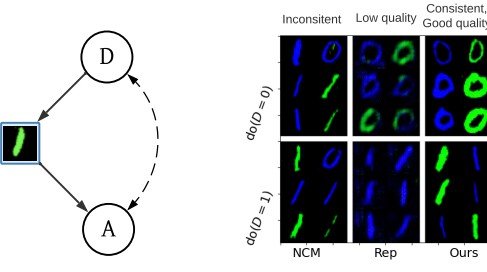
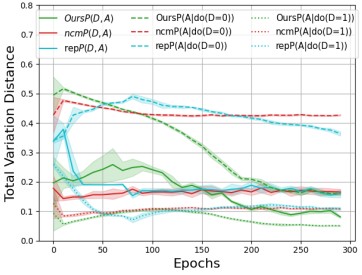

(a) Causal graph    (b) Comparing MNIST generation    (c) Training converges for ours.

Figure 3: Performance on frontdoor. NCM produces good images but not consistent with $do(D)$. WhatIfGAN without modular training (rep) produces consistent but low-quality images. Our modular approach with training order: $\{I\} \to \{D, A\}$ produces consistent, good images and converges faster.

$U$ affects both $D$ and $A$. Image variable $I$ contains the digit value of $D$, and $A$ is some attribute of $I$ obtained from a randomly chosen projection of the image. The digit color can be considered as exogenous noise. Suppose we are given a dataset sampled from $P(D, A, I)$. Our goal is to estimate the causal effect $P(A|do(D))$. We can use the backdoor criterion (Pearl, 1993), to evaluate the ground truth $P(A|do(D)) = \int_U P(A|D, U)P(U)$ since we know $U$ in the true SCM. From the input $\mathcal{L}_1$ dataset, $P(A|do(D))$ is identifiable with the front-door criterion (Pearl, 2009).

The identification algorithm can not be applied since it requires image distribution. If we can train all mechanisms in the DCM to match $P(D, A, I)$, we can produce correct samples from $P(A|do(D))$. For this purpose, we construct the WhatIfGAN architecture with a neural network $\mathbb{G}_D$ having fully connected layers to produce $D$, a CNN-based generator $\mathbb{G}_I$ to generate images, and a classifier $\mathbb{G}_A$ to classify MNIST images into variable $A$ such that $D$ and $A$ are confounded. Now, for this graph, the corresponding $\mathcal{H}$-graph is $[I] \to [D, A]$. Thus, we first train $\mathbb{G}_I$ by matching $P(I|D)$. Instead of training $\mathbb{G}_I$, we can also employ a pre-trained generative model that takes digits $D$ as input and produces an MNIST image showing $D$ digit in it. Thus in Fig 3b, WhatIfGAN shows images of the digit 0 (top, right) and digit 1 (bottom, right) due to do(D = 0) and do(D = 1) intervention. Next, to train $\mathbb{G}_D$ and $\mathbb{G}_A$, we should match the joint distribution $P(D, A, I)$ since $\{I\}$ is ancestor set $\mathcal{A}$ for c-component $\{D, A\}$. GAN convergence becomes difficult using the loss of this joint distribution since the losses generated by low and high dimensional variables are not easily comparable and non-trivial to re-weight (see Appendix F.2). Thus, we map samples of $I$ to a low-dimensional representation, $RI$ with a trained encoder and match $P(D, A, RI)$ instead of the joint $P(D, A, I)$.

In Figure 3b, 3c, we compare our method with Xia et al. (2023): NCM and a version of our method: WhatIfGAN-rep that does not use modular training, with respect to image quality and total variation distance from true $P(D, A)$ and $P(A|do(D))$. We implemented NCM on our architectures as it could not be directly used for images. Since NCM trains all mechanisms with the same loss function calculated from both low and high-dimensional samples, it learns marginal distribution $P(I)$ (Figure 3b left) but does not fully converge to match the joint $P(D, A, I)$ (all red-lines are not going down in Figure 3c). Thus, NCM produces good-quality images but is not consistent with $do(D)$ intervention. WhatIfGAN-rep uses a low-dim representation of images: $RI$ and matches the joint distribution $P(D, A, RI)$ as a proxy to $P(D, A, I)$ without modularization. We observe WhatIfGAN-rep to converge (Figure 3c blue-lines) slower compared to the original WhatIfGAN and produce consistent but low-quality images (Figure 3b middle) for $do(D)$ intervention. Finally, WhatIfGAN modular training matches $P(D, A, RI)$ and converges faster (Figure 3c green-lines) for $P(D, A), P(A|do(D))$ and produces good-quality, consistent $P(I|do(D))$ images (Figure 3b right).

## 5.2 PERFORMANCE ON REAL-WORLD COVIDx CXR-3 DATASET

In this section, we conduct a case study with the COVIDx CXR-3 (Wang et al., 2020) dataset. This dataset contains 30,000 chest X-ray images with Covid $(C)$ and pneumonia $(N)$ labels from over 16,600 patients located in 51 countries. The X-ray images are of healthy patients $(C = 0, N = 0)$, patients with non-Covid pneumonia $(C = 0, N = 1)$, and patients with Covid pneumonia $(C =$

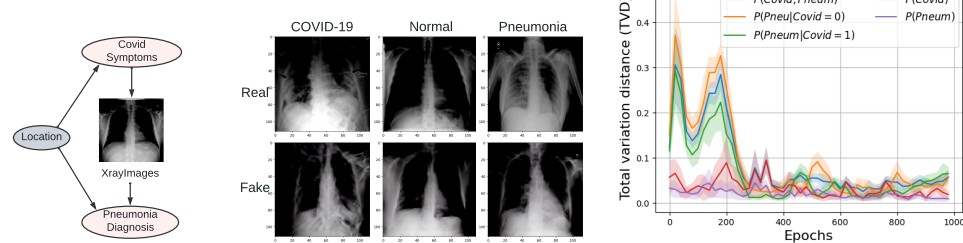

(a) COVIDx CXR-3 graph    (b) Real data vs pre-trained model    (c) Training converges with low TVD

Figure 4: WhatIfGAN converges with pre-trained model on COVIDx CXR-3 dataset.

$1, N = 1$). X-ray images for COVID non-pneumonia ($C = 1, N = 0$) are not present in this dataset as according to health experts those images do not contain enough signal for pneumonia detection.

There is no ground truth causal graph associated with this dataset. In order to demonstrate the convergence behavior of WhatIfGAN on real high-dimensional datasets, we consider the causal graph shown in Figure 4a. Note that the causal effect estimates obtained via this graph may not reflect the true causal effect since the ground truth graph is unknown and there may be other violations of assumptions such as distribution shift and selection bias. However, observe that this graph does not impose conditional independence restrictions on the joint distribution $P(C, Xray, N)$. Therefore, we expect our modular training algorithm to correctly match the observational joint distribution.

Our reasoning for using this causal graph is as follows: we can assume that Covid symptoms determine the X-ray features and the pneumonia diagnosis is made based on the X-rays. Thus we can add direct edges between these variables. A patient's location is hidden and acts as a confounder because a person's socio-economic and health conditions in a specific location might affect both the likelihood of getting Covid and being properly diagnosed with Pneumonia by local health care. The X-ray images are done by chest radiography imaging examination. Due to the standardization of equipment, we assume the difference in X-ray data across hospital locations is minor and can be ignored. Thus, $\text{Location} \not\to \text{XrayImages}$. We aim to learn that if a patient is randomly picked and intervened with Covid (hypothetically), how likely will they be diagnosed with pneumonia, i.e., $P(N|do(C))$? If our mentioned assumptions (including no selection bias, etc.) are correct, we expect WhatIfGAN to correctly sample from interventional distribution after training by Theorem 4.5.

To match the joint distribution $P(C, Xray, N)$, we follow the modular training order: $[\mathbb{G}_{Xray}] \to [\mathbb{G}_C, \mathbb{G}_N]$. Instead of training $\mathbb{G}_{Xray}$ from scratch, we use a pre-trained model (Giorgio Carbone, 2023) that can be utilized to produce Xray images corresponding to $C \in [0, 1]$ input. Figure 4b shows images for the original dataset and output images from the pre-trained model. Next, we train $\mathbb{G}_C$ and $\mathbb{G}_N$ together since they belong to the same c-component. Since the joint distribution contains both low and high-dimensional variables, we map Xray to a low-dimensional representation Rxray with an encoder and match $P(C, Rxray, N)$. In Figure 4c, we plot the total variation distance (TVD) of $P(C), P(N), P(N|C), P(N, C)$. We observe that TVD for all distributions is decreasing. The average treatment effect, i.e., the difference between $E[P(N|do(C = 1))]$ and $E[P(N|do(C = 0))]$ is in $[0.05, 0.08]$ after convergence. This implies that intervention with Covid increases the likelihood of being diagnosed with Pneumonia. However, these results are based on this specific COVIDx CXR-3 dataset and should not be used to make medical inferences without expert opinion.

## 6 CONCLUSION

We propose a modular adversarial training algorithm called WhatIfGAN for learning deep causal generative models for estimating causal effects with high-dimensional variables in the presence of unobserved confounders. After convergence, WhatIfGAN can generate high-dimensional samples from identifiable interventional and counterfactual distributions.

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

CONTENTS

## A    BROADER IMPACTS

Our proposed algorithm WhatIfGAN, can sample from high-dimensional observational, interventional, and counterfactual distributions. As a result, it can be used to explore different creative directions such as producing realistic interventional and counterfactual images that we can not observe in real-world. We can train WhatIfGAN models on datasets and perform intervention on sensitive attributes to detect any bias towards them or any unfairness against them (Xu et al., 2019; van Breugel et al., 2021). However, an adversary might apply our method to produce realistic images which are causal. As a result, it will be harder to detect fake data generated by DCM from real data.

## B    LIMITATIONS AND FUTURE WORK

Similar to most causal inference algorithms, we had to make the assumption of having a fully specified causal graph with latents, as prior. With the advancements in causal discovery with latents, it might be possible to reliably learn part of the structure and leverage the partial identifiability results from the literature. Indeed, this would be one of the future directions we are interested in. Another limitation of this work is that we assumed each confounder to cause only two observed variables which is considered as semi-Markovian in literature. We aim to extend our work for non-Markovian causal models where confounders can cause any number of observed variables.

# C   APPENDIX: WHATIFGAN: ADVERSARIAL TRAINING OF DEEP CAUSAL GENERATIVE MODELS

**Definition C.1** (Identifiability (Shpitser & Pearl, 2007)). Given a causal graph, $G$, let $\mathbf{M}$ be the set of all causal models that induce $G$ and objects $\phi$ and $\theta$ are computable from each model in $\mathbf{M}$. We define that $\phi$ is $\theta$-identifiable in $G$, if there exists a deterministic function $g_G$ determined by the graph structure, such that $\phi$ can be uniquely computable as $\phi = g_G(\theta)$ in any $M \in \mathbf{M}$.

**Definition C.2** (Causal Effects z-Identifiability). Let $\mathbf{X},\mathbf{Y},\mathbf{Z}$ be disjoint sets of variables in the causal graph $G$. If $\phi = P_{\mathbf{x}}(\mathbf{y})$ is the causal effect of the action $\text{do}(\mathbf{X}=\mathbf{x})$ on the variables in $\mathbf{Y}$, and $\theta$ contains $P(\mathbf{V})$ and interventional distributions $P(\mathbf{V} \setminus \mathbf{Z}'|\text{do}(\mathbf{Z}'))$, for all $\mathbf{Z}' \subseteq \mathbf{Z}$, where $\phi$ and $\theta$ satisfies the definition of Identifiability, we define it as z-identifiabililty. (Bareinboim & Pearl, 2012a) proposes a z-identification algorithm to derive $g_G$ for these $\phi$ and $\theta$

## C.1   WHATIFGAN TRAINING LOSS GRADIENTS

Let's assume we have $V_1, V_2, ..., V_n \in \mathcal{V}$, then we can update the model weights for each of these variable's generators based on the following gradients.

$$\nabla^T \times \mathbb{1} = \begin{bmatrix} \frac{\partial L_{obs}}{\partial X_1} & \frac{\partial L_{obs}}{\partial X_2} & \cdots & \frac{\partial L_{obs}}{\partial X_n} \\ \frac{\partial L_{\mathbf{x_1}}}{\partial X_1} & \frac{\partial L_{\mathbf{x_1}}}{\partial X_2} & \cdots & \frac{\partial L_{\mathbf{x_1}}}{\partial X_n} \\ \cdots & \cdots & \cdots & \cdots \\ \frac{\partial L_{\mathbf{x_n}}}{\partial X_1} & \frac{\partial L_{\mathbf{x_n}}}{\partial X_2} & \cdots & \frac{\partial L_{\mathbf{x_n}}}{\partial X_n} \end{bmatrix}^T \times \mathbb{1} = \begin{bmatrix} \nabla_{X_1} \\ \nabla_{X_2} \\ \cdots \\ \nabla_{X_n} \end{bmatrix} \quad (4)$$

Here, $\nabla_{i,j}$ represents loss gradient for the generator of variable $X_j$ for the $i$-th real observational or interventional dataset. For any column $\nabla_{*,j}$, we know that each of the corresponding observational and interventional distributions is produced from a single function, $f_j$ of a true causal model and we want our implicit generative models, $\mathbb{G}_j$ to mimic the true structural functions. Therefore, we sum all the loss gradients of $\nabla_{*,j}$ together as $\nabla_{X_j}$ and update $X_j$'s model, $\mathbb{G}_j$'s weights based on that. This will ensure that after converging, WhatIfGAN models will learn distributions of all the available datasets and according to Theorem4.2, it will be able to produce samples from causal queries which are identifiable from these given distributions.

## C.2   WHATIFGAN INTERVENTIONAL AND COUNTERFACTUAL SAMPLING AFTER TRAINING

After WhatIfGAN training, to perform hard intervention and produce samples accordingly, we manually set values of the intervened variables instead of using their neural network. Then, we feed forward those values into its children's mechanisms and generate rest of the variable like as usual. Figure 5(b) is the WhatIfGAN network for the causal graph in Figure 5(a). Now, in Figure 5(c), we performed $\text{do}(X = x)$. Exogenous variables $U$ and $n_X$ are not affecting $X$ anymore as we manually set $X = x$. For counterfactual sampling of $P_x(Y|x')$, we follow the three steps, $i)$ Abduction: With rejection sampling, we record the posterior exogenous $n_X$, confounding $U$ and the gumbel noises responsible for producing $X = x'$ in the DCM's no intervention forward propagation (Figure 5c). We collect Gumbel Noise as we performed Gumbel-softmax (Jang et al., 2017) to deal with discrete variables. $ii)$ Action: We intervene on $\mathbb{G}_X$ with $X = x$ $iii)$ Prediction: We use pre-recorded exogenous variable values as input to rest of the variable's neural network and push forward through the neural networks. (Figure 5c).

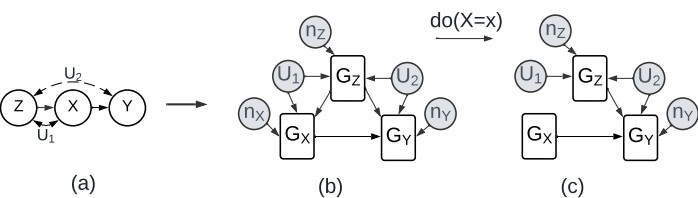

(a)                    (b)                    (c)

Figure 5: (a) Causal Graph with latents. (c), (d) DCM before and after intervention.

### C.3  WHATIFGAN: ADVERSARIAL TRAINING OF DEEP CAUSAL GENERATIVE MODELS (FULL TRAINING)

In this section, we prove that a trained DCM can sample from identifiable causal queries from any causal layer. We assume $M_1$ as true SCM and $M_2$ as DCM of WhatIfGAN.

**Theorem C.3.** *Let $\mathcal{M}_1 = (G = (\mathcal{V}, \mathcal{E}), \mathcal{N}, \mathcal{U}, \mathcal{F}, P(.))$ be an SCM. If a causal query $\mathcal{K}_{\mathcal{M}_1}(\mathcal{V})$ is identifiable from a collection of observational and/or interventional distributions $\{P_i(\mathcal{V})\}_{i \in [m]}$ for graph $G$, then any SCM $\mathcal{M}_2 = (G, \mathcal{N}', \mathcal{U}', \mathcal{F}', Q(.))$ entails the same answer to the causal query if it entails the same input distributions. Therefore, for any identifiable query $\mathcal{K}$, if $\{P_i(\mathcal{V})\}_{i \in [m]} \vdash \mathcal{K}_{\mathcal{M}_1}(\mathcal{V})$ and $P_i(\mathcal{V}) = Q_i(\mathcal{V}), \forall i \in [m]$, then $\mathcal{K}_{\mathcal{M}_1}(\mathcal{V}) = \mathcal{K}_{\mathcal{M}_2}(\mathcal{V})$.*

*Proof.* By definition of identifiability, we have that $\mathcal{K}_{\mathcal{M}_1} = g_G(\{P_i(\mathcal{V})\}_{i \in [m]})$ for some deterministic function $g_G$ that is determined by the graph structure. Since $\mathcal{M}_2$ has the same causal graph, the query $\mathcal{K}_{\mathcal{M}_2}$ is also identifiable and through the same function $g_G$, i.e., $\mathcal{K}_{\mathcal{M}_2} = g_G(\{Q_i(\mathcal{V})_{i \in [m]}\})$. Thus, the query has the same answer in both SCMs, if they entail the same input distributions over the observed variables, i.e., $P_i(\mathcal{V}) = Q_i(\mathcal{V}), \forall i$. □

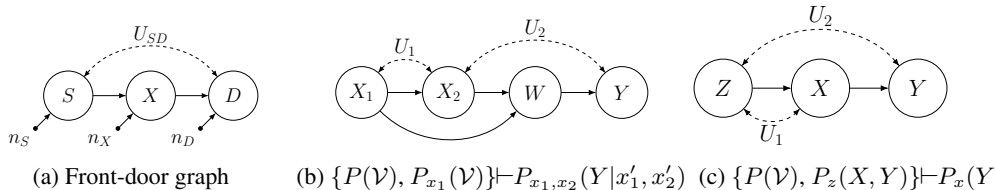

(a) Front-door graph     (b) $\{P(\mathcal{V}), P_{x_1}(\mathcal{V})\} \vdash P_{x_1,x_2}(Y|x_1', x_2')$    (c) $\{P(\mathcal{V}), P_z(X, Y)\} \vdash P_x(Y)$

Figure 6: Causal graphs with latents and respective identifiable causal queries. $\theta$ identifies $\phi$ :$\theta \vdash \phi$

**Corollary C.4.** *Let $\mathcal{M}_1 = (G = (\mathcal{V}, \mathcal{E}), \mathcal{N}, \mathcal{U}, \mathcal{F}, P(.))$ and $\mathcal{M}_2 = (G, \mathcal{N}', \mathcal{U}', \mathcal{F}', Q(.))$ be two SCMs. If $\{P(\mathcal{V})\} \vdash P_x(Y)$ for $X, Y \subset \mathcal{V}$, $X \cap Y = \emptyset$ and $P(\mathcal{V}) = Q(\mathcal{V})$ then $P_x(Y) = Q_x(Y)$*

For example, in Figure 6(b), the interventional query $P_{x_1,x_2}(W)$ is identifiable from $P(\mathcal{V})$. According to the Corollary C.4, after training on $P(\mathcal{V})$ dataset, WhatIfGAN will produce correct interventional sample from $P_{x_1,x_2}(W)$ and along with other queries in $\mathcal{L}_2(P(\mathcal{V}))$.

Bareinboim & Pearl (2012b) showed that we can identify some $\mathcal{L}_2$-queries with other surrogate interventions and $\mathcal{L}_1$-distributions. Similarly, we can apply Theorem 4.2:

**Corollary C.5.** *Let $\mathcal{M}_1 = (G = (\mathcal{V}, \mathcal{E}), \mathcal{N}, \mathcal{U}, \mathcal{F}, P(.))$ and $\mathcal{M}_2 = (G, \mathcal{N}', \mathcal{U}', \mathcal{F}', Q(.))$ be two SCMs and $X, Y$ be disjoint, and $\{S_i\}_i$ arbitrary subsets of variables. If i)$\{P(\mathcal{V}), P_{s_1}(\mathcal{V}), P_{s_2}(\mathcal{V}) \ldots\} \vdash P_x(Y)$, ii)$P(\mathcal{V}) = Q(\mathcal{V})$ and iii)$P_{s_i}(\mathcal{V}) = Q_{s_i}(\mathcal{V})$, $\forall i, s_i$ then $P_x(Y) = Q_x(Y)$.*

In Figure 6(c), the interventional query $P_x(Y)$ is identifiable from $P(\mathcal{V})$ and $P_z(X, Y)$. Therefore, after being trained on datasets sampled from these distributions, WhatIfGAN will produce correct interventional sample from $P_x(Y)$ and all other queries in $\mathcal{L}_2(P(\mathcal{V}), P_z(X, Y))$.

We apply Theorem 4.2 to answer counterfactual queries:

**Corollary C.6.** *Let $\mathcal{M}_1 = (G = (\mathcal{V}, \mathcal{E}), \mathcal{N}, \mathcal{U}, \mathcal{F}, P(.))$ and $\mathcal{M}_2 = (G, \mathcal{N}', \mathcal{U}', \mathcal{F}', Q(.))$ be two SCMs and $X, Y$ be disjoint, and $\{S_i\}_i$ arbitrary subsets of variables. If i)$\{P(\mathcal{V}), P_{s_1}(\mathcal{V}), P_{s_2}(\mathcal{V}), \ldots\} \vdash P(Y_x|e)$, ii)$P(\mathcal{V}) = Q(\mathcal{V})$ and iii)$P_{s_i}(\mathcal{V}) = Q_{s_i}(\mathcal{V})$, $\forall i, s_i$ then $P(Y_x|e) = Q(Y_x|e)$.*

In Figure 6(b), the counterfactual query $P_{x_1,x_2}(Y|x_1', x_2')$ is identifiable from $P(\mathcal{V})$ and $P_{x_1}(\mathcal{V})$ (see Appendix F.4.4). After training on datasets sampled from $P_{x_1}(\mathcal{V})$ and $P(\mathcal{V})$, WhatIfGAN will produce correct counterfactual samples from $P_{x_1,x_2}(Y|x_1', x_2')$ and queries in $\mathcal{L}_3(P(\mathcal{V}), P_{x_1}(\mathcal{V}))$.

### C.4  TRAINING WITH MULTIPLE DATASETS

We propose a method in Algorithm 3 for training WhatIfGAN with both $\mathcal{L}_1$ and $\mathcal{L}_2$ datasets. We use Wasserstein GAN with penalized gradients (WGAN-GP) (Gulrajani et al., 2017) for adversarial

---

**Algorithm 3** WhatIfGAN Training on Multiple Datasets

---

1: **Input:** Causal Graph $G = (\mathcal{V}, \mathcal{E})$, Interventional datasets= $(\mathbf{I}, \mathcal{D})$, DCM $\mathbb{G}$, Critic $\mathbb{D}$, Parameters= $\theta_1, \ldots, \theta_n, \lambda = 10$
2: **while** $\theta_1, \ldots, \theta_n$ has not converged **do**
3:     **for each** $(X, D) \in (\mathbf{I}, \mathcal{D})$ **do**
4:         $compare\_var = \mathcal{V}$
5:         Sample real data $\mathbf{v_x^r} \sim D$ following the distribution $\mathbb{P}_x^r$ with intervention $X$.
6:         $\mathbf{x} \leftarrow X.values$    $//X=(keys, values)$
7:         $\mathbf{v}_x^f = \text{RunGAN}(G, \mathbb{G}, X, compare\_var, \emptyset)$
8:         $\hat{\mathbf{v}}_x = \epsilon \mathbf{v}_x^r + (1 - \epsilon)\mathbf{v}_x^f$
9:         $L_x = \mathbb{D}_{w_x}(\mathbf{v}_x^f) - \mathbb{D}_{w_x}(\mathbf{v}_x^r) * \lambda(\|\nabla_{\hat{\mathbf{v}}_x}\mathbb{D}_{w_x}(\hat{\mathbf{v}}_x)\|_2 - 1)^2$
10:        $G_{loss} = G_{loss} + \mathbb{D}_{w_x}(\mathbf{v}_x^f)$
11:        $w_x = Adam(\nabla_{w_x}\frac{1}{m}\sum_{j=1}^m L_{\mathbb{D}_x}, w_x, \alpha, \beta_1, \beta_2)$
12:     **for** $\theta \in \theta_1, \ldots \theta_n$ **do**
13:         $\theta = Adam(\nabla_\theta - G_{loss}, \theta, \alpha, \beta_1, \beta_2)$
14: **Return:** $\theta_1, \ldots \theta_n$

---

training. $\mathbb{G}$ is the DCM, a set of generators and $\{\mathbb{D}_x\}_{X \in \mathbf{I}}$ are a set of discriminators for each intervention value combinations. The objective function of a two-player minimax game would be

$$\min_{\mathbb{G}} \sum_x \max_{\mathbb{D}_x} L(\mathbb{D}_x, \mathbb{G}),$$

$$L(\mathbb{D}_x, \mathbb{G}) = \mathop{\mathbb{E}}_{v \sim \mathbb{P}_x^r}[\mathbb{D}_x(\mathbf{v})] - \mathop{\mathbb{E}}_{\mathbf{z} \sim \mathbb{P}_Z, \mathbf{u} \sim \mathbb{P}_U}[\mathbb{D}_x(\mathbb{G}^{(\mathbf{x})}(\mathbf{z}, \mathbf{u}))]$$

Here, for intervention $do(X = x), X \in \mathbf{I}, \mathbb{G}^{(x)}(\mathbf{z}, \mathbf{u})$ are generated samples and $v \sim \mathbb{P}_x^r$ are real $\mathcal{L}_1$ or $\mathcal{L}_2$ samples. We train our models by iterating over all datasets and learn $\mathcal{L}_1$ and $\mathcal{L}_2$ distributions (line 3). We produce generated interventional samples by intervening on the corresponding node of our architecture. For this purpose, we call Algorithm 9 RunGAN(), at line 7. We compare the generated samples with the input $\mathcal{L}_1$ or $\mathcal{L}_2$ datasets. For each different combination of the intervened variables $x$, $\mathbb{D}_x$ will have different losses, $L_{X=x}$ from each discriminator (line 9). At line 10, we calculate and accumulate the generator loss over each dataset. If we have $V_1, \ldots, V_n \in \mathcal{V}$, then we update each variable's model weights based on the accumulated loss (line 13). This will ensure that after convergence, WhatIfGAN models will learn distributions of all the available datasets and according to Theorem 4.2, it will be able to produce samples from same or higher causal layers queries that are identifiable from these input distributions. Following this approach, WhatIfGAN Training in Algorithm 3, will find a DCM solution that matches to all the input distributions, mimicking the true SCM. Finally, we describe sampling method for WhatIfGAN after training convergence in Appendix C.2.

**Proposition C.7.** *Let $\mathcal{M}_1$ be the true SCM and Algorithm 3:* **WhatIfGAN Training** *converges after being trained on datasets:* $\mathbf{D} = \{\mathcal{D}_i\}_i$, *outputs the DCM $\mathcal{M}_2$. If for any causal query $\mathcal{K}_{\mathcal{M}_1}(\mathcal{V})$ identifiable from $\mathbf{D}$ then $\mathcal{K}_{\mathcal{M}_1}(\mathcal{V}) = \mathcal{K}_{\mathcal{M}2}(\mathcal{V})$*

*Proof.* Let $\mathcal{M}_1 = (G = (\mathcal{V}, \mathcal{E}), \mathcal{N}, \mathcal{U}, \mathcal{F}, P(.))$ be the true SCM and $\mathcal{M}_2 = (G, \mathcal{N}', \mathcal{U}', \mathcal{F}', Q(.))$ be the deep causal generative model represented by WhatIfGAN. WhatIfGAN Training converges implies that $Q_i(\mathcal{V}) = P_i(\mathcal{V}), \forall i \in [m]$ for all input distributions. Therefore, according to Theorem 4.2, WhatIfGAN is capable of producing samples from correct interventional or counterfactual distributions that are identifiable from the input distributions. $\square$

## C.5   Non-Markovianity

Note that, one can convert a non-Markovian causal model $M_1$ to a semi-Markovian causal model $M_2$ by taking the common confounder among the observed variables and splitting it into new confounders for each pair. Now, for a causal query to be unidentifiable in a semi-Markovian model $M_2$, we can apply the Identification algorithm (Shpitser & Pearl, 2008) and check if there exists a hedge. The unidentifiability of the causal query does not depend on the confounder distribution. Thus, if the causal query is unidentifiable in the transformed semi-Markovian model $M_2$, it will be unidentifiable in the original non-Markovian model $M_1$ as well.

Besides Semi-Markovian, Theorem 4.2 holds for Non-Markovian models, with latents appearing anywhere in the graph and thus can be learned by WhatIfGAN training. Jaber et al. (2019) performs

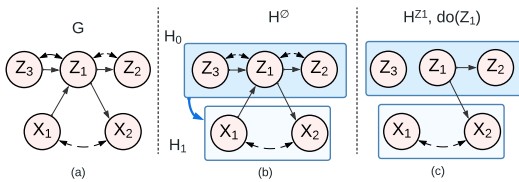

Figure 7: (a) Causal graph $G$, (b) $\mathcal{H}^{\emptyset}$-graph, (c) $H^{Z_1}$-graph

causal effect identification on equivalence class of causal diagrams, a partial ancestral graph (PAG) that can be learned from observational data. Therefore, we can apply their method to check if an interventional query is identifiable from observational data in a Non-Markovian causal model and express the query in terms of observations and obtain the same result as Theorem C.3.

# D APPENDIX: WHATIFGAN MODULAR TRAINING

## D.1 TIAN'S FACTORIZATION FOR MODULAR TRAINING

In Figure 7(a), We apply Tian's factorization (Tian & Pearl, 2002) to get,

$$P(v) = P(x_1, x_2 | \text{do}(z_1)) P(z_1, z_2, z_3 | \text{do}(x_1)) \tag{5}$$

We need to match the following distributions with the DCM.

$$
\begin{aligned}
P(x_1, x_2 | \text{do}(z_1)) &= Q(x_1, x_2 | \text{do}(z_1)) \\
P(z_1, z_2, z_3 | \text{do}(x_1)) &= Q(z_1, z_2, z_3 | \text{do}(x_1))
\end{aligned}
\tag{6}
$$

With modular training, we matched the following alternative distributions:

$$
\begin{aligned}
P(z_1, z_2, z_3 | x_1) &= Q(z_1, z_2, z_3 | \text{do}(x_1)) \\
P(x_1, x_2, z_1, z_3) &= Q(x_1, x_2, z_1, z_3)
\end{aligned}
\tag{7}
$$

Now, for the graph in Figure 7(a),

$$
\begin{aligned}
P(x_1, x_2, z_1, z_2, z_3) &= P(x_1, x_2 | \text{do}(z_1)) \times P(z_1, z_2, z_3 | \text{do}(x_1)) \\
&= \frac{P(x_1, x_2, z_1, z_3)}{P(z_1, z_3 | \text{do}(x_1))} \times P(z_1, z_2, z_3 | x_1) \quad \text{[C-factorization of } P(x_1, x_2, z_1, z_3)] \\
&= \frac{P(x_1, x_2, z_1, z_3)}{P(z_1, z_3 | x_1)} \times P(z_1, z_2, z_3 | x_1) \quad \text{[Do-calculus rule-2 applies]} \\
&= \frac{P(x_1, x_2, z_1, z_3)}{\sum_{z_2} P(z_1, z_2, z_3 | x_1)} \times P(z_1, z_2, z_3 | x_1) \\
&= \frac{Q(x_1, x_2, z_1, z_3)}{\sum_{z_2} Q_{x_1}(z_1, z_2, z_3)} \times Q_{x_1}(z_1, z_2, z_3) \quad \text{[According to Equation 7]} \\
&= Q(x_1, x_2, z_1, z_2) \quad \text{[We can follow the same above steps as } P(.) \text{ for } Q(.)]
\end{aligned}
\tag{8}
$$

Therefore, if we match the distributions in Equation 7 with the DCM, it will match $P(\mathcal{V})$ as well.

## D.2 MODULAR TRAINING FOR INTERVENTIONAL DATASET

### D.2.1 MODULAR TRAINING BASICS

Suppose, for the graph in Figure 8, we have two datasets $D^{\tilde{p}} \sim P(\mathcal{V})$ and $D^{z_1} \sim P_{Z_1}(V)$, i.e., intervention set $\mathcal{I} = \{\emptyset, Z_1\}$. Joint distributions in both dataset factorize like below:

$$
\begin{aligned}
P(v) &= P_{z_1}(x_1, x_2) P_{x_1}(z_1, z_2, z_3) \\
P_{z_1}(v) &= P_{z_1}(x_1, x_2) P(z_3) P_{z_1}(z_2) \\
&= P_{z_1}(x_1, x_2) P_{z_1}(z_2, z_3) [\text{Since } Z_2, Z_3 \text{ independent in } G_{\overline{Z_1}} \text{ graph}] \\
&= P_{z_1}(x_1, x_2) P_{x_1, z_1}(z_2, z_3) [\text{Ignores intervention using do calculus rule-2}]
\end{aligned}
\tag{9}
$$

We change the c-factors for $P_{z_1}(V)$ to keep the variables in each c-factor same in all distributions. This factorization suggests that to match $P(\mathcal{V})$ and $P_{Z_1}(V)$ we have to match each of the c-factors using $D^{\tilde{p}}$ and $D^{z_1}$ datasets. In Figure 2 graph $G$, $P_{x_1}(z_1, z_2, z_3) = P(z_1, z_2, z_3|x_1)$ since do-calculus rule-2 applies. And in $G_{\overline{Z_1}}$, $P(z_3)P_{z_1}(z_2)$ can be combined into $P_{x_1,z_1}(z_2, z_3)$. Thus we can use these distributions to train part of the DCM: $\mathbb{G}_{Z_1}, \mathbb{G}_{Z_2}, \mathbb{G}_{Z_3}$ to learn both $Q(z_1, z_2, z_3|\mathrm{do}(x_1))$ and $Q(z_2, z_3|\mathrm{do}(x_1, z_1))$. However, $P(x_1, x_2|\mathrm{do}(z_1)) \neq P(x_1, x_2|z_1)$ in $P(\mathcal{V})$. But we have access to $P_{z_1}(\mathcal{V})$. Thus, we can train $\mathbb{G}_{X_1}, \mathbb{G}_{X_2}$ with only dataset $D^{Z_1} \sim P_{z_1}(\mathcal{V})$ (instead of both $D^{\tilde{p}}, D^{Z_1}$) and learn $Q(x_1, x_2|\mathrm{do}(z_1))$. This will ensure the DCM has matched both $P(\mathcal{V})$ and $P_{z_1}(\mathcal{V})$ distribution.

Thus, we search proxy distributions to each c-factor corresponding to both $P(\mathcal{V})$ and $P_{Z_1}(V)$ dataset, to train the mechanisms in a c-component $\mathbf{Y}$. For each of the c-factors corresponding to $\mathbf{Y}$ in $P(\mathcal{V})$ and $P_{z_1}(V)$, we search for two ancestor sets $\mathcal{A}_{\tilde{p}}, \mathcal{A}_{Z_1}$ in both $P(\mathcal{V})$ and $P_{z_1}(V)$ datasets such that the parent set $Pa(\mathbf{Y} \cup \mathcal{A}_{\tilde{p}})$ satisfies rule-2 for the joint $\mathbf{Y} \cup \mathcal{A}_{\tilde{p}}$ and $Pa(\mathbf{Y} \cup \mathcal{A}_{Z_1})$ satisfies rule-2 for the joint $\mathbf{Y} \cup \mathcal{A}_{Z_1}$ with $\mathrm{do}(Z_1)$ intervention.

We update Definition 4.4as **modularity condition-I** for multiple interventional datasets as below:

**Definition D.1** (Modularity condition-I). Given a causal graph $G$, an intervention $I \in \mathcal{I}$ and a c-component variable set $\mathbf{Y}$, a set $\mathcal{A} \subseteq An_{G_{\overline{I}}}(\mathbf{Y}) \setminus \mathbf{Y}$ is said to satisfy the modularity condition if it is the smallest set that satisfies $P(\mathbf{Y} \cup \mathbf{X}|\mathrm{do}(Pa(\mathbf{Y} \cup \mathbf{X})), \mathrm{do}(I)) = P(\mathbf{Y} \cup \mathbf{X}|Pa(\mathbf{Y} \cup \mathbf{X}), \mathrm{do}(I))$, i.e., do-calculus rule-2 (Pearl, 1995) applies.

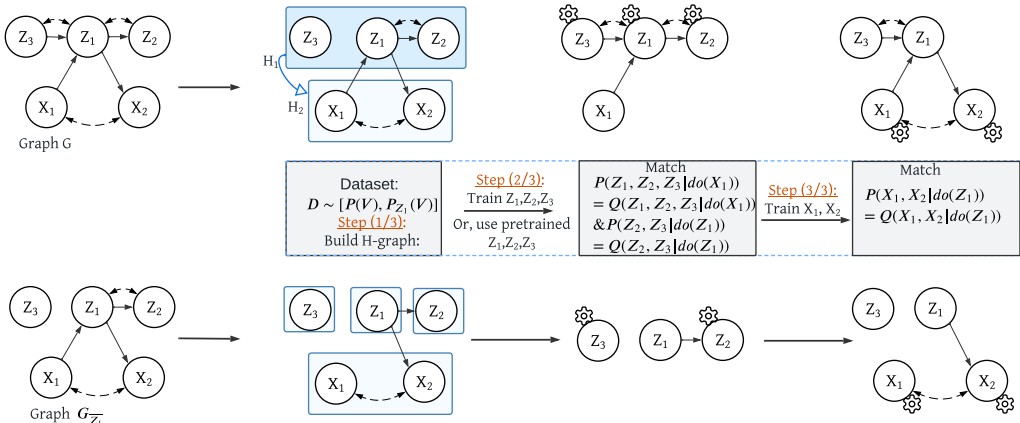

Figure 8: Modular training on H-graph: $H_1 : [Z_1, Z_2, Z_3] \rightarrow H_2 : [X_1, X_2]$ with dataset $D \sim P(V)$.

Unlike before, we have access to $P_{z_1}(\mathcal{V})$ and we can use that to match $P_{z_1}(x_1, x_2)$ in both cases. To match the $\mathcal{L}_1$ and $\mathcal{L}_2$ joint distributions according to (9), we train each c-component one by one. For each c-component, we identify the modularity conditions of all c-factors $P_{pa(\mathbf{Y})\cup I}(\mathbf{Y})$, $\forall I \in \mathcal{I}$ and use them to train $\mathbf{Y}$. We train the mechanisms in $\mathbf{Y}$ to learn an alternative to each c-factor $P_{pa(\mathbf{Y})\cup I}(\mathbf{Y})$, $\forall I \in \mathcal{I}$. For some ancestor set $\mathcal{A}_I$, the alternative distribution is in the form $P(\mathbf{Y} \cup \mathcal{A}_I|\mathrm{do}(Pa(\mathbf{Y} \cup \mathcal{A}_I)), \mathrm{do}(I))$ which should be equivalent to $P(\mathbf{Y} \cup \mathcal{A}|Pa(\mathbf{Y} \cup \mathcal{A}), \mathrm{do}(I))$. We will find an $\mathcal{A}_I$ from the $D^I, \forall I \in \mathcal{I}$ such that we do not require $Pa(\mathbf{Y} \cup \mathcal{A})$ to be intervened on.

Now, to match $P(\mathbf{Y} \cup \mathcal{A}|Pa(\mathbf{Y} \cup \mathcal{A}), \mathrm{do}(I)) = Q(\mathbf{Y} \cup \mathcal{A}|\mathrm{do}(Pa(\mathbf{Y} \cup \mathcal{A})), \mathrm{do}(I))$ with our generative models, we pick the observations of $Pa(\mathbf{Y} \cup \mathcal{A})$ from $D^I$ dataset and intervene in our DCM with those values besides intervening on $\mathbb{G}_I$. Since we do not need generated samples for $Pa(\mathbf{Y} \cup \mathcal{A})$ from DCM, rather their observations from the given $D^I$ dataset, we do not require them to be trained beforehand. However, the order in which we train c-components matters and we follow the partial order found for $P(\mathcal{V})$ dataset even thought we train with multiple datasets.

For example, in Figure 2, we have two graphs $G$ and $G_{\overline{Z_1}}$. We follow $G$'s training order for both graphs to train the c-components, i.e., $[\mathbb{G}_{Z_1}, \mathbb{G}_{Z_2}, \mathbb{G}_{Z_3}] \rightarrow [\mathbb{G}_{X_1}, \mathbb{G}_{X_2}]$. Here, for the c-component $\mathbf{Y} = \{Z_1, Z_2, Z_3\}$, we match $P(\mathcal{V})$ c-factor $P_{x_1}(z_1, z_2, z_3)$ and $P_{z_1}(\mathcal{V})$ c-factor $P_{x_1, z_1}(z_2, z_3)$ thus have to find alternative distribution for them. We find the smallest ancestor set $\mathcal{A}_{\tilde{p}}, \mathcal{A}_{Z_1}$ for these

---

**Algorithm 4** TrainModule($\mathbb{G}, G, H_*, \mathcal{A}, \mathbf{D}$)

1: **Input:** DCM $\mathbb{G}$, Graph $G(\mathcal{V}, \mathcal{E})$, h-node $H_*$, Ancestor set $\mathcal{A}$, Data $\mathbf{D}$, Params $\theta_H, \lambda = 10$
2: **while** $\theta_{H_*}$ has not converged **do**
3:    **for each** $(\mathcal{A}_i, X_i, D_i) \in (\mathcal{A}, \mathbf{D})$ **do**
4:       $V_r = H_* \cup \mathcal{A}_i \cup Pa(H_* \cup \mathcal{A}_i) \cup X_i$
5:       Initialize critic $\mathbb{D}_{w_i}$
6:       **for** $t = 1, \ldots, m$ {$m$ samples} **do**
7:          Sample real data $\mathbf{v_x^r} \sim D_i$
8:          $\mathbf{x}^r \leftarrow$ get_intv_values$(X_i, D_i)$
9:          $\mathbf{v}_x^f = $ RunGAN$(\mathbb{G}, \mathbf{x}^r, V_r, \theta_{H_*})$
10:          $\hat{\mathbf{v}}_x = \epsilon \mathbf{v}_x^r + (1 - \epsilon)\mathbf{v}_x^f$
11:          $L_i^{(t)} = \mathbb{D}_{w_i}(\mathbf{v}_x^f) - \mathbb{D}_{w_i}(\mathbf{v}_x^r) * \lambda(\|\nabla_{\hat{\mathbf{v}}_x} \mathbb{D}_{w_i}(\hat{\mathbf{v}}_x)\|_2 - 1)^2$
12:       $w_i = Adam(\nabla_{w_i} \frac{1}{m} \sum_{t=1}^m L_i^{(t)}, w_i)$
13:       $G_{loss} = G_{loss} + \frac{1}{m} \sum_{j=1}^m -\mathbb{D}_{w_i}(\mathbf{v}_x^f)$
14:    **for** $\theta \in \theta_{H_*}$ {All hnode mechanisms} **do**
15:       $\theta = Adam(\nabla_\theta G_{loss}, \theta)$
16: **Return:** $\theta_1, \ldots \theta_n$

---

**Algorithm 5** IsRule2($Y, X, I = \emptyset$ (by default))

1: **Input:** Variable sets $Y$ and $X$, Intervention $I$.
2: **Return:**
3: **if** $P(Y \cup X|\text{do}(Pa(Y \cup X)), \text{do}(I)) = P(Y \cup X|Pa(Y \cup X), \text{do}(I))$ **then**
4:    **Return:1**
5: **else**
6:    **Return:0**

---

c-factors in both $D^{\tilde{p}}$ and $D^{Z_1}$ datasets. $\mathcal{A}_\emptyset = \emptyset$ satisfies modularity condition for $P(\mathcal{V})$ c-factor and their $Pa(\mathbf{Y} \cup \mathcal{A}) = \{X_1\}$. $\mathcal{A}_{Z_1} = \emptyset$ satisfies modularity condition for $P_{z_1}(\mathcal{V})$ c-factor and their $Pa(\mathbf{Y} \cup \mathcal{A}) = \emptyset$. At step (2/3) in Figure 2, We do not need $\mathbb{G}_{X_1}$ to be pre-trained. $[\mathbb{G}_{Z_1}, \mathbb{G}_{Z_2}, \mathbb{G}_{Z_3}]$ converges by matching both $P(z_1, z_2, z_3|x_1) = Q_{x_1}(z_1, z_2, z_3)$ and $P_{x_1,z_1}(z_2, z_3) = Q_{x_1,z_1}(z_2, z_3)$.

Now, we train mechanisms of the next c-component $[\mathbb{G}_{X_1}, \mathbb{G}_{X_2}]$ in our training order (step 3/3). We have to match $P(\mathcal{V})$ c-factor $P_{z_1}(x_1, x_2)$ and $P_{Z_1}(\mathcal{V})$ c-factor $P_{z_1}(x_1, x_2)$. Ancestor set $\mathcal{A}_{\tilde{p}} = \{Z_1, Z_3\}$ satisfies the modularity condition for $\mathbf{Y} = \{X_1, X_2\}$ with $P(\mathcal{V})$ dataset but $\mathcal{A}_{Z_1} = \emptyset$ a smaller set, satisfies the modularity condition for same c-factor with $P_{z_1}(\mathcal{V})$ dataset. Also, $P_{z_1}(\mathcal{V})$ c-factor is $P_{z_1}(x_1, x_2)$. Thus if we train $[\mathbb{G}_{X_1}, \mathbb{G}_{X_2}]$ with only $P_{Z_1}(\mathcal{V})$ dataset, it will learn both c-factors and converge with $P_{z_1}(x_1, x_2) = Q_{z_1}(x_1, x_2)$. Since we have matched all the c-factors, our DCM will match both $P(\mathcal{V})$ and $P_{Z_1}(\mathcal{V})$ distributions. During training of $[\mathbb{G}_{X_1}, \mathbb{G}_{X_2}]$, we had $\mathcal{A} = \emptyset$ for both observation and interventional c-factors. Therefore, we do not need any pre-trained mechanisms, rather we can directly use the observations from $P_{Z_1}(\mathcal{V})$ dataset as parent values. We define $\mathcal{H}^I$-graph for each $I \in \mathcal{I}$ as below:

**Definition D.2** ($\mathcal{H}^I$-graph). For a post-interventional graph $G_{\overline{I}}$, let the set of c-components in $G_{\overline{I}}$ be $\mathcal{C} = \{C_1, \ldots C_t\}$. Choose a partition $\{H_k^I\}_k$ of $\mathcal{C}$ such that the $\mathcal{H}^I$-graph $\mathcal{H}^I = (V_{\mathcal{H}^I}, E_{\mathcal{H}^I})$, defined as follows, is acyclic: $V_{\mathcal{H}^I} = \{H_k^I\}_k$ and for any $s, t$, $H_s^I \rightarrow H_t^I \in E_{\mathcal{H}^I}$, iff $P(H_t^I|\text{do}(pa(H_t^I) \cap H_s^I)) \neq P(H_t^I|pa(H_t^I) \cap H_s^I)$, i.e., do-calculus rule-2 does not hold. Note that one can always choose a partition of $\mathcal{C}$ to ensure $\mathcal{H}^I$ is acyclic: The $\mathcal{H}^I$ graph with a single node $H_1^I = \mathcal{C}$ in $G_{\overline{I}}$. Even though $\mathcal{H}^I$ for different $I$ might have different partial order, during training, every $\mathcal{H}^I$ follows the partial order of $\mathcal{H}^\emptyset$. Since its partial order is valid for other H-graphs as well (Proposition D.14).

Here, $\mathcal{H}^I$ is the $\mathcal{H}$-graph constructed from $G_{\overline{I}}$, for $I \in \mathcal{I}$ where $\mathcal{I}$ is the intervention set. $\mathcal{H}^\emptyset$ is the $\mathcal{H}$-graph constructed from $G$ for observational training. $H_k^I := k$-th h-node in the $\mathcal{H}^I$ graph. During $H^I$-graphs construction, we resolve cycles by combining c-components on that cycle into a single h-node. Please check example in Figure 15. After merging all such cycles, $H^I, \forall I \in \mathcal{I}$ will become directed acyclic graphs. The partial order of this graph will indicate the training order that we can follow to train all variables in $G$. For example in Figure 8, two given datasets $D_1$ and $D_2$, imply two different graphs $G$ and $G_{Z_1}$ respectively. $[Z_3] \rightarrow [Z_1, Z_2] \rightarrow [X_1, X_2]$ is a valid training order for $\mathcal{H}_{Z_1}$, we follow the same order as $\mathcal{H}^\emptyset$. We follow : $[Z_1, Z_2, Z_3] \rightarrow [X_1, X_2]$.

---

**Algorithm 6** Construct-$\mathcal{H}^I$-graph$(G, \mathcal{I})$

---

1: **Input:** Causal Graph $G$, Intervention set, $\mathcal{I}$
2: **for each** $I \in \mathcal{I}$ **do**
3:     $\mathcal{C} \leftarrow$ get_ccomponents$(G_{\bar{I}})$
4:     Construct graph $\mathcal{H}^I$ by creating nodes $H_j^I$ as $H_j^I = C_j, \forall C_j \in \mathcal{C}$
5: **for each** $I \in \mathcal{I}$ **do**
6:     **for each** $H_s^I, H_t^I \in \mathcal{H}^I$ such that $H_s^I \neq H_t^I$ **do**
7:       **if** $P(H_t^I|\text{do}(pa(H_t^I) \cap H_s^I)) \neq P(H_t^I|pa(H_t^I) \cap H_s^I)$ **then**
8:         $\mathcal{H}^I.add(H_s^I \rightarrow H_t^I)$
9:     $\mathcal{H}^I \leftarrow$ merge$(\mathcal{H}^I, cyc) \, \forall cyc \in Cycles(H^I)$
10: **for each** $I \in \mathcal{I}$ **do**
11:     **for each** $H_j^{\emptyset} \in \mathcal{H}^{\emptyset}$ **do**
12:       $H_j^I = \bigcup_k H_k^I$ such that $\mathcal{V}(H_k^I) \subseteq \mathcal{V}(H_j^{\emptyset})$      [All variables in $H_k^I$ h-node is contained in $H_j^{\emptyset}$ h-node.]
13: **Return:** $\{\mathcal{H}^I : I \in \mathcal{I}\}$

---

**Algorithm 7** Modular Training-I$(G, \mathcal{I}, \mathbf{D})$

---

1: **Input:** Causal Graph $G$, Intervention set $\mathcal{I}$, Dataset $\mathbf{D}$.
2: Initialize DCM $\mathbb{G}$
3: $\mathcal{H}^I \leftarrow$ **Construct-H-graphs**$(G, \mathcal{I})$
4: **for each** $H_k \in \mathcal{H}^{\emptyset}$ in partial order   **do**
5:     $\mathcal{A}_{\tilde{p}} \leftarrow \mathcal{V}$      //Initialize with all nodes
6:     **for each** $S \subseteq An_G(H_k)$ **do**
7:       **if** IsRule2$(H_k, S, \tilde{p}) = 1$
      and $|S| < |\mathcal{A}_{\tilde{p}}|$ **then**
8:         $\mathcal{A}_{\tilde{p}} \leftarrow S$
9:     **for each** $I \in \mathcal{I} \cap H_k$ **do**
10:       $\mathcal{A}_I \leftarrow \mathcal{V}$      //Initialize with all nodes
11:       **for each** $S \subseteq An_{G_{\bar{I}}}(H_k)$ **do**
12:         **if** IsRule2$(H_k, S, I) = 1$
        and $|S| < |\mathcal{A}_I|$ **then**
13:          $\mathcal{A}_I \leftarrow S$
14:     $\mathbb{G}_{H_k} \leftarrow$ TrainModule$(\mathbb{G}_{H_k}, G, H_k, \mathcal{A}, \mathbf{D})$
15: **Return:** $\mathbb{G}$

---

We run the subroutine **Contruct-$\mathcal{H}^I$-graph()** in Algorithm 6 to build $\mathcal{H}$-graphs. We check the edge condition at line 5 and merge cycles at line 7 if any. In Figure 2 step (1/3), we build the $\mathcal{H}$-graph $H_1 : [Z_1, Z_2, Z_3] \rightarrow H_2 : [X_1, X_2]$ for $G$.

### D.2.2   Training process of modular WhatIfGAN

We construct the $\mathcal{H}^I$-graph for each $I \in \mathcal{I}$ at Algorithm 7 line 3. Next, we train each h-node $H_k^{\emptyset}$ of $\mathcal{H}^{\emptyset}$, according to its partial order $\mathcal{T}$. Since we follow the partial order of $\mathcal{H}^{\emptyset}$-graph, we remove the superscript to address the hnode. Next, we match alternative distributions for $P_I(\mathcal{V})$ c-factors that correspond to the c-components in $H_k$. (lines 4-14) We initialize a set $\mathcal{A}_I = \{V : V \subseteq An_{G_{\bar{I}}}(H_k)\}, \forall I \in \mathcal{I}$ to keep track of the joint distribution we need to match to train each h-node $H_k$ from $D^I$ datasets. We iterate over each intervention and search for the smallest set of ancestors $\mathcal{A}_I$ in $G_{\bar{I}}$ such that $\mathcal{A}_I$ satisfies the modularity condition for $H_k$ in $D^I$ dataset tested by Algorithm 5: IsRule2(.) (line 7)

$\mathcal{A}_I, \forall I \in \mathcal{I}$ implies a set of joint distributions in Equation 10, which is sufficient for training the current h-node $H_k$ to learn the c-factors $P_{Pa(C_i) \cup I}(C_i), \forall C_i \in H_k, \forall I \in \mathcal{I}$.

$$Q(H_k \cup \mathcal{A}_I|\text{do}(pa(H_k \cup \mathcal{A}_I)), \text{do}(I)) = P(H_k \cup \mathcal{A}_I|pa(H_k \cup \mathcal{A}_I), \text{do}(I)), \text{in } G_{\bar{I}}, \forall I \in \mathcal{I}.$$

$$\text{Training: } H_k, \text{Pre-trained: } \mathcal{A}_I, \text{From } D^I \text{ dataset: } pa(H_k \cup \mathcal{A}_I), \text{Intervened: } \text{do}(I) \tag{10}$$

Training $H_k$ with the $\mathcal{A}_{\tilde{p}}$ found at this step, is sufficient to learn the c-factors $P_{Pa(C_i)}(C_i), \forall C_i \in H_k$. Similarly, if we have an interventional dataset with $I \in H_k$ i.e., the intervened variable lies in the current h-node, we have to match c-factors $P_{Pa(C_i) \cup I}(C_i), \forall C_i \in H_k$. To find alternatives to these

c-factors, we look for the ancestor set $\mathcal{A}_I$ in the $D^I$ dataset. For each $\mathcal{A}_I$, we train $H_k$ to match the interventional joint distribution in Equation 10. We ignore intervention on any descendants of $H_k$ since the intervention will not affect c-factors differently than the c-factor in the $D^{\tilde{p}}$ observational dataset.

### D.2.3  Learn $P(\mathcal{V})$-factors from interventional datasets

When we need the alternative distribution for $P(\mathcal{V})$ c-factor, we search for the smallest ancestor set in $D^\emptyset$ dataset. However, when we have a dataset $D^I$ with $I \in An_G(H_j^\emptyset)$, we can search for an ancestor set in $An_{G_{\overline{I}}}(H_k)$ and train on $D^I$ to match a distribution that would be a proxy to $P(\mathcal{V})$ c-factor. This is possible because when we factorize $P_I(\mathcal{V})$ for $I \in An_G(H_j^\emptyset)$, the c-factors corresponding to the descendant c-components of $I$ are similar to $P(\mathcal{V})$ c-factors of the same c-components.

We update Theorem 4.5 for interventional datasets as below.

**Theorem D.3.** *Let $\mathcal{M}_1 = (G = (\mathcal{V}, \mathcal{E}), \mathcal{N}, \mathcal{U}, \mathcal{F}, P(.))$ be the true SCM and $\mathcal{M}_2 = (G, \mathcal{N}', \mathcal{U}', \mathcal{F}', Q(.))$ be the DCM. Suppose Algorithm 7:* **WhatIfGAN Modular Training-I** *on observational and interventional datasets $\mathbf{D}^I \sim P_I(\mathcal{V}), \forall I \in \mathcal{I}$ converges for each h-node in the $\mathcal{H}^\emptyset$-graph constructed from $G = (\mathcal{V}, \mathcal{E})$ and DCM induces the distribution $Q_I(V), \forall I \in \mathcal{I}$. Then, we have i)$P_I(V) = Q_I(V)$, and ii) for any interventional or counterfactual causal query $\mathcal{K}_{\mathcal{M}_1}(\mathcal{V})$ that is identifiable from $\mathbf{D}^I, \forall I \in \mathcal{I}$, we have $\mathcal{K}_{\mathcal{M}_1}(\mathcal{V}) = \mathcal{K}_{\mathcal{M}2}(\mathcal{V})$.*

We provide the proof in Appendix D.6.

### D.3  Training

WhatIfGAN utilizes conditional Wasserstein GAN with penalized gradients (Gulrajani et al., 2017) for adversarial training on $\mathcal{L}_1$ and $\mathcal{L}_2$ datasets in Algorithm 4. $\mathbb{G}$ is the DCM, a set of generators and $\{\mathbb{D}_{w_i}\}$ are a set of critics for each intervention dataset. Here, for intervention $do(X = x), X \in \mathbf{I}$, $\mathbb{G}^{(x)}(\mathbf{z}, \mathbf{u})$ are generated samples and $v \sim \mathbb{P}_x^r$ are real $\mathcal{L}_1$ or $\mathcal{L}_2$ samples. We train our models by iterating over all datasets and learn $\mathcal{L}_1$ and $\mathcal{L}_2$ distributions (lines 3-15). We produce fake interventional samples at line 9 by intervening on the corresponding node of our architecture with Algorithm 9 RunGAN(). Each critic $\mathbb{D}_x$ will obtain different losses, $L_{X=x}$ by comparing the generated samples with different true datasets (line 11). Finally, at line 15, we update each variable's model weights located at the current hnode based on the accumulated generated loss over each dataset at line 13. After calling Algorithm 4: **TrainModule()** for each of the h-nodes according to the partial order of $\mathcal{H}^\emptyset$-graph, WhatIfGAN will find a DCM equivalent to the true SCM that matches all dataset distributions. According to, theorem 4.2, it will be able to produce correct $\mathcal{L}_1, \mathcal{L}_2$ and $\mathcal{L}_3$ samples identifiable from these input distributions (Appendix C.2).

### D.4  Identifiable distributions from WhatIfGAN Modular Training

In this section, we show that WhatIfGAN modular training will match the observational joint distribution $P(\mathcal{V})$ (Proposition D.13) and the interventional joint distribution $P_I(\mathcal{V}), \forall I \in \mathcal{I}$ (Proposition D.15). We start with some definitions that would be required during our proofs.

**Definition D.4** (Intervention Set, $\mathcal{I}$)**.** Intervention Set, $\mathcal{I}$ represents the set of all available interventional variables such that after performing intervention $I \in \mathcal{I}$ on $G$, we observe $G_{\overline{I}}$. $\mathcal{I}$ includes $I = \emptyset$, which refers to "no intervention" and implies the original graph $G$ and the observational data $P(\mathcal{V})$.

**Definition D.5** (Sub-graph, $(G_{\overline{I}})_V$)**.** Let $G_V$ be a sub-graph of $G$ containing nodes in $V$ and all arrows between such nodes. $(G_{\overline{I}})_V$ refers to the sub-graph of $G_{\overline{I}}$ containing nodes in $V$ only, such that variable $I$ is intervened on i.e., all incoming edges to $I$ is cut off.

**Proposition D.6.** *Let $V \in \mathcal{V}, I \in \mathcal{I}$ be some arbitrary variable sets. The set of c-components formed from a sub-graph $(G_{\overline{I}})_V$ with intervention $I$ is not affected by additional interventions on their parents from outside of the sub-graph. Formally, $(G_{\overline{Pa(V) \cup I}})_V$ and $(G_{\overline{I}})_V$ has the same set of c-components.*

*Proof.* Let $C((G_{\overline{I}})_V)$ be the c-components which consists of nodes of $V$ in graph $(G_{\overline{I}})_V$. In sub-graph $(G_{\overline{Pa(V) \cup I}})_V$, no extra intervention is being done on any node in $V$ rather only on $Pa(V)$ where $V$ and $Pa(V)$ are two disjoint sets. Therefore, the c-components can be produced from this sub-graph will be same as for $G_{\overline{I}}$. i.e., $C(G_{V\overline{Pa(V) \cup I}}) = C(G_{\overline{I}})$. $\qquad\square$

**Lemma D.7.** *Let $V'$ be a set called focus-set. $V'$ and intervention $I$ be arbitrary subsets of observable variables $\mathcal{V}$ and $\{C_i\}_i$ be the set of c-components in $G_{\overline{I}}$. Let $\{Pa(V') \cup I\}$ be a set called action-set. and $S$ be a set called remain-set, defined as $S := \mathcal{V} \backslash \{V' \cup Pa(V') \cup I\}$, $S(i)$ as $S(i) = S \cap C_i$ i.e., some part of the remain-set that are located in c-component $C_i$. Thus, $S = \bigcup_i S(i)$. We also define active c-components $C_i^+$ as $C_i^+ := C_i \setminus \{S(i) \cup Pa(V') \cup I\}$ i.e., the variables in focus-set that are located in c-component $C_i$. Given these sets, Tian's factorization can be applied to a sub-graph under proper intervention. Formally, we can factorize as below:*

$$P_{Pa(V') \cup I}(V') = \prod_i P_{Pa(C_i^+) \cup I}(C_i^+)$$

*Proof.* $(G_{\overline{Pa(V') \cup I}})_{V'}$ and $(G_{\overline{I}})_{V'}$ have the same c-components according to Proposition D.6. According to Tian's factorization for causal effect identification (Tian & Pearl, 2002), we know that

$$P_{Pa(V') \cup I}(\mathcal{V}) = \prod_i P_{Pa(C_i) \cup Pa(V') \cup I}(C_i)$$

$$[\text{let } \eta = Pa(V') \cup I, \text{ i.e., action-set}]$$

$$\implies P_\eta(\eta) \times P_\eta(\mathcal{V} \setminus \eta | \eta) = \prod_i P_{Pa(C_i) \cup \eta}(C_i) \tag{11}$$

$$\implies P_\eta(\mathcal{V} \setminus \{Pa(V') \cup I\}) = \prod_i P_{Pa(C_i) \cup \eta}(C_i)$$

We ignore conditioning on action-set $\eta = Pa(V') \cup I$ since we are intervening on it. Now, we have a joint distribution of focus-set and remain-set with action-set as an intervention.

$$\implies P_\eta(V' \cup S) = \prod_i P_{Pa(C_i) \cup \eta}(C_i)$$

$$[\text{Here, } S := \mathcal{V} \backslash \{V' \cup Pa(V') \cup I\} \implies \mathcal{V} \setminus \{Pa(V') \cup I\} = V' \cup S]$$

$$\implies \sum_S P_\eta(V' \cup S) = \sum_S \prod_i P_{Pa(C_i) \cup \eta}(C_i) \tag{12}$$

$$\implies \sum_S P_\eta(V' \cup S) = \prod_i \sum_{S(i)} P_{Pa(C_i) \cup \eta}(C_i)$$

$$[\text{Since, } S(i) = S \cap C_i \text{ and } \forall(i,j), i \neq j, C_i \cap C_j = \emptyset \implies S_i \cap S_j = \emptyset]$$

Here, $\forall_i, S(i)$ are disjoint partitions of the variable set $S$ and contained in only c-component $C_i$, i.e, $S(i) = S \cap C_i$. Since $\forall_{i,j}, C_i \cap C_j = \emptyset$, this implies that $S_i \cap S_j = \emptyset$ would occur as well. Intuitively, remain-sets located in different c-components do not intersect. Therefore, each of the probability terms at R.H.S, $P_{Pa(C_i) \cup \eta}(C_i)$ is only a function of $S(i)$ instead of whole $S$. This gives us the opportunity to push the marginalization of $S(i)$ inside the product and marginalize the probability term. After marginalizing $S(i)$ from the joint, we define rest of the variables as active c-components $C_i^+$. The following Figure 9 helps to visualize all the sets.

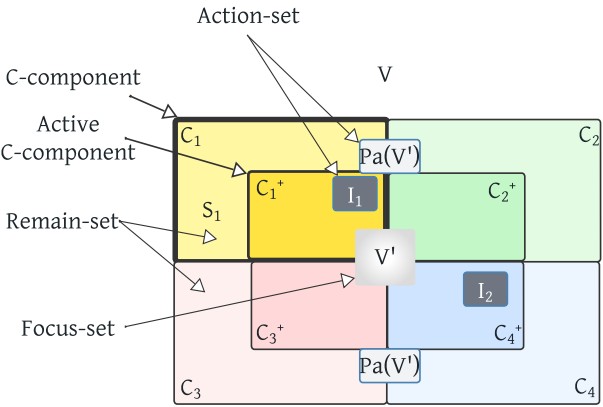

Figure 9: Visualization of focus-sets, action-sets, and remain-sets

We continue the derivation as follows:

$$\implies P_{Pa(V')\cup I}(V') = \prod_i P_{Pa(C_i)\cup Pa(V')\cup I}(C_i^+)$$

$$[Here, C_i^+ = C_i \setminus S(i), \text{ i.e., active c-component: focus-set elements located in } C_i]$$

$$= \prod_i P_{Pa(C_i^+)\cup I\cup\{Pa(C_i)\setminus Pa(C_i^+)\}\cup Pa(V')}(C_i^+)$$

$$= \prod_i P_{X\cup Z}(C_i^+) \quad [\text{Let, } X = Pa(C_i^+) \cup I \text{ and } Z = \{Pa(C_i)\cup Pa(V')\}\setminus X]$$

$$(13)$$

Here, we have variable set $C_i^+$ in the joint distribution. Now, if we intervene on the parents $Pa(C_i^+)$ and $I$, rest of the intervention which is outside $C_i^+$ becomes ineffective. Therefore, we have $X = Pa(C_i^+)\cup I$, the intervention which shilds the rest of the interventions, $Z = \{Pa(C_i)\cup Pa(V')\}\setminus X$. Therefore, we can apply do-calculus rule 3 on $Z$ and remove those interventions. Finally,

$$\implies P_{Pa(V')\cup I}(V') = \prod_i P_{Pa(C_i^+)\cup I}(C_i^+) \quad [\text{We apply Rule 3 since } C_i \perp\!\!\!\perp Z | X_{G_{\overline{X}}}] \quad (14)$$

$$\square$$

Corollary D.8, suggests that Tian's factorization can be applied on the h-nodes of $H^I \in \mathcal{H}$.

**Corollary D.8.** *Consider a causal graph $G$. Let $\{C_i\}_{i\in[t]}$ be the c-components of $G_{\overline{T}}$. For some intervention target $I$, let $H^I = (V_{\mathcal{H}^I}, E_{\mathcal{H}^I})$ be the h-graph constructed by Algorithm 6 where $V_{\mathcal{H}^I} = \{H_k^I\}_k$. Suppose $H_k^I$ is some node in $\mathcal{H}^I$. We have that $H_k^I = \{C_i\}_{i\in T_k^I}$ for some $T_k^I \subseteq [t]$. With slight abuse of notation we use $H_k^I$ interchangeably with the set of nodes that are in $H_k^I$. Then,*

$$P_{Pa(H_k^I)\cup I}(H_k^I) = \prod_{i\in[t]} P_{Pa(C_i)\cup I}(C_i) \quad (15)$$

*Proof.* Let, $V' = H_k^I, C_i^+ = C_i \setminus \emptyset = C_i$. Then, this corollary is direct application of Lemma D.7.

$$\square$$

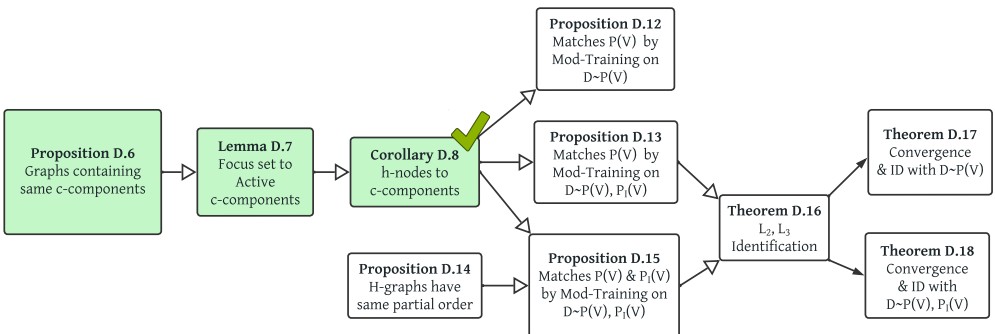

Figure 10: Flowchart of proofs

## D.5 Matching Distributions with WhatIfGAN Modular Training

We provide Definition D.2: $\mathcal{H}^I$-graph here again.

**Definition D.9** ($\mathcal{H}^I$-graph). For a post-interventional graph $G_{\overline{I}}$, let the set of c-components in $G_{\overline{I}}$ be $\mathcal{C} = \{C_1, \ldots C_t\}$. Choose a partition $\{H_k^I\}_k$ of $\mathcal{C}$ such that the $\mathcal{H}^I$-graph $\mathcal{H}^I = (V_{\mathcal{H}^I}, E_{\mathcal{H}^I})$, defined as follows, is acyclic: $V_{\mathcal{H}^I} = \{H_k^I\}_k$ and for any $s, t$, $H_s^I \to H_t^I \in E_{\mathcal{H}^I}$, iff $P(H_t^I | do(pa(H_t^I) \cap H_s^I)) \neq P(H_t^I | pa(H_t^I) \cap H_s^I)$, i.e., do-calculus rule-2 does not hold. Note that one can always choose a partition of $\mathcal{C}$ to ensure $\mathcal{H}^I$ is acyclic: The $\mathcal{H}^I$ graph with a single node $H_1^I = \mathcal{C}$ in $G_{\overline{I}}$. Even though $\mathcal{H}^I$ for different $I$ might have different partial order, during training, every $\mathcal{H}^I$ follows the partial order of $\mathcal{H}^{\emptyset}$. Since its partial order is valid for other H-graphs as well.

**Training order, $\mathcal{T}$:** We define a training order, $\mathcal{T} = \{\sigma_0, \ldots, \sigma_m\}$ where $\sigma_i = \{H_k\}_k$. If $H_{k_1}^I \to H_{k_2}^I$, $H_{k_1}^I \in \sigma_i$, $H_{k_2}^I \in \sigma_j$ then $i < j$.

**Definition D.10** (Notation for distributions). $Q(.)$ is the observational distribution induced by the deep causal SCM. $P(.)$ is the true (observational/interventional) distribution. With a slight abuse of notation, if we have $P(\mathbf{V})$ and intervention $I$, then $P_I(\mathbf{V})$ indicates $P_I(\mathbf{V} \setminus I)$. Algorithm 7 is said to have converged if training attains zero loss every time line 14 is visited.

**Definition D.11** (Ancestor set $\mathcal{A}_I$ in $G_{\overline{I}}$). Let parents of a variable set $\mathbf{V}$ be $Pa(\mathbf{V}) = \bigcup_{V \in \mathbf{V}} Pa(V) \setminus \mathbf{V}$. Now, for some h-node $H_k^I \in \mathcal{H}^I$-graph, we define $\mathcal{A}_I :=$ the minimal subset of ancestors exists in the causal graph $G_I$ with intervention $I$ such that the following holds,

$$p(H_n^I \cup \mathcal{A}_I | do(pa(H_n^I \cup \mathcal{A}_I)), do(I) = p(H_n^I \cup \mathcal{A}_I | pa(H_n^I \cup \mathcal{A}_I), do(I)) \tag{16}$$

For training any h-node in the training order $\mathcal{T} = \{\sigma_0, \ldots, \sigma_m\}$, i.e., $H_k^{\emptyset} \in \sigma_j$, $0 < j \leq m$, if only observational data is available, i.e., $I = \emptyset$, we search for an ancestor set $\mathcal{A}_{\emptyset}$ such that $\mathcal{A}_{\emptyset}$ satisfies modularity condition for $H_k^{\emptyset}$:

$$P(H_k^{\emptyset} \cup \mathcal{A}_{\emptyset} | do(pa(H_k^{\emptyset} \cup \mathcal{A}_{\emptyset}))) = P(H_k^{\emptyset} \cup \mathcal{A}_{\emptyset} | pa(H_k^{\emptyset} \cup \mathcal{A}_{\emptyset})) \tag{17}$$

Similarly, for $I \in An_G(H_k^I)$, i.e., intervention on ancestors, we can learn $P_{pa(H_k^{\emptyset})}(H_k^{\emptyset})$ from available interventional datasets since $H_k^I = H_k^{\emptyset}$, i.e., contains the same c-factors, according to $\mathcal{H}^I$-graphs construction. These c-factor distributions are identifiable from $P_I(V)$ as they can be calculated from the c-factorization of $P_I(V)$. Thus we have,

$$P_{pa(H_k^I)}(H_k^I) = P_{pa(H_k^{\emptyset})}(H_k^{\emptyset}) \tag{18}$$

Therefore, to utilize ancestor interventional datasets, We search for smallest ancestor set $\mathcal{A}_I \subseteq An_{G_{\overline{I}}}(H_k^I)$ in $G_{\overline{I}}$ such that do-calculus rule-2 applies,

$$P(H_k^I \cup \mathcal{A}_I | do(pa(H_k^I \cup \mathcal{A}_I)), do(I)) = P(H_k^I \cup \mathcal{A}_I | pa(H_k^I \cup \mathcal{A}_I), do(I)) \tag{19}$$

Then we can train the mechanisms in $H_k^\emptyset$ to learn the $P(\mathcal{V})$ c-factors by matching the following alternative distribution from $D^I$ dataset,

$$
\begin{aligned}
& P(H_k^I \cup \mathcal{A}_I | pa(H_k^I \cup \mathcal{A}_I), \mathrm{do}(I)) = Q(H_k^I \cup \mathcal{A}_I | \mathrm{do}(pa(H_k^I \cup \mathcal{A}_I)), \mathrm{do}(I)) \\
\implies & P(H_k^I \cup \mathcal{A}_I | \mathrm{do}(pa(H_k^I \cup \mathcal{A}_I)), \mathrm{do}(I)) = Q(H_k^I \cup \mathcal{A}_I | \mathrm{do}(pa(H_k^I \cup \mathcal{A}_I)), \mathrm{do}(I))
\end{aligned}
\tag{20}
$$

Matching the alternative distributions with $D^I$ will imply that we match $P(\mathcal{V})$ c-factor as well. Formally:

$$
\begin{aligned}
Q_{pa(H_k^I)}(H_k^I) &= P_{pa(H_k^I)}(H_k^I) \\
Q_{pa(H_k^I)}(H_k^I) &= P_{pa(H_k^\emptyset)}(H_k^\emptyset)
\end{aligned}
\tag{21}
$$

### D.5.1 MATCHING OBSERVATIONAL DISTRIBUTIONS WITH MODULAR TRAINING ON $D \sim P(\mathcal{V})$

Now, we provide the theoretical proof of the correctness of WhatIfGAN Modular Training matching observational distribution by training on observational dataset $D^{\tilde{p}}$. Since, we have access to only observational data we remove the intervention-indicating superscript/subscript and address $\mathcal{H}^{\tilde{p}}$ as $\mathcal{H}$, ancestor set $\mathcal{A}_I$ as $\mathcal{A}$ and dataset $D^{\tilde{p}}$ as $D$.

**Proposition D.12.** *Suppose Algorithm 2: **WhatIfGAN Modular Training** converges for each h-node in $\mathcal{H}^\emptyset$-graph constructed from $G = (\mathcal{V}, \mathcal{E})$. Suppose the observational distribution induced by the deep causal model is $Q(\mathcal{V})$ after training on data sets $D \sim P(\mathcal{V})$. Then,*

$$
P(\mathcal{V}) = Q(\mathcal{V})
\tag{22}
$$

*Proof.* According to Tian's factorization we can factorize the joint distributions into c-factors as follows:

$$
P(\mathcal{V}) = P(\mathcal{H}) = \prod_{H_k \in \mathcal{H}} \prod_{C_i \in H_k} P_{pa(C_i)}(C_i)
\tag{23}
$$

We can divide the set of c-components $\mathcal{C} = \{C_1, \dots C_t\}$ into disjoint partitions or h-nodes as $H_k = \{C_i\}_{i \in T_k}$ for some $T_k \subseteq [t]$. Following Corollary D.8, we can combine the c-factors in each partitions and rewrite it as:

$$
\prod_{H_k \in \mathcal{H}} \prod_{C_i \in H_k} P_{pa(C_i)}(C_i) = P_{pa(H_0)}(H_0) \times P_{pa(H_1)}(H_1) \times \dots \times P_{pa(H_n)}(H_n)
\tag{24}
$$

Now, we prove that we match each of these terms according to the training order $\mathcal{T}$.

**For any root h-nodes $H_k \in \sigma_0$ :**
Due to the construction of $\mathcal{H}$ graphs in Algorithm 6, the following is true for any root nodes, $H_k \in \sigma_0$.

$$
P(H_k | Pa(H_k)) = P_{Pa(H_k)}(H_k)
\tag{25}
$$

WhatIfGAN training convergence for the DCM in $H_k \in \sigma_0$. (Algorithm 2, line 8) ensures that the following matches:

$$
\begin{aligned}
& P(H_k | Pa(H_k)) = Q_{Pa(H_k)}(H_k) \\
\implies & P_{Pa(H_k)}(H_k) = Q_{Pa(H_k)}(H_k)
\end{aligned}
\tag{26}
$$

Since, Equation 25 is true, observational data is sufficient for training the mechanisms in $H_k \in \sigma_0$. Thus, we do not need to train on interventional data.

**For the h-node $H_k \in \sigma_1$ :**
Now we show that we can train mechanisms in $H_k$ by matching $P(\mathcal{V})$ c-factors with $D \sim P(\mathcal{V})$ data set. Let us assume, $\exists \mathcal{A} \subseteq \sigma_0$ such that $\mathcal{A} = An(H_k)$,i.e., ancestors set of $H_k$ in the $\mathcal{H}$-graph that we have already trained with available $D$ dataset. To apply Lemma D.7 in causal graph $G$, consider $V' = H_k \cup \mathcal{A}$ as the focus-set, $Pa(V')$ as the action-set. Thus, active c-components: $C_j^+ := C_j \cap V'$.

Then we get the following:

$$P(H_k \cup \mathcal{A}|\text{do}(Pa(H_k \cup \mathcal{A}))) = \prod_{C_i \in H_k} P_{pa(C_i)}(C_i) \times \prod_{H_S \in \{\mathcal{A}\}} \prod_{C_j^+ \subseteq H_S} P_{Pa(C_j^+)}(C_j^+)$$

[Here, 1st term is the factorization of the current h-node and 2nd term is the factorization of the ancestors set.] (27)

$$\implies P(H_k \cup \mathcal{A}|\text{do}(Pa(H_k \cup \mathcal{A}))) = P_{pa(H_k)}(H_k) * \prod_{H_S \in \mathcal{A}} \prod_{C_j^+ \subseteq H_S} P_{Pa(C_j^+)}(C_j^+)$$

Here according to Corollary D.8, we combine the c-factors $P_{pa(C_i)}(C_i)$ for c-components in $H_k$ to form $P_{pa(H_k)}(H_k)$. We continue the derivation as follows:

$$\implies P_{Pa(H_k)}(H_k) = \frac{P(H_k \cup \mathcal{A}|\text{do}(Pa(H_k \cup \mathcal{A})))}{\prod_{H_S \in \mathcal{A}} \prod_{C_j^+ \subseteq H_S} P_{Pa(C_j^+)}(C_j^+)}$$

$$\implies P_{Pa(H_k)}(H_k) = \frac{Q(H_k \cup \mathcal{A}|\text{do}(Pa(H_k \cup \mathcal{A})))}{\prod_{H_S \in \mathcal{A}} \prod_{C_j^+ \subseteq H_S} Q_{Pa(C_j^+)}(C_j^+)}$$

(28)

Here the R.H.S numerator follows from previous line according to Equation 20. For the denominator at R.H.S, $\forall H_S \in \mathcal{A}$, we have already matched $P(H_S \cup \mathcal{A}|\text{do}(pa(H_S \cup \mathcal{A})))$, during training of $\mathcal{A} = An(H_k)$ h-nodes. According to Lemma D.7, matching these distribution is sufficient to match the distribution at R.H.S denominator. Therefore, our DCM will produce the same distribution as well. This implies that from Equation 28 we get,

$$P_{pa(H_k)}(H_k) = Q_{pa(H_k)}(H_k)$$

$$\implies P_{pa(H_k)}(H_k) = Q_{pa(H_k)}(H_k) \quad \text{[According to Equation 18]}$$

(29)

Similarly, we train each h-node following the training order $\mathcal{T}$ and match the distribution in Equation 24. This finally shows that,

$$P(V) = \prod_{j \leq n} P_{pa(H_j)}(H_j) = \prod_{j \leq n} Q_{pa(H_j)}(H_j) = Q(V)$$

(30)

$\square$

### D.5.2 MATCHING OBSERVATIONAL DISTRIBUTIONS WITH MODULAR TRAINING ON $\mathbf{D} \sim P_I(\mathcal{V}), \forall I \in \mathcal{I}$

Now, we provide the theoretical proof of the correctness of WhatIfGAN Modular Training matching observational distribution from multiple datasets $D^I, \forall I \in \mathcal{I}$.

**Proposition D.13.** *Suppose Algorithm 7: **WhatIfGAN Modular Training** converges for each h-node in $\mathcal{H}^\emptyset$-graph constructed from $G = (\mathcal{V}, \mathcal{E})$. Suppose the observational distribution induced by the deep causal model is $Q(\mathcal{V})$ after training on data sets $\mathbf{D}^I, \forall I \in \mathcal{I}$. Then,*

$$P(\mathcal{V}) = Q(\mathcal{V}) \tag{31}$$

*Proof.* According to Tian's factorization we can factorize the joint distributions into c-factors as follows:

$$P(\mathcal{V}) = P(H^\emptyset) = \prod_{H_k^\emptyset \in H^\emptyset} \prod_{C_i \in H_k^\emptyset} P_{pa(C_i)}(C_i) \tag{32}$$

We can divide the set of c-components $\mathcal{C} = \{C_1, \ldots C_t\}$ into disjoint partitions or h-nodes as $H_k^{\tilde{p}} = \{C_i\}_{i \in T_k}$ for some $T_k \subseteq [t]$. Following Corollary D.8, we can combine the c-factors in each partitions and rewrite it as:

$$\prod_{H_k^\emptyset \in H^\emptyset} \prod_{C_i \in H_k^\emptyset} P_{pa(C_i)}(C_i) = P_{pa(H_0^\emptyset)}(H_0^\emptyset) \times P_{pa(H_1^\emptyset)}(H_1^\emptyset) \times \ldots \times P_{pa(H_n^\emptyset)}(H_n^\emptyset) \tag{33}$$

Now, we prove that we match each of these terms according to the training order $\mathcal{T}$.

**For any root h-nodes $H_k^\emptyset \in \sigma_0$ :**

Due to the construction of $\mathcal{H}^\emptyset$ graphs in Algorithm 6, the following is true for any root nodes, $H_k^\emptyset \in \sigma_0$.

$$P(H_k^\emptyset | Pa(H_k^\emptyset)) = P_{Pa(H_k^\emptyset)}(H_k^\emptyset) \tag{34}$$

WhatIfGAN training convergence for the DCM in $H_k^\emptyset \in \sigma_0$. (Algorithm 7, line 14) ensures that the following matches:

$$P(H_k^\emptyset | Pa(H_k^\emptyset)) = Q_{Pa(H_k^\emptyset)}(H_k^\emptyset)$$
$$\implies P_{Pa(H_k^\emptyset)}(H_k^\emptyset) = Q_{Pa(H_k^\emptyset)}(H_k^\emptyset) \tag{35}$$

Since, Equation 34 is true, observational data is sufficient for training the mechanisms in $H_k^\emptyset \in \sigma_0$. Thus, we do not need to train on interventional data.

**For the h-node $H_k^\emptyset \in \sigma_1$ :**

Now we show that we can train mechanisms in $H_k^\emptyset$ by matching $P(\mathcal{V})$ c-factors with either $\mathcal{L}_1$ or $\mathcal{L}_2$ datasets. Let us assume, $\exists \mathcal{A}_I \subseteq \sigma_0$ such that $\mathcal{A}_I = An_{G_{\overline{T}}}(H_k^I)$,i.e., ancestors set of $H_k^I$ in the $\mathcal{H}^I$-graph that we have already trained with available $D^I$ dataset. To apply Lemma D.7 in $G_{\overline{T}}$ with $|I| \geq 0$, consider $V' = H_k^I \cup \mathcal{A}_I$ as the focus-set, $\{Pa(V') \cup I\}$ as the action-set. Thus, active c-components: $C_j^+ := C_j \cap V'$

Then we get the following:

$$P(H_k^I \cup \mathcal{A}_I | \mathrm{do}(Pa(H_k^I \cup \mathcal{A}_I)), \mathrm{do}(I)) = \prod_{C_i \in H_k^I} P_{pa(C_i)}(C_i) \times \prod_{H_S^I \in \{\mathcal{A}_I\}} \prod_{C_j^+ \subseteq H_S^I} P_{Pa(C_j^+) \cup I}(C_j^+)$$

[Here, 1st term is the factorization of the current h-node and 2nd term is the factorization of the ancestors set.]

$$\implies P(H_k^I \cup \mathcal{A}_I | \mathrm{do}(Pa(H_k^I \cup \mathcal{A}_I)), \mathrm{do}(I)) = P_{pa(H_k^I)}(H_k^I) * \prod_{H_S^I \in \mathcal{A}_I} \prod_{C_j^+ \subseteq H_S^I} P_{Pa(C_j^+) \cup I}(C_j^+) \tag{36}$$

Here according to Corollary D.8, we combine the c-factors $P_{pa(C_i)}(C_i)$ for c-components in $H_k^I$ to form $P_{pa(H_k^I)}(H_k^I)$. We continue the derivation as follows:

$$\implies P_{Pa(H_k^I)}(H_k^I) = \frac{P(H_k^I \cup \mathcal{A}_I | \mathrm{do}(Pa(H_k^I \cup \mathcal{A}_I)), \mathrm{do}(I))}{\prod_{H_S^I \in \mathcal{A}_I} \prod_{C_j^+ \subseteq H_S^I} P_{Pa(C_j^+) \cup I}(C_j^+)}$$

$$\implies P_{Pa(H_k^I)}(H_k^I) = \frac{Q(H_k^I \cup \mathcal{A}_I | \mathrm{do}(Pa(H_k^I \cup \mathcal{A}_I)), \mathrm{do}(I))}{\prod_{H_S^I \in \mathcal{A}_I} \prod_{C_j^+ \subseteq H_S^I} Q_{Pa(C_j^+) \cup I}(C_j^+)} \tag{37}$$

Here the R.H.S numerator follows from previous line according to Equation 20. For the denominator at R.H.S, the intervention is an ancestor of the current hnode, i.e., $I \in \{An(H_k^I) \setminus H_k^I\}$. Now, $\forall H_S^I \in \mathcal{A}_I$, we have already matched $P(H_S^I \cup \mathcal{A}_I | \mathrm{do}(pa(H_S^I \cup \mathcal{A}_I)), \mathrm{do}(I))$, during training of $\mathcal{A}_I = An(H_k^I)$ h-nodes. According to Lemma D.7, matching these distribution is sufficient to match the distribution at R.H.S denominator. Therefore, our DCM will produce the same distribution as well. This implies that from Equation 37 we get,

$$P_{pa(H_k^I)}(H_k^I) = Q_{pa(H_k^I)}(H_k^I)$$
$$\implies P_{pa(H_k^\emptyset)}(H_k^\emptyset) = Q_{pa(H_k^\emptyset)}(H_k^\emptyset) \quad \text{[According to Equation 18]} \tag{38}$$

Similarly, we train each h-node following the training order $\mathcal{T}$ and match the distribution in Equation 33. This finally shows that,

$$P(V) = \prod_{j \le n} P_{pa(H_j^\emptyset)}(H_j^\emptyset) = \prod_{j \le n} Q_{pa(H_j^\emptyset)}(H_j^\emptyset) = Q(V) \tag{39}$$

$\square$

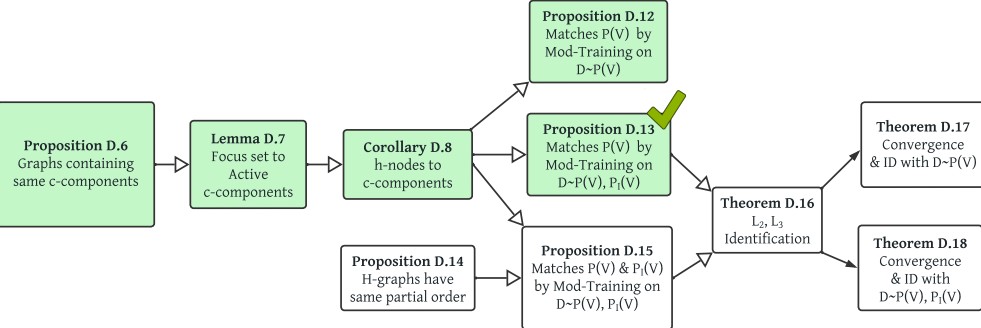

Figure 11: Flowchart of proofs

### D.5.3 MATCHING INTERVENTIONAL DISTRIBUTIONS WITH MODULAR TRAINING ON $\mathbf{D} \sim P_I(\mathcal{V}), \forall I \in \mathcal{I}$

Before showing our algorithm correctness with both observational and interventional data, we first discuss the DAG property of $\mathcal{H}$-graphs.

**Proposition D.14.** *Any $\mathcal{H}$-graph constructed according to Definition D.2 is a directed acyclic graph (DAG) and a common partial order $\mathcal{T}$, exists for all $\mathcal{H}^I$-graphs, $\forall I \in \mathcal{I}$.*

*Proof.* We construct the $\mathcal{H}$-graphs following Algorithm 6. By checking the modularity condition we add edges between any two h-nodes. However, if we find a cycle $H_j^I, H_k^I \to \ldots \to H_j^I$, then we combine all h-nodes in the cycle and form a new h-node in $\mathcal{H}^I$. This new h-node contains the union of all outgoing edges to other h-nodes. Therefore, at the end of the algorithm, the final $\mathcal{H}$-graph, $\mathcal{H}^I$ will always be a directed acyclic graph. Note that one can always choose a partition of the c-components $\mathcal{C}$ to ensure $\mathcal{H}^I$ is acyclic: The $\mathcal{H}^I$ graph with a single node $H_1^I = \mathcal{C}$.

Next in Algorithm 7, training is performed according to the partial order of $\mathcal{H}^\emptyset$ which corresponds to the original graph $G$ without any intervention. This is the most dense $\mathcal{H}$-graph and thus imposes the most restrictions in terms of the training order. Let $I$ be an intervention set. For any intervention $I$, suppose $\mathcal{H}^I$-graph is obtained from $G_{\overline{I}}$ and $I$ is located in $H_k^\emptyset$ h-node of $\mathcal{H}^{\tilde{p}}$-graph obtained from $G$. The only difference between $\mathcal{H}^\emptyset$ and $\mathcal{H}^I$ is that the h-node $H_k^\emptyset$ might be split into multiple new h-nodes in $\mathcal{H}^I$ and some edges with other h-nodes that were present in $\mathcal{H}^\emptyset$, might be removed in $\mathcal{H}^I$.

However, according to Algorithm 7, we do not split these new h-nodes rather bind them together to form $H_k^I$ that contain the same nodes as $H_k^\emptyset$. Therefore, no new edge is being added among other h-nodes. This implies that the partial order of $\mathcal{H}^\emptyset$ is also valid for $\mathcal{H}^I$. After intervention no new edges are added to the constructed $\mathcal{H}$-graphs, thus we can safely claim that,

$$An_{G_{\overline{I}}}(H_k^I) \subseteq An_G(H_k^\emptyset), \forall I \in \mathcal{I} \tag{40}$$

Since all $\mathcal{H}$-graphs are DAGs and the above condition holds, any valid partial order for $\mathcal{H}^\emptyset$ is also a valid partial order for all $\mathcal{H}^I, \forall I \in \mathcal{I}$, i.e., they have a common valid partial order.

$\square$

In general, for $I \in H_k^I$, i.e., when the intervention is inside $H_k^I$, we utilize interventional datasets and search for minimum size variable set $\mathcal{A}_I \subseteq An_{G_{\overline{T}}}(H_k^I)$ in $G_{\overline{T}}$ such that do-calculus rule-2 satisfies,

$$P(H_k^I \cup \mathcal{A}_I | pa(H_k^I \cup \mathcal{A}_I), \mathrm{do}(I)) = P(H_k^I \cup \mathcal{A}_I | \mathrm{do}(pa(H_k^I \cup \mathcal{A}_I)), \mathrm{do}(I)) \tag{41}$$

Then we can train the mechanisms in $H_k^I$ to match the following distribution,

$$P(H_k^I \cup \mathcal{A}_I | pa(H_k^I \cup \mathcal{A}_I), \mathrm{do}(I)) = Q(H_k^I \cup \mathcal{A}_I | \mathrm{do}(pa(H_k^I \cup \mathcal{A}_I)), \mathrm{do}(I))$$
$$\implies P(H_k^I \cup \mathcal{A}_I | \mathrm{do}(pa(H_k^I \cup \mathcal{A}_I)), \mathrm{do}(I) = Q(H_k^I \cup \mathcal{A}_I | \mathrm{do}(pa(H_k^I \cup \mathcal{A}_I)), \mathrm{do}(I)) \tag{42}$$

**Proposition D.15.** *Suppose Algorithm 7: **WhatIfGAN Modular Training** converges for each h-node in $\mathcal{H}^{\emptyset}$-graph constructed from $G = (\mathcal{V}, \mathcal{E})$. Suppose the interventional distribution induced by the deep causal model is $Q_I(V)$ after training on data sets $\mathbf{D}^I, \forall I \in \mathcal{I}$. then,*

$$P_I(V) = Q_I(V) \tag{43}$$

*Proof.* Suppose, intervention $I$ belongs to a specific c-component $C_i$, i.e., $I \in C_i$. According to Tian's factorization, we can factorize the $\mathrm{do}(I)$ interventional joint distributions for $G_{\overline{I}}$ causal graph, into c-factors as follows:

$$P_I(V) = P_I(\mathcal{H}^I) = P_{pa(C_i) \cup I}(C_i) \times \prod_{H_k^I \in \mathcal{H}^I} \prod_{C_{i'} \in H_k^I} P_{pa(C_{i'})}(C_{i'}) \tag{44}$$

The difference between the c-factorization for $P(V)$ and $P_I(V)$ is that when intervention $I$ is located inside c-component $C_i$, we have $P_{pa(C_i) \cup I}(C_i)$ instead of $P_{pa(C_i)}(C_i)$. We can divide c-components $\mathcal{C} = \{C_1, \ldots C_t\}$ into disjoint partitions or h-nodes as $H_k^{\tilde{p}} = \{C_i\}_{i \in T_k}$ for some $T_k \subseteq [t]$.

Let, the c-component $C_i$ that contains intervention $I$ belong to hnode $H_k^I$, i.e., $C_i \in H_k^I$. Following Corollary D.8, we can combine the c-factors in each partitions and rewrite R.H.S of Equation 44 as:

$$P_{pa(C_i) \cup I}(C_i) \times \prod_{H_k^I \in \mathcal{H}^I} \prod_{C_{i'} \in H_k^I} P_{pa(C_{i'})}(C_{i'}) = P_{pa(H_k^I) \cup I}(H_k^I) \times \prod_{H_{k'}^I \in \{\mathcal{H}^I \setminus H_k^I\}} P_{pa(H_{k'}^I)}(H_{k'}^I) \tag{45}$$

Now, we prove that we match each of these terms in Equation 45 according to the training order $\mathcal{T}$.
**For any root h-nodes $H_k^I \in \sigma_0$:**
Due to the construction of $H^I$ graphs in Algorithm 6, the following is true for any root nodes, $H_k^I \in \sigma_0$.

$$P_I(H_k^I | Pa(H_k^I)) = P_{Pa(H_k^I) \cup I}(H_k^I) \tag{46}$$

WhatIfGAN training convergence for the mechanisms in $H_k^I \in \sigma_0$. Algorithm 7, line 14 ensures that the following matches:

$$P_I(H_k^I | Pa(H_k^I)) = Q_{Pa(H_k^I) \cup I}(H_k^I)$$
$$\implies P_{Pa(H_k^I) \cup I}(H_k^I) = Q_{Pa(H_k^I) \cup I}(H_k^I) \tag{47}$$

**For the h-node $H_k^I \in \sigma_1$ with $I \in H_k^I$:**
Now we show that we can train mechanisms in $H_k^I$ by matching $P_I(\mathcal{V})$ c-factors with $\mathcal{L}_1$ and $\mathcal{L}_2$ datasets. Let us assume, $\exists \mathcal{A}_I \subseteq \sigma_0$ such that $\mathcal{A}_I = An_{G_{\overline{T}}}(H_k^I)$, i.e., ancestors of $H_k^I$ in the $\mathcal{H}^I$-graph that we have already trained with available $D^I$ dataset, $\forall I \in \mathcal{I}$.

To apply Lemma D.7 in $G_{\overline{I}}$ with $|I| \geq 0$, consider $V' = H_k^I \cup \mathcal{A}_I$ as the focus-set, $\{Pa(V') \cup I\}$ as the action-set. Thus, active c-components: $C_j^+ := C_j \cap V'$. We apply the lemma as below:

$$P(H_k^I \cup \mathcal{A}_I | \mathrm{do}(Pa(H_k^I \cup \mathcal{A}_I)), \mathrm{do}(I)) = \prod_{C_i \in H_k^I} P_{pa(C_i) \cup I}(C_i) \times \prod_{H_S^I \in \mathcal{A}_I} \prod_{C_{i'}^+ \in H_S^I} P_{Pa(C_{i'}^+)}(C_{i'}^+) \tag{48}$$

Here, the 1st term is the factorization of the current h-node and the 2nd term is the factorization of the ancestors set. The intervened variable $I$ is located in the current h-node $H_k^I$. Therefore, the factorized

c-components, i.e., $C_i \in H_k^I$ has $I$ as intervention along with their parent intervention. The above equation implies:

$$P(H_k^I \cup \mathcal{A}_I | \mathrm{do}(Pa(H_k^I \cup \mathcal{A}_I)), \mathrm{do}(I)) = P_{pa(H_k^I) \cup I}(H_k^I) \times \prod_{H_S^I \in \mathcal{A}_I} \prod_{C_{i'}^+ \in H_S^I} P_{Pa(C_{i'}^+)}(C_{i'}^+) \quad (49)$$

According to Corollary D.8, we combine the c-factors $P_{pa(C_i) \cup I}(C_i)$ for c-components in $H_k^I$ to form $P_{pa(H_k^I) \cup I}(H_k^I)$. We continue the derivation as follows:

$$\begin{aligned} \implies P_{Pa(H_k^I) \cup I}(H_k^I) &= \frac{P(H_k^I \cup \mathcal{A}_I | \mathrm{do}(Pa(H_k^I \cup \mathcal{A}_I)), \mathrm{do}(I))}{\prod_{H_S^I \in \mathcal{A}_I} \prod_{C_j^+ \subseteq H_S^I} P_{Pa(C_j^+)}(C_j^+)} \\ \implies P_{Pa(H_k^I) \cup I}(H_k^I) &= \frac{Q(H_k^I \cup \mathcal{A}_I | \mathrm{do}(Pa(H_k^I \cup \mathcal{A}_I)), \mathrm{do}(I))}{\prod_{H_S^I \in \mathcal{A}_I} \prod_{C_j^+ \subseteq H_S^I} Q_{Pa(C_j^+)}(C_j^+)} \end{aligned} \quad (50)$$

Here the R.H.S numerator follows from previous line according to Equation 42 since training has converged for the current h-node. For the R.H.S, denominator, $\forall H_S^I \in \mathcal{A}_I$ appear before $H_k^I$ in the partial order. When we trained h-nodes $H_S^I \in \mathcal{A}_I$ on $P(\mathcal{V})$ and $P_I(\mathcal{V})$ datasets, we matched the joint distribution $P(H_S^I \cup \mathcal{A}_I | \mathrm{do}(pa(H_S^I \cup \mathcal{A}_I)), \mathrm{do}(I)), \forall H_S^I \in \mathcal{A}_I$. According to Lemma D.7, matching these distribution is sufficient to match the distribution at the R.H.S denominator. Therefore, our DCM will produce the same distribution as well. This implies that from Equation 50 we get,

$$P_{pa(H_k^I) \cup I}(H_k^I) = Q_{pa(H_k^I) \cup I}(H_k^I) \quad (51)$$

Similarly, we train each h-node following the training order $\mathcal{T}$ and match the distribution in Equation 45. We train the c-factor that contains interventions with our available interventional dataset and the c-factors that do not include any interventions can be trained with $P(V)$ dataset. This finally shows that,

$$\begin{aligned} P_I(V) &= P_{pa(H_k^I) \cup I}(H_k^I) \times \prod_{H_{k'}^I \in \{H^I \setminus H_k^I\}} P_{pa(H_{k'}^I)}(H_{k'}^I) \\ &= Q_{pa(H_k^I) \cup I}(H_k^I) \times \prod_{H_{k'}^I \in \{H^I \setminus H_k^I\}} Q_{pa(H_{k'}^I)}(H_{k'}^I) \\ &= Q_I(V) \end{aligned} \quad (52)$$

$\square$

Figure 12: Flowchart of proofs

## D.6  IDENTIFIABILITY OF ALGORITHM 7:WHATIFGAN MODULAR TRAINING

**Theorem D.16.** *Let $\mathcal{M}_1$ be the true SCM and Algorithm 7:* **WhatIfGAN Modular Training** *converge for each h-node in $\mathcal{H}$ constructed from $G = (\mathcal{V}, \mathcal{E})$ after training on data sets $\mathbf{D} = \{\mathcal{D}^I\}_{\forall I \in \mathcal{I}}$ and output the DCM $\mathcal{M}_2$. Then for any $\mathcal{L}_2, \mathcal{L}_3$ causal query $\mathcal{K}_{\mathcal{M}_1}(\mathcal{V})$, identifiable from $\mathbf{D}$, $\mathcal{K}_{\mathcal{M}_1}(\mathcal{V}) = \mathcal{K}_{\mathcal{M}2}(\mathcal{V})$ holds.*

*Proof.* Let $\mathcal{M}_1 = (G = (\mathcal{V}, \mathcal{E}), \mathcal{N}, \mathcal{U}, \mathcal{F}, P(.))$ be the true SCM and $\mathcal{M}_2 = (G, \mathcal{N}', \mathcal{U}', \mathcal{F}', Q(.))$ be the deep causal generative model represented by WhatIfGAN. For any $H_k^I \in \mathcal{H}^I, I \in \mathcal{I}$, we observe the joint distribution $P(H_k^I \cup \mathcal{A}^I \cup Pa(H_k^I \cup \mathcal{A}^I), \text{do}(I))$ in the input $D^I$ datasets. Thus we can train all the mechanisms in the current h-node $H_k^I$ by matching the following distribution from the partially observable datasets:

$$P(H_k^I \cup \mathcal{A}_I | pa(H_k^I \cup \mathcal{A}_I), \text{do}(I)) = Q(H_k^I \cup \mathcal{A}_I | \text{do}(pa(H_k^I \cup \mathcal{A}_I)), \text{do}(I)) \tag{53}$$

Now, as we are following a valid partial order of the $\mathcal{H}^\emptyset$-graph to train the h-nodes, we train the mechanisms of each h-node to match the input distribution only once and do not update it again anytime during the training of rest of the network. As we move to the next h-node of the partial order for training, we can keep the weights of the Ancestor h-nodes fixed and only train the current one and can successfully match the joint distribution in Equation 53. In the same manner, we would be able to match the distributions for each h-node and reach convergence for each of them. WhatIfGAN Training convergence implies that $Q_I(\mathcal{V}) = P_I(\mathcal{V}), \forall I \in \mathcal{I}$ i.e., for all input dataset distributions. Therefore, according to Theorem 4.2, WhatIfGAN is capable of producing samples from correct interventional or counterfactual distributions that are identifiable from the input distributions. □

**Theorem D.17.** *Suppose Algorithm 2:* **WhatIfGAN Modular Training** *converges for each h-node in the $\mathcal{H}$-graph constructed from $G = (\mathcal{V}, \mathcal{E})$ and after training on observational dataset $D \sim P(\mathcal{V})$, the observational distribution induced by the DCM is $Q(V)$. Then, we have $i) P(V) = Q(V)$, and $ii)$ for any $\mathcal{L}_2$ or $\mathcal{L}_3$ causal query $\mathcal{K}_{\mathcal{M}_1}(\mathcal{V})$ that is identifiable from $\mathbf{D}$, we have $\mathcal{K}_{\mathcal{M}_1}(\mathcal{V}) = \mathcal{K}_{\mathcal{M}2}(\mathcal{V})$*

*Proof.* Theorem 4.5 is restated here. The first part of the theorem is proved in Proposition D.12. The second part can be proved with Theorem D.16. □

**Theorem D.18.** *Suppose Algorithm 7:* **WhatIfGAN Modular Training** *converges for each h-node in the $\mathcal{H}^\emptyset$-graph constructed from $G = (\mathcal{V}, \mathcal{E})$ and after training on observational and interventional datasets $\mathbf{D}^I \sim P_I(\mathcal{V}) \forall I \in \mathcal{I}$, the distribution induced by the DCM is $Q_I(V), \forall I \in \mathcal{I}$. Then, we have $i) P_I(V) = Q_I(V)$, and $ii)$ for any $\mathcal{L}_2$ or $\mathcal{L}_3$ causal query $\mathcal{K}_{\mathcal{M}_1}(\mathcal{V})$ that is identifiable from $\mathbf{D}^I, \forall I \in \mathcal{I}$, we have $\mathcal{K}_{\mathcal{M}_1}(\mathcal{V}) = \mathcal{K}_{\mathcal{M}2}(\mathcal{V})$*

*Proof.* The first part of the theorem is proved in Proposition D.13 and Proposition D.15. Then it is a direct implication of Theorem D.16, This theorem is equivalent to Theorem 4.5 if we consider $\mathcal{I} = \{\tilde{p}\}$. □

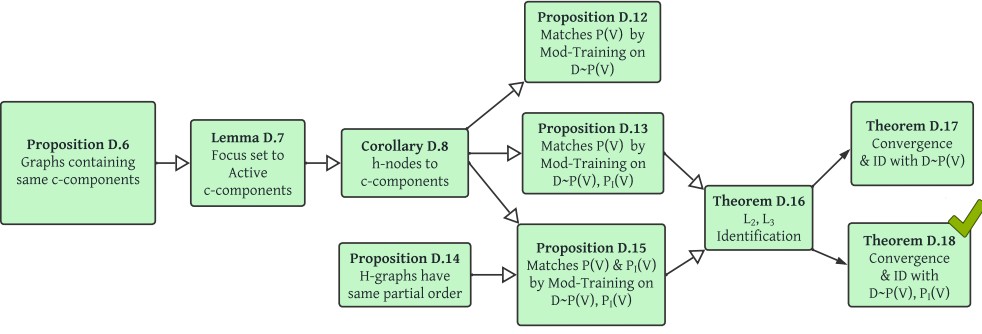

Figure 13: Flowchart of proofs

# E    MODULAR TRAINING ON DIFFERENT GRAPHS

## E.1    MODULAR TRAINING EXAMPLE 1

We provide an example in Figure 14 to better understand our modular algorithm.

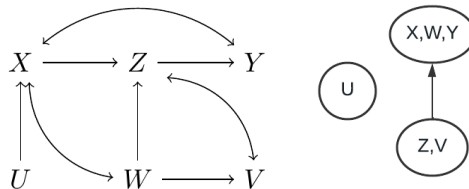

Figure 14: Example graphs for mechanism training

- Train the mechanism of $U$ using $\mathcal{D}_0 \sim p(U)$

- Leverage the fact that $P(z, v | \text{do}(x, w)) = p(z, v | x, w)$ to train the networks of $Z, V$ by intervening on $X, W$ and minimize the loss $\mathcal{L}((Z_{x,w}^F, V_{x,w}^F) || (Z_{x,w}^R, V_{x,w}^R))$

- Consider that $p(x, y, w, z | \text{do}(u)) = p(x, y, w, z | u)$ and that the mechanism of $Z$ was previously trained or we have access to pre-trained network $Z$. Leverage this fact to train the networks of $X, Y, W$ by intervening on $U$ to minimize the loss $\mathcal{L}((X_u^F, Z_u^F, Y_u^F, W_u^F) || (X_u^R, Z_u^R, Y_u^R, W_u^R))$

## E.2    MODULAR TRAINING EXAMPLE 2

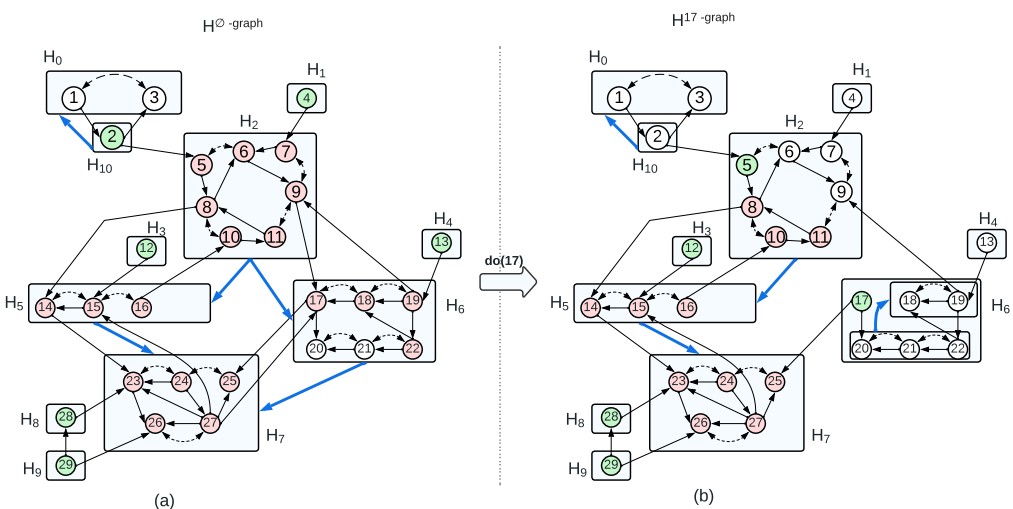

Figure 15: $\mathcal{H}^\emptyset$-graph and $H^{17}$-graph construction

In Figure 15, we construct the $\mathcal{H}^\emptyset$-graph as below. We describe the H-graph edges (thick blue edges) and the backdoor path (thin black edges) responsible for those edges.
$H_{10} \to H_0 : 3 \leftarrow 2 \leftarrow 1$
$H_2 \to H_5 : 14 \leftarrow 8 \leftarrow 11 \leftarrow 10 \leftarrow 16,$
$H_2 \to H_6 : 17 \leftarrow 9 \leftarrow 19,$
$H_5 \to H_7 : 23 \leftarrow 14 \leftarrow 15 \leftarrow 27,$
$H_6 \to H_7 : 25 \leftarrow 17 \leftarrow 27$

In Figure 15, we construct the $H^{17}$-graph as below:

$H_{10} \to H_0 : 3 \leftarrow 2 \leftarrow 1$

$H_2 \to H_5 : 14 \leftarrow 8 \leftarrow 11 \leftarrow 10 \leftarrow 16,$

$H_5 \to H_7 : 23 \leftarrow 14 \leftarrow 15 \leftarrow 27,$

Now, notice that due to $do(17)$, $H_6^{17}$ gets splitted into two new h-nodes, $[18, 19]$ and $[20, 21, 22]$ with a new edge $[20, 21, 22] \to [18, 19]$. However, according to our H-graph construction algorithm, we keep these two new h-nodes of $H^{17}$ combined inside $H_6^{17}$ same as $\mathcal{H}^{\emptyset}$-graph. Therefore, $\mathcal{H}^{\emptyset}$ and $H^{17}$'s common partial order does not change.

For training $H_7^{\emptyset}$ node: $\{23, 24, 25, 26, 27\}$, we match the following distribution found by applying do-calculus rule 2.

$$P(23, 24, 25, 26, 27, 14, 15, 16, 17, 18, 19, 22, 5, 6, 7, 8, 9, 10, 11, |do(2, 12, 13, 28, 29, 4)) \quad (54)$$

In Figure 15(a), joints are shown as red nodes and their parents as green nodes. However, consider, we have both observational and interventional datasets from $P(\mathcal{V})$ and $P(\mathcal{V}|do(17))$ and we have already trained all the ancestor h-nodes of $H_7^{17}$. Then we can train the mechanisms that lie in $H_7^{\emptyset}$ to learn both observational and interventional distribution by matching a smaller joint distribution compared to Equation 54:

$$P(23, 24, 25, 26, 27, 8, 10, 11, 14, 15, 16|do(12, 28, 29, 5, 17)) \quad (55)$$

In Figure 15(b), joints are shown as red nodes and their parents as green nodes. We see that the number of red nodes is less for $H^{17}$ graph compared to $\mathcal{H}^{\emptyset}$ graph when we were matching the mechanisms in h-node, $H_7^{\emptyset}$.

## F  EXPERIMENTAL ANALYSIS

In this section, we provide implementation details and algorithm procedures of our WhatIfGAN training.

### F.1  EXPERIMENT CHECKLIST

#### F.1.1  TRAINING DETAILS AND COMPUTE

We performed our experiments on a machine with an RTX-3090 GPU. The experiments took 1-4 hours to complete. We ran each experiment for 300 epochs. We repeated each experiment multiple times to observe the consistent behavior. Our datasets contained $20-40K$ samples, and the batch_size was 200, and we used the ADAM optimizer. For evaluation, we generated 20k fake samples after a few epochs and calculated the target distributions from these 20k fake samples and 20k real samples. We calculated TVD and KL distance between the real and the learned distributions. For Wassertein GAN with gradient penalty, we used LAMBDA_GP=10. We had learning_rate $= 5 * 1e - 4$. We used Gumbel-softmax with a temperature starting from 1 and decreasing it until $0.1$. We used different architectures for different experiments since each experiment dealt with different data types: low-dimensional discrete variables and images. Details are provided in the code. For low-dimensional variables, we used two layers with 256 units per layer and with BatchNorm and ReLU between each layer. Please check our code for architectures of other neural networks such as encoders and image generators

#### F.1.2  REPRODUCIBILITY

For reproducibility purposes, we share our implementation as supplementary materials. Besides, in the following sections, we provided explanations of each experiment along with model settings and hyper-parameters.

### F.2  IMAGE MEDIATOR EXPERIMENT

In this section, we provide additional information about the experiment described in Section 5.1. The front-door graph has been instrumental for a long time in the causal inference literature. However,

it was not shown before that modular training with high dimensional data was possible, even in the front door graph. This is why we demonstrate the utility of our work on this graph.

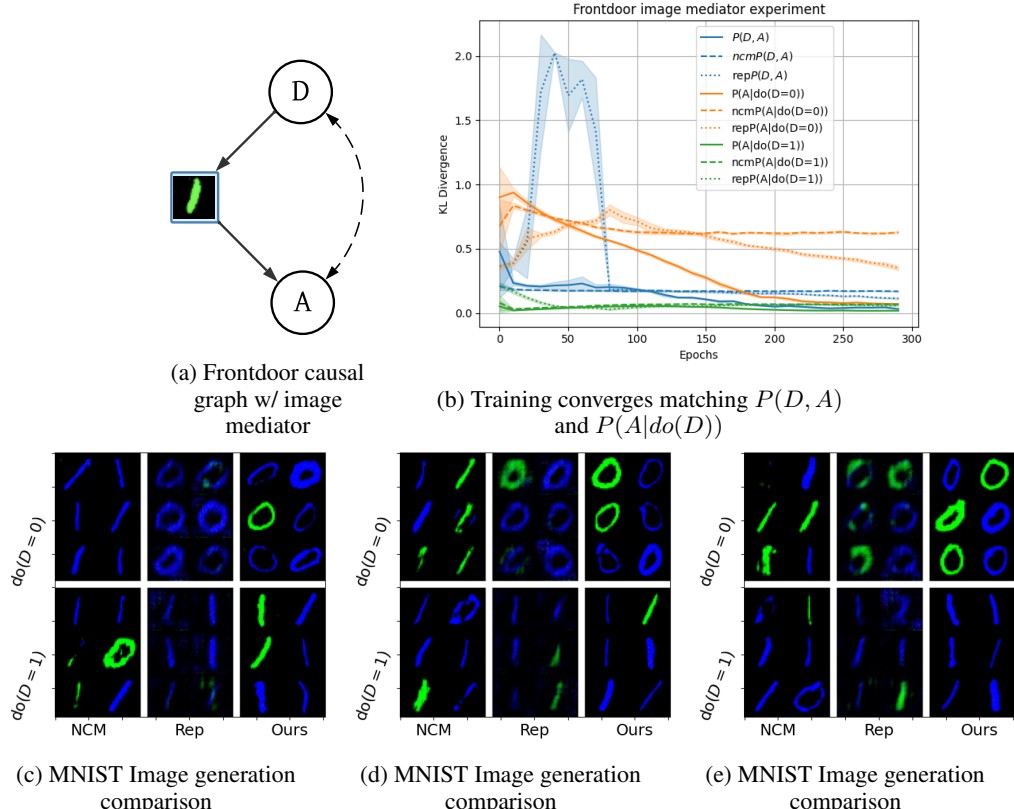

(a) Frontdoor causal graph w/ image mediator

(b) Training converges matching $P(D, A)$ and $P(A|do(D))$

(c) MNIST Image generation comparison

(d) MNIST Image generation comparison

(e) MNIST Image generation comparison

Figure 16: Modular Training on frontdoor causal graph with training order: $\{I\} \to \{D, A\}$

We have domain $D = [0, 1]$, Image size=$3 \times 32 \times 32$ and $C = [0, 1, 2]$. Let $U_0, e_1, e_2, e_3$ are randomly generate exogenous noise. $D = U_0 + e_1, Image = f_2(D, e_2), A = f_3(Image, e_3, U_0)$. $f_2$ is a function which takes $D$ and $e_2$ as input and produces different colored images showing $D$ digit in it. $f_3$ is a classifier with random weights that takes $U_0, e_3$ and $Image$ as input and produces $A$ such a way that $|P(A|do(D = 0)) - P(A|D = 0)|, |P(A|do(D = 1)) - P(A|do(D = 0))| and |P(A|D = 1) - P(A|D = 0)|$ is enough distant. The target is to make sure that the backdoor edge $D \leftrightarrow A$ and the causal path from $D$ to $A$ is active. Since we have access to $U_0$ as part of the ground truth, we can calculate the true value of $P(A|do(D))$ with the backdoor criterion:

$$P(A|do(D)) = \int_{U_0} P(A|D, U_0)P(D|U_0)$$

During training, $U_0$ is unobserved but still, the query is identifiable with the front door criterion. Image is a mediator here.

$$P(A|do(D)) = \int_{Image} P(Image|D) \sum_{D'} P(A|D', Image)P(D')$$

However, this inference is not possible with the identification algorithm. But WhatIfGAN can achieve that by producing $Image$ samples instead of learning the explicit distribution. If we can train all mechanisms in the WhatIfGAN DCM to match $P(D, A, I)$, we can produce correct samples from $P(A|do(D))$. We construct the WhatIfGAN architecture with a neural network $\mathbb{G}_D$ having fully connected layers to produce $D$, a deep convolution GAN $\mathbb{G}_I$ to generate images, and a classifier $\mathbb{G}_A$ to classify MNIST images into variable $A$ such that $D$ and $A$ are confounded. Now, for this graph, the corresponding $\mathcal{H}$-graph is $[I] \to [D, A]$. Thus, we first train $\mathbb{G}_I$ by matching $P(I|D)$. Next, to

train $\mathbb{G}_D$ and $\mathbb{G}_A$, we should match the joint distribution $P(D, A, I)$ since $\{I\}$ is ancestor set $\mathcal{A}$ for c-component $\{D, A\}$. GAN convergence becomes difficult using the joint distribution loss since the losses generated by low and high dimensional variables are not easily comparable and it is non-trivial to find a correct re-weighting of such different loss terms. To the best of our knowledge, no current causal effect estimation algorithm can address this problem since there is no estimator that does not contain explicit image distribution, which is practically impossible to estimate. To deal with this problem, we map samples of $I$ to a low-dimensional representation, $RI$ with a trained encoder and match $P(D, RI, A)$ instead of $P(D, Image, A)$.

Note that, we use the mechanism training order $[I] \rightarrow [D, A]$ specified by the H-graph (Algorithm 1) to match the joint distribution $P(D, Image, A)$. It is not feasible to follow any other sequential training order such as $[D] \rightarrow [Image] \rightarrow [A]$ as training them sequentially with individual losses can not hold the dependence in $D \leftrightarrow A$.

### F.3 Performance on Real-world COVIDx CXR-3 dataset

In this section, we provide some more results of our experiment on COVIDx CXR-3 dataset (Wang et al., 2020). This dataset contains 30,000 chest X-ray images with Covid ($C$) and pneumonia ($N$) labels from over 16,600 patients located in 51 countries. The X-ray images are of healthy patients ($C = 0, N = 0$), patients with non-Covid pneumonia ($C = 0, N = 1$), and patients with Covid pneumonia ($C = 1, N = 1$). X-ray images corresponding to COVID non-pneumonia ($C = 1, N = 0$) are not present in this dataset as according to health experts those images do not contain enough signal for pneumonia detection. However, to make the GAN training more smooth we replaced a few ($C = 1, N = 1$) real samples with ($C = 1, N = 0$) dummy samples. We also normalized the X-ray images before training. To obtain the low dimensional representation of both real and fake X-ray images, we used a Covid conditional trained encoder. Instead of training $\mathbb{G}_{Xray}$ from scratch, we use a pre-trained model (Giorgio Carbone, 2023) that can be utilized to produce Xray images corresponding to $C \in [0, 1]$ input. Note that, this pre-trained model takes value 0 for Covid, 1 for normal, and 2 for Pneumonia as input and produces the corresponding images. If a fake Covid sample indicates Covid=1, we map it to the 0 input of the pre-trained GAN. If a fake Covid sample indicates Covid=0, this might be either mapped to 1 (normal) or 2 (Pneumonia). Instead of randomly selecting the value, we use the real Pneumonia sample to decide this (either 1 or 2). After that, we produce X-ray images according to the decided input values. Since we are using the GAN-generated fake samples for Covid=1, the computational graph for auto grad is not broken. Rather the mentioned modification can be considered as a re-parameterization trick.

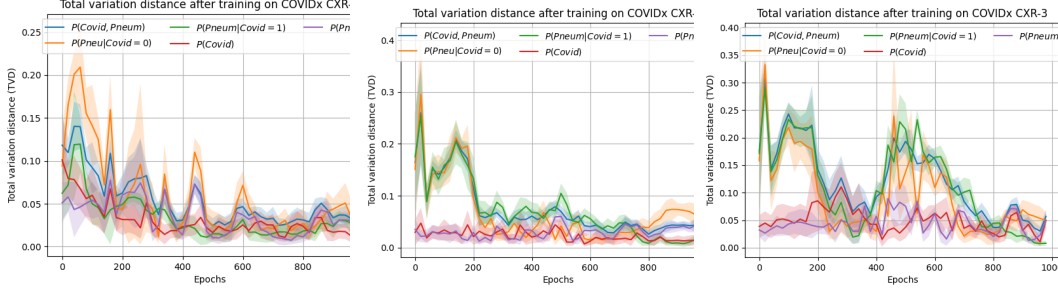

Figure 17: Total variation distance plots show WhatIfGAN converges on COVIDx CXR-3 dataset. (consecutive 20 epochs were averaged)

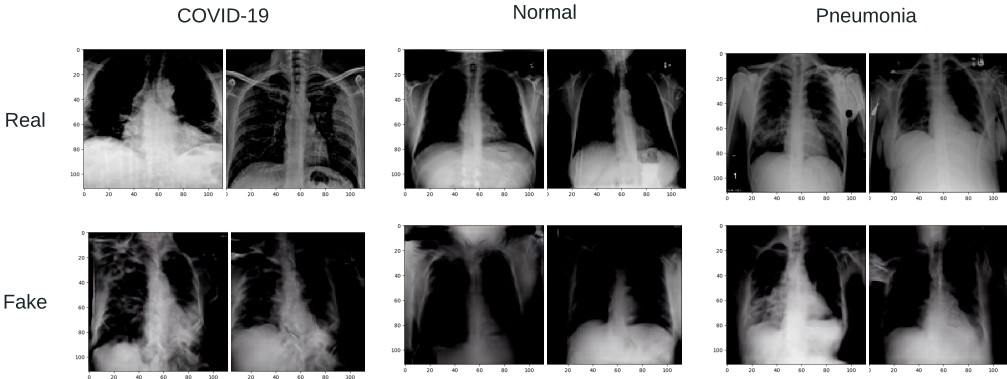

Figure 18: Real images from dataset vs pre-trained GAN generated images

### F.4 MNIST EXPERIMENT 2: $\mathcal{L}_2$ AND $\mathcal{L}_3$ AMPLING FROM MNIST DATASET

In this section, we provide additional information about the experiment Colored MNIST experiment.

#### F.4.1 MNIST EXPERIMENTAL SETUP

Here, we describe the synthetic SCM that was used for generating the dataset for this experiment.

$$p(U_1), p(U_2) \sim Dirichlet([0.1, 0.1, \ldots]) \qquad X_1 = U_1 + \epsilon_{X_1}$$
$$X_2 = U_2\%3 + X_1 * Uniform[3,6] + U_1 + \epsilon_{X_2} \qquad W = X_1 + X_2 + \epsilon_W$$
$$Y_1 = \lfloor W^2/10 \rfloor + \epsilon_{Y_1} \qquad Y_2 = W^2\%10 + \epsilon_{Y_2}$$
$$Color = U_2\%3 + \epsilon_c \qquad Thick = min(\lfloor (W^2/10)/2 \rfloor, 1) + \epsilon_t$$

We sample confounding variable $U_1, U_2$ and independent exogenous noises, $\epsilon_{X_i}, \forall i$ from Dirichlet distribution. We assign $X_1$ as a variable of binary uniform distribution in $[0, 1]$. $X_2 \in [0, 8]$, is correlated with $U_2$ and stays in $[0, 2]$ when $X_1 = 0$, and ranges in $[3, 8]$ when $X_1 = 1$. Also, $X_1$ and $X_2$ are confounded with $U_1$. Variable $W \in [0, 9]$ is the sum of $X_1$ and $X_2$. Next, $Y_1 \in [0, 9]$ is the first digit of $W^2$ and $Y_2 \in [0, 9]$ is the 2nd digit of $W^2$. $Color \in [0, 2]$ refers to [R=red, G=green, B=blue] respectively and is confounded with $X_2$ by $U_2$. $Thick \in [0, 1]$ referring to [th=thin, **Th**=thick] depends on the first digit of $W^2$. We introduce $25\%$ noise with all these relations to prevent them from becoming deterministic functions and also from having zero probabilities for any outcomes so that the strictly positive distribution assumption holds. Finally, we employ a procedure suggested in Castro et al. (2019) to produce $Image_1$ by using $Y_1$ as the digit parameter and produce $Image_2$ by using $Y_2$ as the digit parameter besides using $Color$ and $Thick$ as color and thickness parameters for both images. We consider we have access to one observational dataset, $D_1 \sim P(\mathcal{V})$ and two interventional datasets $D_2 \sim P_{X=0}(\mathcal{V}), D_3 \sim P_{X=1}(\mathcal{V})$.

#### F.4.2 MNIST LOW DIMENSIONAL VARIABLES

In Figure 19a, each blue box represent the h-nodes of $\mathcal{H}^\emptyset$. Since h-nodes of $Image_1, Image_2$ are disconnected from other h-nodes, we can first train the low-dimensional mechanisms and use them later to train the Image mechanisms. On the other hand, instead of training, we can also utilize pre-trained models for $Image_1, Image_2$. Without violating the training order, (step-1:) we can first train $[\mathbb{G}_{X_1}, \mathbb{G}_{X_2}, \mathbb{G}_W, \mathbb{G}_{Color}]$ together, with both $\mathcal{L}_1, \mathcal{L}_2$ datasets: $D_1, D_2$ and $D_3$. Next, (step-2:) we train $[\mathbb{G}_{Y_1}, \mathbb{G}_{Y_2}, \mathbb{G}_{Thick}]$ with only $\mathcal{L}_1$ dataset $D_1$. Since intervention $X_1$ is not in these h-nodes, according to Corollary C.5, WhatIfGAN can match both $P(Y_1, Y_2, Thick)$ and $P_{X_1}(Y_1, Y_2, Thick)$.

WhatIfGAN training convergence at step-1 and step-2 is shown in Figure 19b. For each distribution in $P(V)$ and $P_{X_1}(V)$, TVD from the true distribution is decreasing. Moreover, both $P(Y_1, Y_2, Thick)$ and $P(Y_1, Y_2, Thick|\text{do}(X_1))$ are converging even though we trained this mechanisms with only $\mathcal{L}_1$ dataset; as expected according to modular training. Furthermore, according to Theorem 4.2,

after training on $D_1, D_2$ and $D_3$, WhatIfGAN can produce correct samples from distributions that are identifiable from $P(\mathcal{V})$ and $P_{X_1}(\mathcal{V})$. In Figure 19d, WhatIfGAN illustrates this by producing counterfactual samples from $P_{x_1,x_2}(Color|x_1', x_2')$ with small TVD as it is identifiable from available distributions $P(\mathcal{V})$ (see derivation Appendix F.4.4 ). This experiment verifies that WhatIfGAN can produce interventional and counterfactual samples correctly.

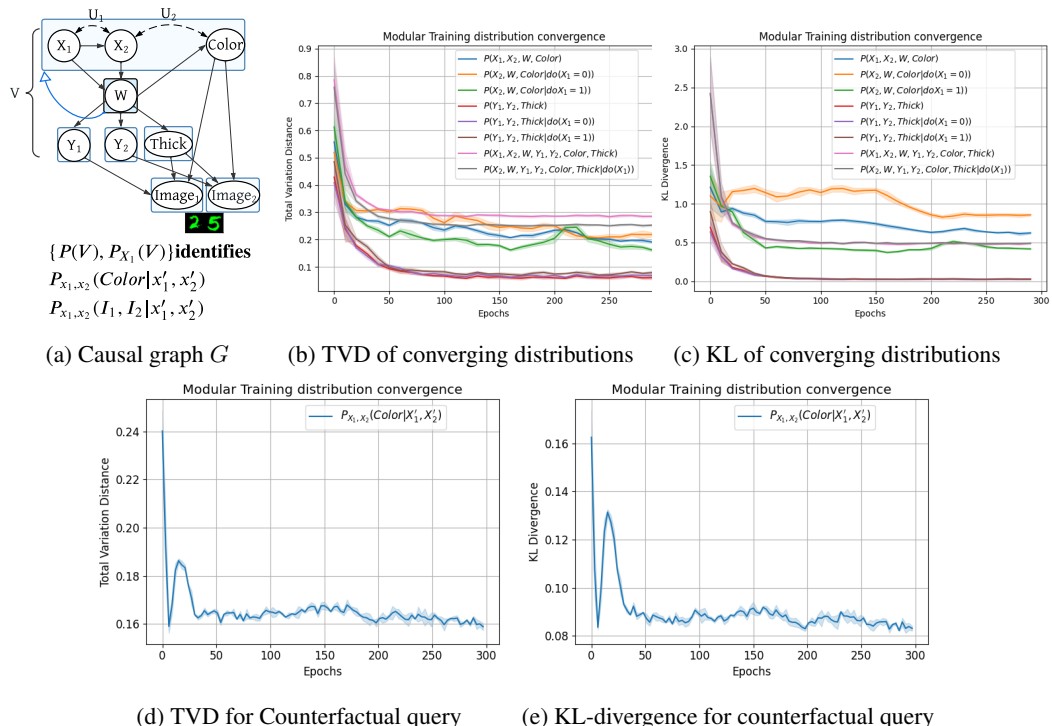

(a) Causal graph $G$  (b) TVD of converging distributions  (c) KL of converging distributions

(d) TVD for Counterfactual query  (e) KL-divergence for counterfactual query

Figure 19: Convergence of $\mathcal{L}_1$ and $\mathcal{L}_2$ distributions of MNIST-low dimensional variables.

### F.4.3  MNIST High dimensional variables

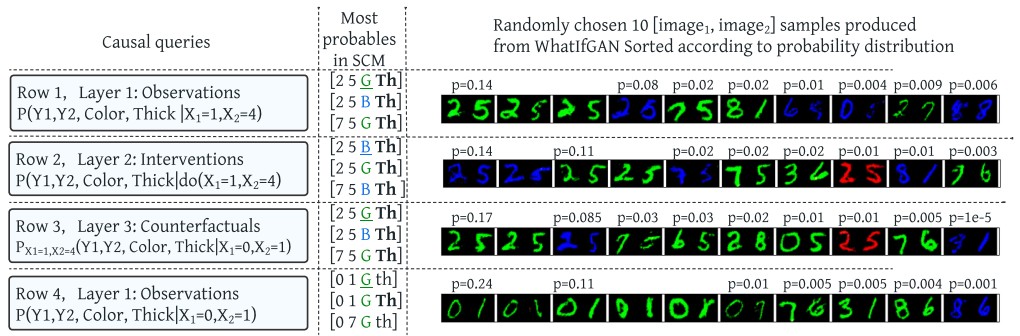

Figure 20: For each query, the top 3 occurred image samples match with true SCM

In this section, we describe the experiment in more detail. After training the mechanisms of $V' = \{X_1, X_2, W, Color, Y_1, Y_2, Thick\}$, we can train $Image_1, Image_2$ mechanisms. We can also utilize pre-trained models of $V'$ if available and use their output values as input to the image mechanisms. After training all the mechanisms in the causal model, we observed the results shown in Figure 20.

In row 1, for query $\in \mathcal{L}_1$ , we condition on $X_1 = 1, X_2 = 4$. Therefore, without considering noise, according to the true SCM, we get $W = 5$ and $Y_1 = 5^2/10 = 2, Y_2 = 5^2\%10 = 5$. $Color = X_2\%3 = 1$ due to the confounder between $X_2$ and $Color$. $Thick = \lfloor (5^2/10)/2 \rceil = 1$. This represents the highest probable outcome 2,5 with green color and thick=1 which is shown as the first outcome [2,5, G, **Th**] (column-2). We observe that, our trained gan also produces images of the same outcome with the highest probability, $p = 0.14$. In row 2, for query $\in \mathcal{L}_2$, we intervene on $X_1 = 1, X_2 = 4$. So, $X_2$ and $Color$ are not confounded anymore and $Color$ will have other outcomes besides green with high probability for example, blue. And as expected, our trained GAN also produces images with blue color with the highest probability.

In row 3, for query $\in \mathcal{L}_3$ we intervene on $X_1 = 1, X_2 = 4$ and $X_2$ does not have any confounding effect on $Color$. However, since we condition on $X_1 = 0, X_1 = 1$, we would have the corresponding confounder $U_2$ which will assign $Color = X_1\%3$,i.e, green. Due to this, the most probable counterfactual outcome according to true SCM is [2 5 G **Th**]. We also find the same outcome with a high probability from our GAN samples. This indicates the ability of WhatIfGAN to correctly produce counterfactual samples and differentiate between $\mathcal{L}_2$ and $\mathcal{L}_3$ queries. Finally, in the 4th row, we convert the previous $\mathcal{L}_3$ query into a $\mathcal{L}_1$ query, by removing the intervention but keeping the same conditions i.e., $X_1 = 0, X_2 = 1$. Our GAN produces green, thin digits of 0,1 with the highest occurrences which is also suggested by the true SCM. This is evidence of the capability of WhatIfGAN to separate different causal layers and produce corresponding samples from each layer.

### F.4.4 MNIST DERIVATION

We follow Shpitser & Pearl (2008) to obtain the counterfactual graph and the interventional expression of the counterfactual query for the causal graph shown in Figure 21. For convenience, we combine the variables $\{Y_1, Y_2, Thick, Image_1, Image_2\}$ into variable $Y$ and will consider the counterfactual query $P(Y_{x_1,x_2} = y|x'_1, x'_2)$.

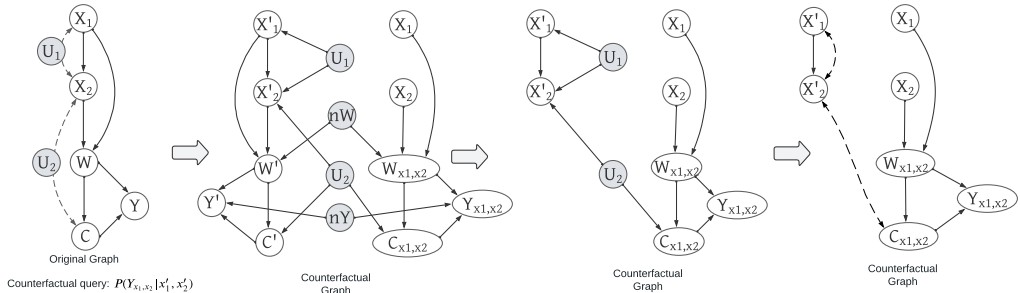

Figure 21: Counterfactual graph construction

In the counterfactual graph, there are three c-components (without fixed nodes: $\{X_1, X_2\}$). They are: $\{X'_1, X'_2, C\}, \{W\}$ and $\{Y\}$ (omitting the subscripts). Therefore, the expression is,

$$P(Y_{x_1,x_2} = y|x'_1, x'_2) = \frac{\sum_{w,c} P_w(x'_1, c, x'_2) * P_{x_1,x_2}(w) * P_{w,c}(y)}{\sum_{w,c,y} P_w(x'_1, c, x'_2) P_{x_1,x_2}(w) * P_{w,c}(y)} \tag{56}$$

Using do-calculus rule 2, we can write $P_{x_1,x_2}(w) = P(w|x_1, x_2)$ and $P_{w,c}(y) = P(y|w, c)$. Next the first term,

$$\begin{aligned}
P_w(x'_1, c, x'_2) &= P_w(x'_1) * P_w(x'_2|x'_1) * P_w(c|x'_1, x'_2) &\text{Factorization formula}\\
&= P(x'_1) * P(x'_2|x'_1) * P_w(c|x'_1, x'_2) &\text{do-calculus rule 3} &\tag{57}\\
&= P(x'_1) * P(x'_2|x'_1) * P(c|x'_1, x'_2, w) &\text{do-calculus rule 2}
\end{aligned}$$

So, finally we can write,

$$P(Y_{x_1,x_2} = y|x'_1, x'_2) = \frac{\sum_{w,c} P(x'_1) * P(x'_2|x'_1) * P(c|x'_1, x'_2, w) * P(w|x_1, x_2) * P(y|w, c)}{\sum_{w,c,y} P(x'_1) * P(x'_2|x'_1) * P(c|x'_1, x'_2, w) * P(w|x_1, x_2) * P(y|w, c)} \tag{58}$$

Therefore, if we can train WhatIfGAN with datasets from only $P(V)$ then we can produce samples from this counterfactual query.

In Shpitser & Pearl (2009), the authors claimed that the counterfactual query $P(Y_{x_1, x_2} = y | x'_1, x'_2)$ is non-identifiable only from $P(V)$ due to the $X_1 \leftarrow U_1 \rightarrow X_2$ edge. After having a conversation with the authors we came to know that the claim was not correct. The counterfactual query is indeed identifiable from $P(V)$ as shown above.

### F.5    ASIA/LUNG CANCER DATASET

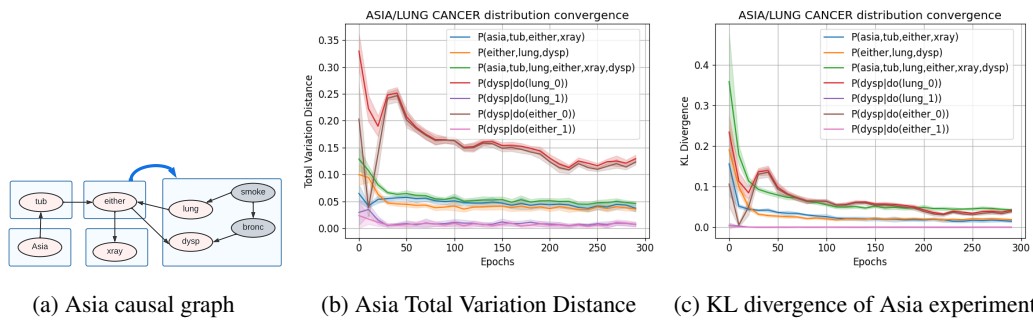

(a) Asia causal graph      (b) Asia Total Variation Distance      (c) KL divergence of Asia experiment

Figure 22: Modular Training on Asia Dataset

***Asia Dataset.*** We evaluate our algorithm performance on ASIA Dataset from bnlearn repository (Scutari & Denis, 2021). The purpose of this experiment is to show that modular training can learn the joint distribution of the Asia dataset formed as semi-Markovian and correctly produces samples from identifiable $\mathcal{L}_2$ distributions. To check the effectiveness of WhatIfGAN for a semi-Markovian causal model, we hide "smoke" and "bronc" variables in the observational dataset as shown in Figure 22a. This action gives us a causal graph with a latent confounder between the "lung" and the "dysp" variables. The $\mathcal{H}$ graph nodes are indicated by the square box containing the variables. According to the algorithm, all $\mathcal{H}$-nodes are disconnected except $[either] \rightarrow [lung, dysp]$. Therefore, we first start training the mechanisms of $asia, tub, either, xray$ and then separately but in parallel train the mechanisms of $lung, dysp$. Here we can also use pre-trained $either$ while we train $lung, dysp$ to match the distribution $P(lung, dysp, either | tub)$. For evaluation, we generated samples from $P(dysp|\text{do}(lung))$ and $P(dysp|\text{do}(either))$ distributions from WhatIfGAN. We can calculated $P(dysp|\text{do}(lung))$ with front-door adjustment and $P(dysp|\text{do}(either))$ with back-door adjustment using the real dataset samples. In Figure 22b, 22c, we can see that our partial training is working well with all of the distributions converging to low TVD and KL loss.

### F.6    REAL-WORLD: SACHS PROTEIN DATASET

For completeness, we test both WhatIfGAN and NCM(Xia et al., 2023) performance on a low-dimensional real-world Sachs dataset (Sachs et al., 2005), which contains a protein signaling causal graph and is given in Figure 23a. The goal is to illustrate WhatIfGAN's capability of utilizing multiple partial $\mathcal{L}_1, \mathcal{L}_2$ datasets. We considered the observational dataset $D_1 \sim P(PKA, Mek, Erk, Akt)$ and the interventional dataset $D_2 \sim P(Mek|\text{do}(PKA = 2))$. The interventional dataset with $PKA = 2$ is chosen since it has a large number of samples. Here we intentionally hide variable $PKC$ and considered it as a confounder. Hence, $P(Mek|\text{do}(PKA = 2))$, $P(Akt|\text{do}(PKA = 2)$ and $P(Erk|\text{do}(PKA = 2))$ are non-identifiable from only $P(V)$. According to Corollary C.5, $P(V), P(Mek|\text{do}(PKA = 2))$ make these distributions identifiable. More precisely, if we have access to $P(V)$ and $P_{PKA}(Mek)$ only, then its sufficient to identify, $P_{PKA}(Mek, Erk, Akt)$. We have datasets $D_1, D_2$ that are sampled from $P(V), P(Mek|\text{do}(PKA = 2))$ for training. If we convert the Sachs graph into the train graph $\mathcal{H}$ in Figure 23a, we see that for only the interventional dataset, we have to train the mechanism of Mek. This is because other variables except Mek belong

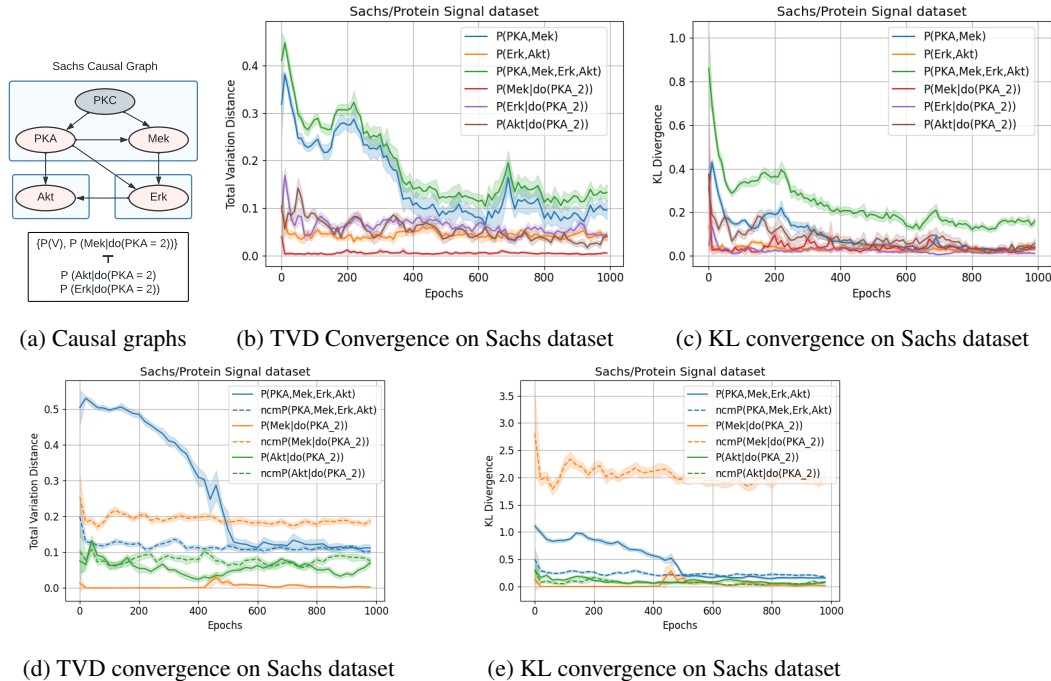

Figure 23: Benchmark and Real-world datasets

(a) Causal graphs     (b) TVD Convergence on Sachs dataset     (c) KL convergence on Sachs dataset

(d) TVD convergence on Sachs dataset     (e) KL convergence on Sachs dataset

to different hnodes or training components. Here,

$$
\begin{aligned}
P_{PKA}&(Mek, Erk, Akt) \\
&= P_{PKA}(Mek)P_{PKA}(Erk|Mek)P_{PKA}(Akt|Erk, Mek) \\
&= P_{PKA}(Mek)P(Erk|Mek, PKA)P(Akt|PKA, Erk, Mek)
\end{aligned}
\tag{59}
$$

Therefore, we train WhatIfGAN i.e., the DCM to match both $P(V)$ and $P(Mek|\text{do}(PKA))$. We *i)* first train $\mathbb{G}_{H_0} = [\mathbb{G}_{PKA}, \mathbb{G}_{Mek}]$ with both $D_1$ and $D_2$, *ii)* next, train $\mathbb{G}_{H_1 \cup H_2} = [\mathbb{G}_{Erk}, \mathbb{G}_{Akt}]$ with only $D_1$. In Figure 23b and 23c, WhatIfGAN converges by training on both $P(V)$ and $P(Mek|\text{do}(PKA = 2))$ datasets. We compared the distributions $P(Akt|\text{do}(PKA = 2)$ and $P(Erk|\text{do}(PKA = 2))$ implicit in WhatIfGAN generated samples with the Sachs $\mathcal{L}_2$-dataset distributions and observed them matching with low TVD and KL loss. Even though, we dont observe $Erk$ and $Akt$ in $D_2$, with modular training, we can still train the mechanisms with $P(V)$ and sample correctly from their $\mathcal{L}_2$- distributions. This reflects the transportability of WhatIfGAN. During WhatIfGAN training, we can use pre-trained models of $\{PKA, Mek\}, \{Akt\}$ or $\{Erk\}$ since they are located in different hnodes.

***Sachs dataset performance comparison with (Xia et al., 2023):*** In Figure 23d and Figure 23e, we compare and show the convergence of both WhatIfGAN and NCM (Xia et al., 2023) with respect to total variation distance and KL-divergence. We observe that for low-dimensional variables, we perform similarly to DCM or better in some cases. However, they do not have the ability to utilize pre-trained models like we do. Besides, unlike NCM, we do not need to run the algorithm again and again for each identifiable queries. Thus, when queries are identifiable, our algorithm can be utilized as an efficient method to train on datasets involving low-dimensional variables.

### F.7 SYNTHETIC EXPERIMENT

We measure the performance of the approach where we train all the mechanisms together (Algorithm 3) and WhatIfGAN Modular Training (Algorithm 2) on synthetic datasets produces from a random SCM consistent with the causal graph in Figure 24. We produce two observational datasets from the true causal model: $D_1 \sim P(V \setminus \{W_0, Y_0\}), D_2 \sim P(V \setminus \{Y_1\})$. We train the causal mechanism of our DCM by following the $\mathcal{H}^\emptyset$-graph in Figure 24. Here, $H_0 = \{W_0\}, H_1 = \{W_1\}, H_2 = \{X_0, Y_0\}, H_3 = \{X_1, X_2, Y_1\}$. Let $p()$ be the (observational) data distribution. We perform modular training following the $\mathcal{H}^\emptyset$-graph as following: *i)* Train $\mathbb{G}_{H_0} = [\mathbb{G}_{W_0}]$ with dataset $D_2$ by matching

$p(w_0|x_0)$. $ii)$ Train $\mathbb{G}_{H_1} = [\mathbb{G}_{W_1}]$ on $D_1$ to match $p(w_1|x_1, x_2, x_0)$. $iii)$ Train $\mathbb{G}_{H_2} = [\mathbb{G}_{X_0}, \mathbb{G}_{Y_0}]$ with dataset $D_2$ by comparing $p_1(x_0, w_1, w_0, y_0|\text{do}(x_1, x_2))$. Here, we can use pre-trained $W_0, W_1$. $iv)$ Finally we train the mechanisms in $\mathbb{G}_{H_3} = [\mathbb{G}_{X_1}, \mathbb{G}_{X_2}, \mathbb{G}_{Y_1}]$ together, with dataset $D_1$ such that after convergence, $p_1(x_1, x_2, w_1, y_1|\text{do}(x_0))$ matches. Here, we can use pre-trained $W_1$.

We show the experimental result in Figure 24. Here, $P_1()$ represents the sample distributions from WhatIfGAN after running Algorithm 3 (i.e. training all the mechanisms together) and $P_2()$ represents the distributions after running Algorithm 2, the Modular Training. We trained each $\mathcal{H}$-nodes in parallel for 150 epochs. Line $4, 5$ represents the distribution convergence corresponding to $H_4, H_5$ in terms of the total variation distance from the true distribution. Line 6 shows the convergence of joint distribution $P_2(V)$ with Modular Training. Line $1, 2, 3$ represent the same distributions but with Algorithm 3. We observe that each partial sample distributions converges with very small total variation distance loss:$< 0.10$ and joint distribution for modular training also converges with low TVD loss: $< 0.16$. For both algorithms, the TVD loss is similarly small, proving the validity of modular training of WhatIfGAN.

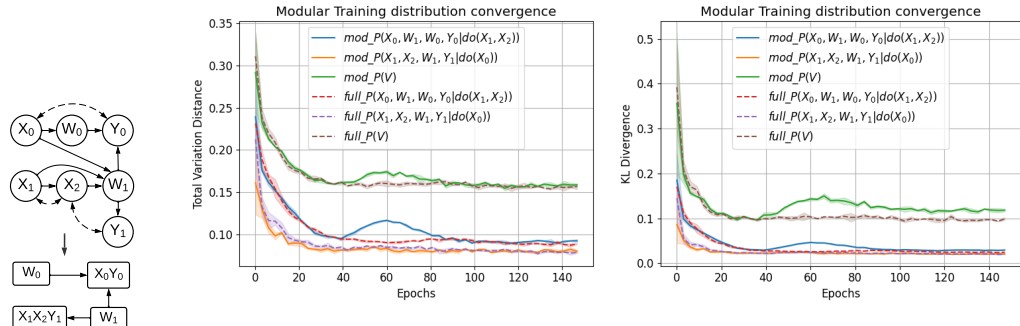

Figure 24: Convergence of full training and modular training on observational datasets.

Training each h-node matches the following distributions of $D_1$ and $D_2$:

$$
\begin{aligned}
H_0 &\implies p(w_0|x_0) = q_{x_0}(w_0) \\
H_1 &\implies p(w_1|x_0, x_1, x_2) = q_{x_0, x_1, x_2}(w_1) \\
H_2 &\implies p(x_0, w_0, y_0, w_1|x_1, x_2) = q_{x_1, x_2}(x_0, w_0, y_0, w_1) \\
H_3 &\implies p(x_1, x_2, w_1, y_1|x_0) = p_{x_0}(x_1, x_2, w_1, y_1)
\end{aligned}
\tag{60}
$$

Therefore, matching each distribution, we match the joint distribution.

$$
\begin{aligned}
p(v) &= p_{w_1}(x_1, x_2, y_1) * p_{w_0, w_1}(x_0, y_0) * p_{x_0}(w_0) * p_{x_0, x_1, x_2}(w_1) \\
&= p_{x_0, w_1}(x_1, x_2, y_1) * p_{x_1, x_2, w_0, w_1}(x_0, y_0) \\
&\quad * p_{x_0}(w_0) * p_{x_0, x_1, x_2}(w_1) \\
&= \frac{p_{x_0}(x_1, x_2, w_1, y_1)}{p_{x_0, x_1, x_2}(w_1)} * \frac{p_{x_1, x_2}(x_0, w_0, y_0, w_1)}{p_{x_0, x_1, x_2}(w_0) * p_{x_0, x_1, x_2}(w_1)} \\
&\quad * p_{x_0}(w_0) * p_{x_0, x_1, x_2}(w_1) \\
&= \frac{p(x_1, x_2, w_1, y_1|x_0)}{p(w_1|x_0, x_1, x_2)} * \frac{p(x_0, w_0, y_0, w_1|x_1, x_2)}{p(w_0|x_0, x_1, x_2) * p(w_1|x_0, x_1, x_2)} \\
&\quad * p(w_0|x_0) * p(w_1|x_0, x_1, x_2) \\
&= \frac{q(x_1, x_2, w_1, y_1|x_0)}{q(w_1|x_0, x_1, x_2)} * \frac{q(x_0, w_0, y_0, w_1|x_1, x_2)}{q(w_0|x_0, x_1, x_2) * q(w_1|x_0, x_1, x_2)} \\
&\quad * q(w_0|x_0) * q(w_1|x_0, x_1, x_2) \\
&= q(v)
\end{aligned}
\tag{61}
$$

# G   ALGORITHMS & PSEUDO-CODES

---

**Algorithm 8** isIdentifiable($G, \mathcal{I}, query$)

---

1: **Input:** Causal Graph $G = (\mathcal{V}, \mathcal{E})$, Interventions = $I$, Causal query distribution= $query$
2: **if** type($query$)=Counterfactual **then**
3:     **Return** Run_IDC($G, query, I$)
4: **else if** type($query$)=Interventional **then**
5:     **Return** Run_ID($G, query$) or hasSurrogates($G, query, I$)

---

---

**Algorithm 9** RunGAN($G, \mathbb{G}, V_{\mathbf{K}}, I, N$)

---

1: **Input:**Causal Graph $G = (\mathcal{V}, \mathcal{E})$, DCM $\mathbb{G}$, target variable set $V_{\mathbf{K}}$, Intervention $I$, Pre-defined noise $N$.
2: **for** $V_i, V_j \in V_{\mathbf{K}}$ such that $i < j$ **do**
3:     **if** $V_i, V_j$ has latent confounder **then**
4:         $z \sim p(z)$
5:         $conf[V_i] \leftarrow Append(conf[V_i], z)$
6:         $conf[V_j] \leftarrow Append(conf[V_j], z)$      // Assigning same confounding noise [fix for multiple confounders]
7: **for** $V_i \in V_{\mathbf{K}}$ in causal graph,$G$ topological order **do**
8:     **if** $V_i \in I.keys()$ **then**
9:         $v_i = I[V_i]$      // Assigning intervened value
10:    **else**
11:        $par = get\_parents(V_i, G)$
12:        **if** $V_i \in N.keys()$ **then**
13:            $exos, conf, gumbel = N[V_i]$
14:        **else**
15:            $exos \sim p(z)$
16:            $conf = conf[V_i]$
17:            $gumbel = \emptyset$.     //New Gumbel noise will be assigned during forward pass
18:        $v_i = \mathbb{G}_{\theta_i}(exos, conf, gumbel, \hat{\mathbf{v}}_{par})$
19:    $\hat{\mathbf{v}} \leftarrow Append(\hat{\mathbf{v}}, v_i)$
20: **Return** Samples $\mathbf{v}$ or Fail

---

---

**Algorithm 10** Evaulate_GAN($G, \mathbb{G}, \mathcal{I}, query$)

---

1: **Input:**Causal Graph $G = (\mathcal{V}, \mathcal{E})$, DCM= $\mathbb{G}$, Available Interventions = $\mathcal{I}$, Causal query distribution=$query$
2: **if** isIdentifiable($G, \mathcal{I}, query$) = False **then**
3:     **Return:** Fail
4: **if** type(query)= observation **then**
5:     $Y = Extract(query)$
6:     $samples \leftarrow$ RunGAN($G, \mathbb{G}, [Y], \emptyset, \emptyset$)
7: **else if** type(query)= Intervention **then**
8:     $Y, (X, x) :=$ Extract($query$)
9:     $samples \leftarrow$ RunGAN($G, \mathbb{G}, [Y], \{X : x\}, \emptyset$)
10: **else if** type(query)= Counterfactual **then**
11:     $Y, (X, x), (X, x') :=$ Extract($query$)
12:     $exos, conf, gumbel \leftarrow$ RejectionSampling($\{X : x'\}$)
13:     $N \leftarrow [exos, conf, gumbel]$
14:     $samples \leftarrow$ RunGAN($G, [Y], \{X : x\}, N$)
15: **Return** $samples$

---

# H   FREQUENTLY ASKED QUESTIONS

In this section, we answer some questions that might come into the reader's mind.

1. **How does our algorithm work for complicated and larger graphs?**
   **Ans:** In Appendix E, we use two graphs with 6 nodes and 27 nodes to show how our

algorithm works. We also show our algorithm performance on different graphs in Appendix F. Specifically, in Appendix F.4, we show our performance on a Colored-MNIST experiment where the causal graph contains 7 discrete variable nodes and 2 image nodes. In Appendix F.5, we perform an experiment on the ASIA causal graph from the online bnlearn repository. In Appendix F.6, we show our algorithm performance on the real-world Sachs protein dataset. In Appendix F.7, we show performance on a synthetic experiment with a 7 node causal graph.

2. **What is the benefit of a pre-trained model over training a conditional generator on the observed dataset? Is there a risk that the distribution the pre-trained model was trained on will not match the distribution of the observed dataset?**

   **Ans:** Training a conditional GAN from scratch might require abundant data and resources but if pre-trained, we can use them as black boxes. This gives us the scope to utilize large image-generative models in causality. For example, for the graph $D \rightarrow Image \rightarrow A$, $D \leftrightarrow A$ in Figure 3a, we could try to match the joint distribution $P(D, Image, A)$ by training all mechanisms together with the same loss from scratch. In that case: i) we could not use any pre-trained models ii) it would not be easy to get convergence due to the contribution of both low and high dimensional variables in the same loss function.

   We assume that a pre-trained model correctly captures a part of the joint distribution of which other parts are captured when we train the rest of the network. Specifically, in the $D \rightarrow Image \rightarrow A$, $D \leftrightarrow A$ graph, we assume the black box $\mathbb{G}_{Image}$ is correctly trained to sample image from the inputs same as $P(Image|D)$. Thus, we can train mechanisms $\mathbb{G}_D, \mathbb{G}_A$ to match $P(D, Image, A)$.

3. **How are the pre-trained generative models playing a role in our algorithm?**
   **Ans:** We follow the partial order of $\mathcal{H}-$graph to train our DCM in a modular fashion. While we are training the mechanisms in an h-node, we always use the pre-trained models of the ancestor variables. This gives us the flexibility to plug in any pre-trained large models in our training mechanism instead of training them from scratch.

4. **Can a better architecture/model be used instead of GANs for our framework?**
   **Ans:** Our modular training routine is not specific to GANs. Essentially, if there is a way to train a conditional generative model such as a diffusion model, our results should apply.

5. **How does the GAN convergence impact the theory of our work?**
   **Ans:** We proposed our theorems with the assumption that GAN will properly converge and our algorithm will give correct results. However, due to different issues with GAN training such as mode collapse or convergence failure, it is not unnatural to not converge. As a result, some errors will be introduced in the sampling process. Nonetheless, our algorithm is independent of the method used to train each module and GAN convergence can not break any of our theorems. Thus, one can use state-of-the-art generative models such as style-gan or diffusion models for training each module to improve the GAN training issues.

6. **Can our algorithm intervene with continuous values?**
   **Ans:** Do-calculus does not require discrete variables under certain regularity conditions. One can easily see this from the proofs of the rules from Pearl (1995). Since the Identification algorithm Shpitser & Pearl (2008) and derivatives can be seen as sequential applications of these rules, our proofs follow identically irrespective of the data type. For example, in the experiment described in section 5.1, the generated images are continuous vectors of pixels. Here we can sample from $P(C|do(D)), P(Image|do(D))$, and $P(C|do(Image))$.

