# OpenReview forum: "Modular Learning of Deep Causal Generative Models for High-dimensional Causal Inference"
_ICLR.cc/2024/Conference — Submitted to ICLR 2024_

### Official Review · Reviewer_LB1R · 2023-10-23

**Soundness:** 3 good
**Presentation:** 3 good
**Contribution:** 3 good
**Rating:** 6
**Confidence:** 3

**Summary:**

The paper proposes a method to modularly train deep generative models to perform causal inference in high dimensional data

**Strengths:**

- The paper is fairly well written and tackles and interesting problem
- To the best of my knowledge this is a fairly novel approach on training Neural Nets for causal inference
- The maths appear to be sound
- The results and evaluation are acceptable

**Weaknesses:**

- There is a large number of related literature that is missing. There have been too many to enumerate here methods that would be considered related work and should be compared against
- The Covid chest x-ray dataset is not of high quality , the reviewer would suggest using a more established test dataset.
- There are multiple typos (trianing -> training) and articles missing
- The method section could be made a bit better to increase clarity

**Questions:**

- GANs are notoriously messy and hard to train , plus they can only approximate the data distributions in question. Why is this the chosen backbone and not something else like normalizing flows ?


Overall i think this is a good paper, that would benefit the community

---

> ### Author Response · Authors · 2023-11-17
> **Response to Reviewer LB1R (1/1)**
>
> We are happy that the reviewer found our paper well-written and our work novel.
> Below we address the reviewer’s concerns.
>
> ## Missing literatures:
>
> > "There is a large number of related literature that is missing"
>
> We apologize if we missed some related works that solve high-dimensional interventional and counterfactual sampling
> problems. We intended to include all the recent works that solve similar problems with deep learning models such as
> GANs, normalizing flow, and diffusion models. We would be happy to include the ones that we missed if the reviewer would
> kindly point us to them.
>
> For the reviewer’s convenience, here is a list of some recent papers we cited and qualitatively compared our approach
> against:
>
> * Balazadeh et al (2022). Partial identification of treatment effects with implicit generative models.
> * Louizos et al (2017). Causal effect inference with deep latent-variable models.
> * Kocaoglu et al (2018). Causalgan: Learning causal implicit generative models with adversarial training.
> * Pawlowski et al (2020). Deep structural causal models for tractable counterfactual inference.
> * Xia et al 2023. Neural causal models for counterfactual identification and estimation.
> * Nemirovsky et al (2020). Countergan: generating realistic counterfactuals with residual generative adversarial nets
> * Xia et al 2021. The causal-neural connection: Expressiveness, learnability, and inference.
> * Bica et al (2020). Estimating the effects of continuous-valued interventions using generative adversarial networks.
>
> ## Datasets:
>
> > "The reviewer would suggest using a more established test dataset"
>
> We believe that the performed experiments in our paper illustrate our motivation, contribution, and technical novelty.
> However, we would greatly appreciate it if the reviewer could point us to other real-world datasets that they believe is
> suitable to evaluate our method for high-dimensional causal inference. We would happily include them in our experiments.
>
> For the reviewer’s convenience, here we provide the set of experiments we discussed in the main paper and in the
> appendix.
>
> * In Section 5.1, we highlight our flexibility to utilize pre-trained models in a front door graph setting with the
>   colored MNIST dataset.
> * In Section 5.2, we experiment on the real-world COVIDx CXR-3 dataset with 30k samples that we associate with a front
>   door graph.
> * In Appendix F.4, we showed our performance on a Colored-MNIST experiment where the causal graph contains 7 discrete
>   variables and 2 image variables.
> * In Appendix F.5, we experiment with the ASIA causal graph (6 nodes) from the popular bnlearn repository (
>   bnlearn.com/bnrepository).
> * In Appendix F.6, we show our algorithm’s performance on the real-world Sachs protein dataset from the bnlearn
>   repository which includes both observational and interventional data.
> * In Appendix F.7, we show performance on a synthetic experiment with a causal graph with 7 nodes.
>
> ## Chosen neural network architecture:
>
> > "Why is GAN the chosen backbone?"
>
> Even though we learn the desired distribution with GAN training, our modular training routine is not specific to GANs as
> we mentioned in Appendix H. Essentially, if there is a way to train multiple conditional generative models with the same
> noise, our results should apply. For example, conditional VAE can be used instead of GANs since its decoder is not
> different from the generator of a GAN in terms of sampling. It just requires a different way of training.
>
>
> > "Why not something like normalizing flows"
>
> This is an interesting question. For a causal graph without any latent: $X \rightarrow Z \rightarrow Y$, we can use
> normalizing flow to get exogenous noise $n_X$, $n_Z$, and $n_Y$ by learning some invertible functions. However, suppose,
> there exists an unobserved confounder $U$ between $X$ and $Y$. The causal graph is $X \rightarrow Z \rightarrow Y; X
> \leftarrow U \rightarrow Y$. Now, obtaining $U$ from $X$ and $Y$ and getting back $X$ and $Y$ from $U$ might not be an
> invertible function. Thus, normalizing flow may not be directly applied. Although Pawlowski et al. (2020) proposed a
> normalizing flow-based method for interventional and counterfactual sampling, the solution was for causal graphs with no
> unobserved variables. In GAN or conditional VAE, this issue is dealt with by initializing some random Gaussian noise to
> represent U and feeding the same noise to multiple networks $X$ and $Y$ while training. As a result, when $X$ and $Y$
> are trained to learn their mechanisms their output will be correlated since they use the same initial random noise. We
> will add this discussion on the challenges in using different generative models in the presence of unobserved
> confounders to a discussion section in the camera-ready.
>
> We again thank the reviewer again for their constructive feedback. We hope we addressed all their concerns. We are looking forward to answering any other questions the reviewer might have.

---

### Official Review · Reviewer_g68Y · 2023-10-31

**Soundness:** 3 good
**Presentation:** 2 fair
**Contribution:** 2 fair
**Rating:** 5
**Confidence:** 4

**Summary:**

In this paper, the authors proposed a new modular training algorithm to learn deep causal generative models, given a known causal graph and observational dataset. The method proceeds via factorizing the full joint distribution into its different c-components in the form of $\mathcal{L}_2$ distributions, and then match each components sequentially leveraging rule 2 of do-calculus. In this way, the authors suggest that in certain cases, pre-trained model might be utilized to improve convergence of the learning process. For example, when certain c-component is a high-dimensional node like images, one can simply plugin a pre-trained generative model. Finally, this enables the identification of interventional or counterfactual queries in the presence of high-dimensional variables.

**Strengths:**

- This paper is built upon well-known results in causality and proposed a modular training method for deep causal models, which, to the best of my knowledge, is a new contribution to the field. This might have potential use cases in learning realistic counterfactual data generation.
- For certain cases demonstrated in the paper, the proposed method achieved good performance in terms of faster convergence and better identification quality.

**Weaknesses:**

My primary reservations regarding this work stem from the possibility that the good performance demonstrated in a few hand-crafted scenarios may not extend to more general cases involving high dimensional variables.

- More specifically, in all the experiments presented in the paper, the underlying causal graph fortuitously satisfies the condition wherein the solitary high-dimensional nodes are isolated c-components. However, in more general cases, such as the example depicted in Figure 2, the method would still need to match $p(Z_1, Z_2,Z_3|X_1)$ and $p(X_1, X_2, Z_1, Z_3)$, which does not significantly simplify the original problem of matching $p(X_1, X_2, Z_1, Z_2, Z_3)$, especially considering the added complexity of the proposed algorithm.

- Furthermore, even in the scenarios where the proposed method was successful (experiments 1 and 2), the authors neglected to provide a comparison with an evident baseline: initializing the neural net architecture of the corresponding image node using a pre-trained marginal distribution, followed by fine-tuning the joint model in an NCM fashion. This baseline could serve as a robust ablation study to justify the necessity of c-component decomposition, a critical aspect that is currently absent from the current paper.

- Lastly, the paper's problem setting requires access to the true causal graph, which is a reasonable assumption. However, in many scenarios where high-dimensional deep causal models are required, the true causal graph may not be known. This limitation could potentially restrict the applicability of the proposed method. In contrast, the joint training approach can easily adapt to unknown graph settings, thereby offering more flexibility.

**Questions:**

My questions and concerns are listed above.

---

> ### Author Response · Authors · 2023-11-17
> **Response to Reviewer g68Y (1/2)**
>
> We thank the reviewer for considering our work as a new contribution to the literature and appreciate their positive
> perspective about our work.
>
> ## Modular training for c-component containing both low and high dimensional nodes
>
> The reviewer mentioned that the challenge we face while matching the full joint distribution containing low and high
> dimensional variable still exists if a c-component contains both the image and low-dimensional variables.
>
> Yes, we agree with the reviewer’s insight. However, please allow us to explain that even in these cases, there is
> utility in using our method compared to the existing work.
>
> Suppose, a causal graph has $N$ variables. Without modularization, we have to match the joint distribution containing
> $N$ (might be large) number of low and high dimensional variables in a single training phase. Matching that joint
> distribution with deep-learning models, and a complicated confounded causal structure could be difficult since we are
> attempting to minimize a very complicated loss function for a very large neural network. Our proposed method allows us
> to reduce the complexity of this problem tremendously by modularizing the training process to c-components. The size of
> a c-component is generally a lot smaller than the whole graph. Thus, even though we have to train mechanisms in a
> c-component together and match a joint distribution involving high and low dimensional variables, the complexity will be
> much lower. Without our approach, there is no existing work that can modularize and simplify the training process for a
> causal graph with latents.
>
> To back up our argument with empirical evidence, we have considered matching the joint distribution for the following
> graph.
>
> $I_1 \rightarrow Digit \rightarrow I_2 \rightarrow Color ; I_1 \leftrightarrow Color \leftrightarrow Digit$.
>
> Here $I_1$ and $I_2$ are image nodes and the rest are discrete.
> $I_1, Digit, Color$ belong to the same c-component.
>
> Our baseline NCM will attempt to match $P(I_1, D, I_2, C)$ by training mechanisms of all variables at the same time.
> Whereas, we i) first train $I_2$ to match $P(I_2|D)$ and then
> ii) train $I_1, D,C$ to match $P(I_1, D, I_2, C)$. At the first step, $I_2$ trains well. In the 2nd step, since we don’t
> have to train $I_2$ anymore, matching the joint gets easier. Below we report the image quality of our baseline and our
> method.
>
> FID scores representing the image quality of $I_1$ (lower is better):
>
> | Epochs/FID | 25     | 50     | 75    | 100   | 125   | 150   | 175   | 200   | 225   | 250   | 275   | 300   |
> |------------|--------|--------|-------|-------|-------|-------|-------|-------|-------|-------|-------|-------|
> | NCM        | 101.00 | 110.11 | 92.11 | 66.00 | 61.03 | 67.82 | 68.09 | 66.01 | 41.91 | 30.78 | 74.41 | 80.54 |
> | Ours       | 79.96  | 65.84  | 41.98 | 33.68 | 35.02 | 35.74 | 29.55 | 36.28 | 31.09 | 32.38 | 30.11 | 32.88 |
>
> FID scores representing the image quality of $I_2$ (lower is better):
>
> | Epochs/FID | 25     | 50    | 75    | 100   | 125   | 150   | 175   | 200   | 225   | 250   | 275   | 300   |
> |------------|--------|-------|-------|-------|-------|-------|-------|-------|-------|-------|-------|-------|
> | NCM        | 112.61 | 98.09 | 78.50 | 62.94 | 60.87 | 67.56 | 77.89 | 43.45 | 50.81 | 45.90 | 64.50 | 65.17 |
> | Ours       | 22.22  | 15.77 | 14.53 | 14.10 | 17.27 | 15.47 | 16.47 | 14.92 | 13.48 | 12.98 | 11.38 | 11.80 |
>
> The total variation distance for P(D, C) in both methods seems to be similar. But as we observe above, our method
> achieves better image quality than NCM.
>
> > "Still need to match $P(Z_1,Z_2,Z_3|X_1)$ and $P(X_1, X_2, Z_1, Z_3)$; does not simplify matching $P(
> > X_1,X_2,Z_1,Z_2,Z_3)$"
>
> We mention the causal graph here for the reviewer’s convenience: $Z_3 \rightarrow Z_1 \rightarrow Z_2; X_1 \rightarrow
> Z_1 \rightarrow X_2; Z_3 \leftrightarrow Z_1 \leftrightarrow Z_2; X_1 \leftrightarrow X_2.$
>
> To match the full joint distribution $P(X_1, X_2, Z_1, Z_2, Z_3)$, the joint training (NCM) has to train mechanisms of
> all $X_1, X_2, Z_1, Z_2, Z_3$ together at the same time. Whereas, in our approach, to match the joint distribution, we first
> train $Z_1, Z_2, Z_3$ at the first step and train $X_1, X_2$ at the 2nd step. Based on the above argument, the training
> complexity to match the joint distribution reduces in both steps compared to NCM.

---

> > ### Author Response · Authors · 2023-11-17
> > **Response to Reviewer g68Y (2/2)**
> >
> > ## Comparison with baseline with pre-trained model:
> >
> > > "Initializing the neural net architecture of the image node using a pre-trained model, followed by fine-tuning the
> > > joint model in an NCM fashion"
> >
> > We appreciate the reviewer for suggesting such an interesting baseline.
> > We kindly remind the reviewer that for the causal graph $D \rightarrow I \rightarrow A ; D \leftrightarrow A$, our
> > method i) first trains a model for $I$ and ii) uses this model or any other better pre-trained model with its weights
> > frozen to train only $D$ and $A$. Our
> > baseline NCM updates mechanisms of all $D, I, A$ variables at the same time with a single loss function.
> >
> > To show our algorithm performance according to the reviewer’s suggested setup, we plugged the same pre-trained model we
> > used for our method, in NCM’s architecture. Then we followed the NCM approach to train all the models for matching the
> > joint distribution. Our method and NCM both matched $P(D, A)$ quite well. However, even after using a smaller learning
> > rate (i.e., $1e-5$, compared to $D$ and $A$: $5*1e-4$) for the pre-trained model of $I$ in NCM, the generated image quality started to drop with
> > more training epochs. Here we provide the FID scores (lower is better) of both methods:
> >
> > | Epochs/FID | 0     | 25    | 50    | 75     | 100    | 125    | 150    | 175    | 200    | 225    | 250    | 275    | 300    |
> > |------------|-------|-------|-------|--------|--------|--------|--------|--------|--------|--------|--------|--------|--------|
> > | NCM        | 26.60 | 39.81 | 76.71 | 130.34 | 166.18 | 182.36 | 184.40 | 186.16 | 188.08 | 186.09 | 188.51 | 190.55 | 192.27 |
> > | Our method | 27.02 | 27.37 | 28.50 | 27.25  | 27.10  | 28.61  | 27.37  | 26.58  | 27.88  | 28.00  | 28.03  | 26.94  | 27.20  |
> >
> > On the other hand, since we are only training $D$, $A$ and perform no update on the pre-trained $I$, our image quality
> > did not drop. We believe this ablation study illustrates the significance of modular training, as the reviewer
> > suspected.
> >
> > ## True graph and joint training with unknown graph:
> >
> > > "High-dimensional deep causal models are required, the true causal graph may not be known"
> >
> > We agree with the reviewer that this is a valid concern. There is ongoing research on the learning representation of
> > high dimensional variables or learning causal relations among them [1]. However, for our work, we assume that the causal
> > graph is given as input to our algorithm. This graph might be an output of some causal discovery algorithm that
> > generates acyclic directed mixed graphs (ADMG) or might be constructed with expert help.
> >
> > Please note that our main goal is to identify causal and counterfactual effects and identifiability typically requires
> > the causal structure. Although there are cases when the equivalence class (family of graphs with the same property) is
> > sufficient for identifiability, most of the existing works such as [2] and [3] have to make this assumption as well. An
> > interesting future direction is to extend our results to the unknown graph setting by using the output (structure with
> > more uncertainty than ADMG) of an off-the-shelf causal discovery algorithm.
> >
> >
> > > "The joint training approach can easily adapt to unknown graph settings.".
> >
> > In our paper, by the term “joint training” we referred to the training process of our baseline [2] where they utilize a
> > single loss function to update the network weights of all variables. [2] assumes that the neural mechanisms of the
> > causal variables are arranged according to a given causal graph.
> >
> > Now, we believe that the reviewer used the “joint training approach” to refer to a setup where we don’t have any causal
> > graph and we train a single neural network to match some conditional distribution. That setup can only sample from
> > conditional distribution and not from interventional or counterfactual distribution since there is no identifiability
> > guarantee.
> >
> > We again thank the reviewer for their insightful comments. We hope we addressed all the concerns of our reviewer. We are
> > highly interested in having more discussions if the reviewer has any other concerns or follow-up comments.
> >
> > [1]Schölkopf et al. Toward causal representation learning." Proceedings of the IEEE 109.5 (2021): 612-634. \
> > [2]  Xia et al.(2023). Neural causal models for counterfactual identification and estimation\
> > [3]Kocaoglu et al. (2018). Causalgan: Learning causal implicit generative models with adversarial training

---

### Official Review · Reviewer_UCfX · 2023-11-04

**Soundness:** 2 fair
**Presentation:** 3 good
**Contribution:** 2 fair
**Rating:** 5
**Confidence:** 3

**Summary:**

This paper deals with factorization of joint distributions over variables when there are latent and high dimensional variables exist and how to use pre-trained neural networks to model complex distributions among variables.

The key is to determine which subgraphs can be modeled by one NN (to learn distribution) and in which order. The results are not surprising given existing results on interventional distribution and the factorization of ADMG, but it is novel in a way to combine with generative models and handle high-dimensional variables.

**Strengths:**

The method is novel in a way to combine with generative models and handle high-dimensional variables.

**Weaknesses:**

Some potential issues or weakness:
    1. Lack of discussions on assumption 3 and their violation, and some assumptions are hidden in the text besides assumption 1 to 3.
    2. What is the time efficiency of various approaches?

**Questions:**

Please see weakness.

---

> ### Author Response · Authors · 2023-11-17
> **Response to Reviewer UCfX (1/2)**
>
> We thank the reviewer for acknowledging our novelty in causal sampling involving high-dimensional variables. We happily
> address each of the mentioned points by the reviewer.
>
> ## Lack of discussions on assumption 3:
>
> > "Lack of discussions on assumption 3 and their violation"
>
> We apologize to the reviewer for not clearly explaining it. We will add a simpler statement about this assumption in the
> paper. For the reviewer’s convenience, we restate the assumption here.
>
> Assumption 3: for any h-node $H_k$ in the $\mathcal{H}$-graph, GAN training of DCM (our method) on dataset $\mathbf{D}$
> converges to sample from the conditional distribution $P(H_k , \mathcal{A}_k | {pa(H_k , \mathcal{A}_k)})$ for all
> $H_k$, where $\mathcal{A}_k$ is from Algorithm 2: Modular Training.
>
> Although the statement was written in a formal manner, assumption 3 simply says that we assume that GAN training
> converges and can match the conditional distribution after training on the given dataset. This is a common assumption
> used in many GAN papers in literature ([1], [2], [3], [5]). In our algorithm, we are training the set of all models
> modularly and matching corresponding conditional distributions. With sufficient representative power of the neural
> networks, and motivated by the success of deep generative models, we believe this assumption often holds in practice.
>
> If this assumption is violated, i.e., if the generator models do not converge properly, our algorithm will output a
> near-optimal solution. We expect this to only smoothly affect the causal queries of interest but it is a very
> interesting future direction that we are interested in analyzing, i.e., how imperfect distribution matching affects the
> interventional samples.
>
> ## Hidden assumptions in the text:
>
> > "Some assumptions are hidden in the text besides assumptions 1 to 3."
>
>
> We agree with the reviewer that we stated some assumptions in the text. However, these are basically restatements, and
> were already stated in Section 4.2 on page 7 of our paper. For convenience, we restate those main assumptions here:
>
> 1) The causal graph is given
> 2) The causal model is semi-Markovian and
> 3) GAN training can correctly learn the desired conditional data distribution.
>
> To address the reviewer’s comment, we will cross-reference in-text assumptions with those we listed above.
>
> In section 4, we assumed that the neural networks have sufficiently many parameters to induce the observed distribution
> induced by the true SCM. Please note that this is basically assumption 3: GAN convergence.
>
> Only in section 5.2, for the Covid X-ray experiment, we did not have access to the true causal graph (assumption 1 was
> violated). Thus, we had to fill its gap with some extra assumptions about the data distribution (no distribution shift
> and selection bias) and the causal relations among variables.

---

> > ### Author Response · Authors · 2023-11-17
> > **Response to Reviewer UCfX (2/2)**
> >
> > ## Time complexity:
> >
> > > "What is the time efficiency of various approaches?"
> >
> > The time efficiency/complexity of deep learning-based methods depends on different factors such as dataset size, chosen
> > hyperparameters, number of epochs, etc. In our case, for most of our experiments, we ran each method for 300 epochs. Our
> > datasets contained $20-40K$ samples, and the batch size was fixed at 200. For Wasserstein GAN with gradient penalty, we
> > used LAMBDA GP=10. We had learning rate = $5 * 1e - 4$. With this configuration, our method and the baselines took
> > around 3-4 hours to complete each experiment,
> >
> > Our proposed algorithm and the related works ([3], [4], [5]), that represent the structural causal model with neural networks, use one neural network per variable in the causal graph to mimic its mechanism. Therefore, for a fixed set of
> > hyperparameters, the main difference in time complexity for all these methods boils down to the complexity of achieving
> > convergence.
> >
> > Our closest baseline NCM [4] can mimic semi-Markovian causal models. This method trains all neural networks
> > corresponding to the mechanisms of all variables with a common loss function to match the joint distribution $P(V)$. On
> > the other hand, we train the same set of neural networks but modularly c-component by c-component. Since we train only
> > variables located in a single c-component during each training phase, this process allows us to optimize a comparatively
> > less complicated loss function while updating a smaller number of model weights compared to NCM as they update model
> > weights for the whole causal graph. As a result, we empirically observed that our method converges faster (as shown in
> > Figure 3c, Section 5.1) resulting in lower empirical time complexity compared to NCM.
> >
> > We hope we addressed all the concerns of our reviewer. We are highly interested in answering any further questions the
> > the reviewer might have.
> >
> > [1] Goodfellow et al. (2016). Generative adversarial networks. \
> > [2] Gulrajani et al. (2017). Improved training of Wasserstein gans.\
> > [3] Kocaoglu et al. (2018). Causalgan: Learning causal implicit generative models with adversarial training. In
> > International Conference
> > on Learning Representations.\
> > [4] Xia et al. (2023). Neural causal models for counterfactual
> > identification and estimation.\
> > [5] Pawlowski et al. (2020). Deep structural causal models for tractable counterfactual inference.

---

### Official Review · Reviewer_Zypu · 2023-11-07

**Soundness:** 2 fair
**Presentation:** 1 poor
**Contribution:** 2 fair
**Rating:** 3
**Confidence:** 3

**Summary:**

The paper presents an algorithm, WhatIfGAN, for modular learning of generative models for that can then be used for causal inference. WhatIfGAN identifies so-called H-groups of variables from a for which the do-calculus rule-2 holds and allows for separate training of generative models for the different groups. The paper compares the performance of WhatIfGAN to NCMs as well as a GAN without modular training. The results seem to indicate that WhatIfGAN generates superior high-dimensional counterfactuals.

[1] Kevin Xia, Kai-Zhan Lee, Yoshua Bengio, and Elias Bareinboim. The causal-neural connection: Expressiveness, learnability, and inference. Advances in Neural Information Processing Systems, 34:10823–10836, 2021.

**Strengths:**

The paper tackles the interesting and important problem of high-dimensional causal estimation with unobserved confounding. The authors propose a sound method for modularising the training of deep generative models which allows for efficient estimation of those causal queries. The experiments seem to indicate superior performance to relevant baselines.

**Weaknesses:**

The presentation of the paper could be improved including the presentation of the results and choice of experiments. The notation is inconsistent and changes from $Pa(..)$ to $pa(...)$ or from $P(...\mid do(x=..))$ to $P_x(...)$. The evaluation relies heavily on visual comparison of sampled data rather than quantifiable metrics. I suggest having a look at [1] for ideas of how to quantify counterfactual estimation capabilities for the CovidX dataset. The synthetic MNIST dataset can be validated against ground-truth distributions and samples using the TVD as done, or likelihood measurements or the likes. Also MSEs for ATE differences or CFs could be interesting.

[1] Monteiro, Miguel, et al. "Measuring axiomatic soundness of counterfactual image models." The Eleventh International Conference on Learning Representations. 2022.

**Questions:**

- Could this be achieved without modular training but by enforcing the noise to be the same between confounded variables? E.g. training $p(z_3)$ using an invertible function to find the noise to pass into training $p(z_1\mid z_3)$?
- It would be helpful to better introduce D and A in the MNIST use-case.
- I would encourage to also test the method on a more complicated synthetic dataset with more than 3 nodes.
- The paper would benefit from an introduction of TVD.

---

> ### Author Response · Authors · 2023-11-17
> **Response to Reviewer Zypu (1/2)**
>
> We are honored that the reviewer found the problem we are solving interesting and important. We also thank them for
> appreciating the soundness of our algorithm.
> Here we address the reviewer’s concerns below:
>
> ## Choice of experiments:
>
> > “I would encourage to also test the method on a more complicated synthetic dataset with more than 3 nodes.”
>
> We agree with the reviewer that the main paper considers causal graphs with 3 nodes. The intention was to illustrate the
> flexibility to utilize available pre-trained models (For example Covid-Xray model) and the significance of
> modularization. However, we would like to draw the reviewer’s attention to Appendix F of our paper where we also showed
> our algorithm performance on more complicated graphs.
> Here we list the causal graphs we used for our experiments more specifically:
>
> * 3 nodes: in Section 5.1, we highlight our flexibility to utilize pre-trained models and illustrate our performance
>   compared to baselines on the MNIST dataset.
> * 3 nodes: in Section 5.2, we experiment on the real-world COVIDx CXR-3 dataset with 30k samples that we associate with
>   a front door graph.
> * 9 nodes: in Appendix F.4, we showed our performance on a Colored-MNIST experiment where the causal graph contains 7
>   discrete variables and 2 image variables.
> * 6 nodes: in Appendix F.5, we experiment with the ASIA causal graph (6 nodes) from the popular bnlearn repository (
>   bnlearn.com/bnrepository).
> * 4 nodes: in Appendix F.6, we show our algorithm performance on the real-world Sachs protein dataset that includes both
>   observational and interventional data.
> * 7 nodes: in Appendix F.7, we show performance on a synthetic experiment with a causal graph with 7 nodes.
>
> We believe that our experiments clearly illustrate our motivation, contribution, and technical novelties. Having said
>   that if the reviewer has any specific synthetic/real-world datasets in mind, we would be happy to perform more
>   experiments and demonstrate our performance on them.
>
> ## Quantifiable metrics for image results:
>
> > “The evaluation relies heavily on visual comparison of sampled data rather than quantifiable metrics”.
>
> We agree with the reviewer that the evaluation relies on visual comparison. Since our aim was not to improve image
> quality but rather to consistently generate interventional and counterfactual samples, we modeled our synthetic dataset
> in such a way that visual inspection is enough to assess the correctness.
>
> To measure the distance between our predicted distribution and the ground truth, we used the total variation distance (
> TVD) and the KL divergence in all of our experiments. We would like to kindly point to our results in Appendix F (page
> 33), where we showed KL divergence as a metric for evaluation of our method.
>
> However, to improve our paper according to the reviewer’s suggestion, we added the popular Fréchet inception distance (
> FID) as our third metric that represents the quality of generated images. We compare our method with a baseline for the
> front door MNIST experiment (section 5.1).
>
> This table shows that we achieve better image quality (lower FID) with more
> training epochs compared to the non-modular baseline.
>
> | Epochs/FID     | 25     | 50     | 75    | 100   | 125   | 150   | 175   | 200   | 225   | 250   | 275   | 300   |
> |----------------|--------|--------|-------|-------|-------|-------|-------|-------|-------|-------|-------|-------|
> | Non-modular    | 151.41 | 110.02 | 91.71 | 74.95 | 86.77 | 78.69 | 74.38 | 76.62 | 77.86 | 74.32 | 66.96 | 80.65 |
> | Modular (Ours) | 74.34  | 59.26  | 59.19 | 59.56 | 63.05 | 59.46 | 60.81 | 63.11 | 63.58 | 61.05 | 61.41 | 62.83 |
>
> Note that Mean squared error (MSE) is typically used in regression and is not directly applicable to our method since we
> measure the distance between the true and fake joint distributions. Although the average treatment effect (ATE) is
> suitable for binary variables,
> in this simulation we deal with larger support sizes. For variables with larger support sizes, TVD and KL metrics
> contain more information about the distance between true and fake distribution.

---

> > ### Author Response · Authors · 2023-11-17
> > **Response to Reviewer Zypu (2/2)**
> >
> > ## Quantifying counterfactual estimation
> > We thank the reviewer for referring to the counterfactual evaluation metrics in [1].
> > We would like to kindly point out that the goal of the empirical analysis on the front door MNIST experiment in section 5.1
> > and the COVIDx CXR-3 dataset in section 5.2 is to illustrate our ability to sample from interventional distribution not
> > evaluate counterfactuals. Thus, the metrics in [1] might not be directly applicable to the COVID-19 CXR-3 dataset as the
> > reviewer suggested.
> >
> > However, we plan to follow the metrics used in [1] such as “composition” and evaluate our learned model performance.
> > In our front door MNIST causal graph: $D \rightarrow Image \rightarrow A; D \leftrightarrow A$, the image variable is caused by digit, $D$.
> > Based on our understanding, we can measure composition as:
> > $$\mathrm{composition}^{(m)}(\mathrm{image},\mathrm{digit}) = d_{X} (\mathrm{image}, \hat{f}^{(m)}(\mathrm{image}, \mathrm{digit}, \mathrm{digit})) $$
> > where, $d_{X}$ is some distance metric, $\hat{f}(.)$ is an approximation of the counterfactual function that helps us to obtain the
> > exogenous noise from observations of $\mathrm{image}$ and $\mathrm{digit}$ and again get back the corresponding image. It would be interesting to see if our model
> > can generate the same images with good quality after m-iterations of back and forth between observation and noise. We will include the
> > observations in the final version of the paper.
> >
> >
> > [1] Monteiro, Miguel, et al. “Measuring axiomatic soundness of counterfactual image models.” The Eleventh International
> > Conference on Learning Representations. 2022.
> >
> > ## Enforcing the unobserved noise to be the same:
> >
> > > “Could this be achieved without modular training but by enforcing the noise to be the same between confounded
> > > variables.
> > > E.g. training $P(Z_3)$ using an invertible function to find the noise to pass into training $P(Z_1|Z_3)$.”
> >
> > Note that for this, we need to assume that functions in the SCM are invertible, whereas our algorithm does not need to
> > make such an assumption. Ignoring this issue, below we demonstrate why we cannot train each confounded pair but have
> > to consider the whole c-components:
> > We restate part of the causal graph as: $Z_3 \rightarrow Z_1 \rightarrow Z_2, Z_3 \leftarrow U_1 \rightarrow Z_1, Z_1
> > \leftarrow U_2
> > \rightarrow Z_2$.
> >
> > We can map the reviewer’s suggestion to this graph, and it would be equivalent to estimating $U_1$ i.e., the noise
> > between
> > $Z_3$ and $Z_1$, and utilize that noise during training of $P(Z_1|Z_3)$.
> >
> > To identify $U_1$ we need all the variables that $U_1$ is dependent on. Since $U_1$ directly affects $Z_3$ and $Z_1$, we
> > at least need both $Z_3$ and $Z_1$. Now suppose, we only use $Z_3$ and $Z_1$ to identify $U_1$, i.e, we estimate $P(
> > U_1|Z_3,Z_1)$. However, conditioning on $Z_1$ makes $U_1$ dependent on $Z_2$ since $Z_1$ is a collider between $U_1$ and
> > $Z_2$. This means to identify $U_1$ we have to utilize $Z_2$ also. This chain of conditioning will make all the
> > variables in the same c-component dependent on $U_1$. Thus, only one child ($Z_3$) of $U_1$ is not enough to identify
> > $U_1$. In our algorithm, we train all the variables in the same c-component together to match the corresponding
> > distribution. This approach maintains the dependency among the variables due to the confounders.
> >
> > ## Variables in MNIST use-case:
> >
> > > “It would be helpful to better introduce D and A in the MNIST use-case.”
> >
> > We described the variable $D$ as the digit value of image $I$ and $A$ as some attribute of $I$ obtained from a randomly
> > chosen projection of the image. We chose such a projection to make sure that there is enough distance between the two
> > distributions: $P(Y|X)$ and $P(Y|do(X))$. This allowed us to illustrate our ability to generate interventional samples.
> > However, to introduce more meaningful relations among variables, we will add experiments where we consider color,
> > thickness, and rotation properties of MNIST digits and include the results in the camera-ready version of our paper.
> >
> > ## Definition and Changes in notations:
> >
> > > "Notation changes from i) $Pa()$ to $pa()$ or from ii) $P(...|do(x=...))$ to $P_x(...)$".
> >
> > We apologize for the notation inconsistencies in parts of the paper. We will of course correct these in the
> > camera-ready. We will also introduce the Total Variation Distance (TVD) and the Kullback–Leibler divergence (KL) in more
> > detail.
> >
> > We are deeply grateful for the reviewer’s time and feedback, which will undoubtedly improve our paper. We hope that we have
> > addressed all the reviewer’s concerns. We would be very happy to have further discussions if the reviewer has a follow-up
> > questions.

---

### Author Response · Authors · 2023-11-21
**A gentle reminder about the rebuttal phase discussion**

Dear AC and reviewers,

We are really grateful and cordially thank all of you for the effort and time you spent on our paper.

Your concerns allowed us to: \
i) critically analyze and clearly illustrate our contributions in practical scenarios, \
ii) add new evaluation metrics to our experiments, and \
iii) compare against different combinations of baselines and demonstrate that our method is superior.

We presented the above in greater detail in our responses to each of our reviewers. We would humbly request you to take a look at those and consider a re-evaluation of our paper’s merit and contributions. We are prepared to answer any of your questions.

Thank you.\
-The authors

---

### Meta-Review · Area_Chair_uR5b · 2023-12-08

**Metareview:**

The authors have proposed WhatIfGAN, an intriguing approach to estimating causal effects when the underlying causal graph is known. During review, most reviewers were on the fence about this paper. One reviewer pointed out some obvious baselines to consider, which, to the authors credit, were considered during the discussion. Ultimately this was a borderline decision, and it seems that a major revision and a new round of review incorporating the points from the reviewers and the new experiments is needed. Also, the decision to defer some of the most challenging and interesting experiments to the appendix was unfortunate, and I would suggest moving the simple 3-node cases to the appendix in favour of the more interesting examples. I encourage the authors to revise and resubmit their paper to another ML conference down the line.

**Note:** Although this did not affect the final decision, the authors may want to catch up on recent work that consider causal estimation without knowledge of the underlying true causal graph. There has been some progress on this in the past 1-2 years.

**Justification For Why Not Higher Score:**

See meta-review

**Justification For Why Not Lower Score:**

N/A

---

### Decision · Program_Chairs · 2024-01-16

Reject